# INTER-TASK LEARNING DYNAMICS IN DEEP LINEAR MULTI-TASK NETWORKS

## ABSTRACT

Despite significant empirical progress in Multi-Task Learning (MTL), the theo-
retical understanding of task interactions and their dynamics remains limited. We
present a theoretical analysis of how task alignment shapes learning dynamics in
linear MTL, providing a theoretical justification for why task importance is in-
herently dynamic and why loss weighting schemes should adapt during training.
Leveraging the Riccati formulation of gradient flow, we analytically characterize
the evolution and interaction of shared and task-specific components in deep lin-
ear neural networks. For a broad class of initializations, we show how task align-
ment and magnitude differences govern the trajectories of task outputs, losses,
and neural representations throughout training, as well as the representations at
convergence. Our analysis reveals that task alignment impacts learning speed and
modulates the relative importance of tasks throughout training, with magnitude
differences further amplifying these effects. We further show that these factors de-
termine how the structural relationships of the tasks are encoded at convergence in
deep linear networks. Our framework provides a principled comparison between
single-task and multi-task settings, grounded solely in data and task alignment.
These results establish a theoretical foundation for understanding task interac-
tions and pave the way toward principled approaches to adaptive loss weighting
and task grouping.

## 1 INTRODUCTION

Multi-Task Learning (MTL) (Caruana, 1997) has been successfully applied in Deep Learning (DL)
across a wide range of domains, including Computer Vision (Liu et al., 2019; Leang et al., 2020;
Zhang et al., 2023), Reinforcement Learning (He et al., 2024a; Sodhani et al., 2021) and Natural
Language Processing (Zhao et al., 2019; 2020; Chen et al., 2024). While learning multiple tasks
through a shared representation can improve generalization and data efficiency, negative interference
between tasks can lead to degraded performance compared to Single-Task Learning (STL) (Zhang
& Yang, 2021). As a result, there has been growing interest in *Task Grouping*, i.e., identifying
which tasks should be trained together (Standley et al., 2020; Fifty et al., 2021; Song et al., 2022;
Sherif et al., 2024), and *Dynamic Task Weighting Algorithms* that adapt each task's influence during
training (Liu et al., 2023; 2021; He et al., 2024b).

Despite progress on the empirical front, our theoretical understanding of task interactions in MTL
remains limited. Existing approaches often rely on computationally expensive methods based on
gradient-based heuristics, without a clear consensus on what task interactions truly are or when they
emerge (Xin et al., 2022). This important gap between empirical practice and theoretical insight
is further highlighted by recent work showing that simple heuristics, such as uniform or random
weighting, can rival the performance of sophisticated dynamic weighting methods (Elich et al.,
2023; Lin et al., 2022).

This work addresses this gap by examining how the (mis)alignment of tasks influences the dynamics
in linear multi-task networks, particularly in terms of learning speed, task importance and the nature
of the resulting neural representations. To this end, we integrate the notion of multiple tasks and
task-specific weights and losses into the continuous-time limit of gradient descent, known as *gra-
dient flow*. This continuous version of gradient descent has been extensively studied in single-task
literature for deep linear networks (Saxe et al., 2014; 2019) and extended to transfer and continual

learning settings (Lee et al., 2021; Lampinen & Ganguli, 2018) for different classes of initializations. Recently, the matrix *Riccati formulation* of gradient flow proposed by (Fukumizu, 1998) was connected to matrix factorisation problems to obtain analytical expressions for the evolution of key statistics and the network function over training in a two-layer linear network for a broad class of initializations (Braun et al., 2022; Tarmoun et al., 2021; Dominé et al., 2025). In this work, we integrate the notion of different tasks into the Riccati formulation and use the resulting analytical solutions to study the impact of training different tasks simultaneously on the learning dynamics.

Our analysis characterizes task (mis)alignment by examining differences in the singular vectors of the input-output correlation matrices, following (Lampinen & Ganguli, 2018), and show that the degree of alignment, together with differences in the task's magnitudes determines whether tasks are learned sequentially or simultaneously and determine how relative task importance evolves through learning. Leveraging the convergence properties of the analytical solutions obtained from the Riccati formulation, we compare the shared and task-specific representations at convergence with those obtained from training tasks independently and explain how alignment and magnitude differences shape the corresponding differences. These theoretical predictions are validated using deep linear networks trained on both synthetic and real datasets. Our results provide a principled lens for understanding task interactions and we believe these results will lead to insights for Dynamic Task Weighting Algorithms and Task Grouping strategies.

Prior work has demonstrated that insights from deep linear models can meaningfully transfer to more complex nonlinear settings (Saxe et al., 2014; Pennington et al., 2017; Arora et al., 2018), positioning our results as an important step toward understanding the dynamics of nonlinear MTL networks. We include additional experiments on deeper and nonlinear networks in Appendix D.

**Contributions.** This work makes the following key contributions:

- We analytically characterize the evolution of shared and task-specific components in linear multi-task networks under a broad class of initializations (Section (2) and (3)).

- We show that task alignment, amplified by magnitude differences, governs learning speed and induces time-varying relative task importance, providing a theory-driven motivation for dynamic loss weighting (Section 4).

- We develop predictive criteria from data statistics and task alignment to determine transfer and to compare single-task and multi-task representations at convergence (Section 5).

## 1.1 RELATED WORK

A more expansive Related Work section is included in Appendix A.

**Gradient flow.** The continuous-time limit of gradient descent, known as gradient flow, has been extensively studied as an approximation for understanding the training dynamics of neural networks. In the case of deep linear networks, exact analytical solutions under tabula rasa (random) initialization and whitened inputs were first introduced by Saxe et al. (2014). These solutions revealed key features of learning dynamics, including layer-wise alignment of weights and the sequential learning of input-output modes via singular values. Later works extended gradient flow analyses to more general settings, such as varying network architectures (Pinson et al., 2023; Saxe et al., 2022), alternative initialization schemes (Tarmoun et al., 2021), and structured input data (Gidel et al., 2019). Recently, a line of work has adopted a Riccati-based formulation originally proposed by Fukumizu (1998), where the gradient flow equations for two-layer linear networks are recast as solvable matrix differential equations (Yan et al., 1994). This approach has been extended to accommodate richer initialization structures (Tarmoun et al., 2021; Dominé et al., 2025), and task transfer scenarios (Braun et al., 2022), but has not yet been analyzed in MTL scenarios.

**Theoretical multi-task learning.** The theoretical understanding of MTL, especially in the context of neural networks, remains limited (Elich et al., 2023; Xin et al., 2022). Early work focused on kernel-based MTL (Xue et al., 2007) and states that MTL works well on similar tasks, where the notion of similarity is quantified based on the similarity of the corresponding single-task models. More recent studies, such as Wu et al. (2020), extends this notion of task similarity to neural networks where task data similarity is considered as a second-order factor that determines whether interference between multiple tasks can occur in MTL networks. Similarly, Lampinen & Ganguli

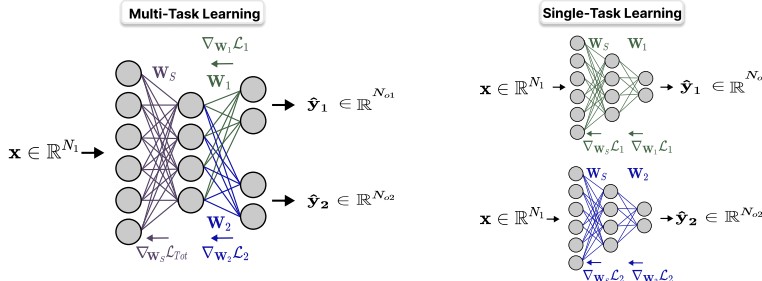

Figure 1: Difference between Multi-Task Learning (MTL, our setup) and Single-Task Learning (STL). MTL separates shared $\mathbf{W}_S$ (purple) and task-specific weights $\mathbf{W}_1$ (green), $\mathbf{W}_2$ (blue) with independent gradients $\nabla_{\mathbf{W}_1}\mathcal{L}_1$ and $\nabla_{\mathbf{W}_2}\mathcal{L}_2$. In STL, no explicit distinction is made between task(s).

(2018) quantifies task alignment using the singular value decomposition of the input-output correlation matrix, focusing on its impact on generalization performance under early stopping for low-rank teacher-student setups. Our work goes beyond this by extending the proposed notion of task alignment, including magnitude imbalance and describing their impact on the learning dynamics.

## 2 PRELIMINARY: MULTI-TASK LEARNING AND TASK ALIGNMENT

This work considers a Multi-Task Learning (MTL) (Ruder, 2017) setup in which the objective is to simultaneously learn $N_T$ tasks $\{\mathcal{T}_1, \ldots, \mathcal{T}_{N_T}\}$. All tasks share a common input vector $\boldsymbol{x} \in \mathbb{R}^{N_1}$, where $N_1$ is the input dimensionality, but each task $\mathcal{T}_i$ is associated with task-specific output vectors $\boldsymbol{y}_i \in \mathbb{R}^{N_{io}}$, where $N_{io}$ denotes the output dimensionality of task $\mathcal{T}_i$. To this end, a standard hard-parameter sharing architecture is adopted (Zhang & Yang, 2021). In this setup, the tasks are trained jointly using shared weight matrices $\mathbf{W}_S = \prod_{\ell=1}^{L_S} \mathbf{W}_S^{(\ell)}$ followed by $\mathcal{T}_i$-specific weight matrices $\mathbf{W}_i = \prod_{\ell=1}^{L_i} \mathbf{W}_i^{(\ell)}$. The MTL network is trained by minimizing a composite loss function $\mathcal{L}_{\text{Tot}}$, defined as a weighted sum of task-specific losses $\mathcal{L}_i$,

$$\mathcal{L}_{\text{Tot}}(t) = \sum_{i=1}^{N_T} \omega_i(t)\mathcal{L}_i(\mathbf{W}_i, \mathbf{W}_S, t), \tag{1}$$

where $\omega_i(t)$ denotes the time-dependent loss weighting coefficient for task $\mathcal{T}_i$, controlling its relative contribution to the overall loss at training iteration $t$. Here, $t$ corresponds to the number of epochs. We assume uniform static weighting $\omega_i = 1$ but discuss the impact of $\omega_i \neq 1$ in Appendix D.

In this work, the neural network is trained with full-batch gradient descent with a mean squared error (MSE) task-specific loss function $\mathcal{L}_i^{\text{MSE}}$ for each task,

$$\mathcal{L}_i^{\text{MSE}} = \frac{1}{N} \sum_{\mu=1}^{N} ||\boldsymbol{y}_i^\mu - \mathbf{W}_i\mathbf{W}_S\boldsymbol{x}^\mu||^2, \tag{2}$$

where $N$ is the number of input samples and $\boldsymbol{x}^\mu$, $\boldsymbol{y}_i^\mu$ denote the $\mu$-th input and corresponding target output for task $\mathcal{T}_i$. While the shared weights $\mathbf{W}_S$ are updated based on the gradient computed on the total loss $\mathcal{L}_{\text{Tot}}(t)$ (equation 1), the task-specific weights are updated through task-specific gradients computed on the task-specific losses $\mathcal{L}_i$ and learning rate $\alpha$ according to the update rule,

$$\mathbf{W}_S^\ell(t+1) \leftarrow \mathbf{W}_S^\ell(t) - \alpha\nabla_{\mathbf{W}_S^\ell(t)}\mathcal{L}_{Tot}(t), \qquad \mathbf{W}_i^\ell(t+1) \leftarrow \mathbf{W}_i^\ell(t) - \alpha\nabla_{\mathbf{W}_i^\ell(t)}\mathcal{L}_i(t), \tag{3}$$

such that $\forall i \neq j : \nabla_{\mathbf{W}_i}\mathcal{L}_j = 0$. As a result, task signals are explicitly disentangled in the task-specific layers, in contrast to single-task learning (STL), where task signals are not explicitly separated (Figure 1). We discuss the differences and applications to STL in Appendix B.2.

### 2.1 MTL GRADIENT FLOW EQUATIONS IN TWO-LAYER LINEAR NEURAL NETWORKS

In line with prior studies on the dynamics of single-task learning (Saxe et al., 2014; 2019; Braun et al., 2022), we analyse a two-layer linear MTL neural network. In this setup, the tasks are trained

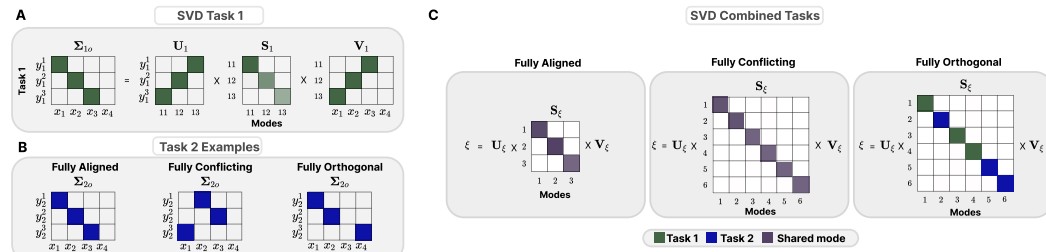

Figure 2: **A**. SVD of the input-output correlation matrix of Task 1. **B**. Example input-output correlation matrices for a fully aligned, conflicting, and orthogonal Task 2 (relative to Task 1). **C**. SVDs resulting from combining Task 1 with each distinct Task 2. Aligned tasks reinforce (shared modes strengthen), conflicting tasks interfere (modes mix across shared modes), and orthogonal tasks remain independent (each mode maps to one task).

jointly using one shared weight matrix $\mathbf{W}_S \in \mathbb{R}^{N_S \times N_1}$, where $N_S$ denotes the dimension of the shared layer, followed by task-specific weight matrices $\mathbf{W}_i \in \mathbb{R}^{N_{io} \times N_S}$ for each task $\mathcal{T}_i$ (Figure 1), yielding the task-i-specific predicted outputs,

$$\hat{\boldsymbol{y}}_i = \mathbf{W}_i \mathbf{W}_S \boldsymbol{x} \quad \forall i \in \{1, \dots, N_T\}. \tag{4}$$

In order to study the dynamics of the task-specific functions under gradient descent, we consider the continuous-time limit (i.e., assuming a small learning rate $\alpha \to 0$) of gradient descent. This limit, known as *gradient flow*, has been extensively studied in single-task literature for deep linear networks (Saxe et al., 2014; Tarmoun et al., 2021; Dominé et al., 2025). Taking the continuous-time limit of equation (3), while explicitly distinguishing different tasks yields,

$$\tau \frac{d\mathbf{W}_S}{dt} = \sum_{i=1}^{N_T} \mathbf{W}_i^\top (\boldsymbol{\Sigma}_{io} - \mathbf{W}_i \mathbf{W}_S \boldsymbol{\Sigma}_{11}), \quad \tau \frac{d\mathbf{W}_i}{dt} = (\boldsymbol{\Sigma}_{io} - \mathbf{W}_i \mathbf{W}_S \boldsymbol{\Sigma}_{11}) \mathbf{W}_S^\top \quad \forall i \in \{1, \dots, N_T\}, \tag{5}$$

where $\tau \equiv \frac{1}{\alpha}$ and the input-output correlation matrices are defined as,

$$\boldsymbol{\Sigma}_{11} \equiv \frac{1}{N} \sum_{\mu=1}^{N} \boldsymbol{x}^\mu (\boldsymbol{x}^\mu)^T \quad \boldsymbol{\Sigma}_{io} \equiv \frac{1}{N} \sum_{\mu=1}^{N} \boldsymbol{y}_i^\mu (\boldsymbol{x}^\mu)^T \quad \forall i \in \{1, \dots, N_T\}. \tag{6}$$

In section 3, we use solutions of the gradient flow equations derived for single-task learning in Fukumizu (1998); Braun et al. (2022) to solve the equations (5) while keeping the notion of different tasks explicit. We use these solutions to describe how different tasks impact each other's learning speed (section 4) and network representations (section 5). To this end, we introduce the notion of task alignment and magnitude imbalance in subsection 2.2.

## 2.2 TASK ALIGNMENT, MAGNITUDE IMBALANCE AND SHARED MTL SPECTRUM

Central to the formal definition of task alignment are the task-specific input-output correlation matrices $\boldsymbol{\Sigma}_{io}$ and their singular value decomposition (SVD),

$$\boldsymbol{\Sigma}_{io} = \mathbf{U}_i \mathbf{S}_i \mathbf{V}_i^T, \tag{7}$$

where $\mathbf{U}_i$ and $\mathbf{V}_i$ denote the left- and right- $\mathcal{T}_i$ singular vectors and $\mathbf{S}_i$ is the diagonal matrix containing the singular values $s_i^\alpha$ with $s_i^1 \geq s_i^2 \geq \cdots \geq s_i^{N_1}$ which capture the strength of $\mathcal{T}_i$-singular mode $\alpha$. Following Lampinen & Ganguli (2018), the amount of task alignment is quantified through the overlap matrix $\mathbf{O}_{12}$ defined below. We introduce three notions of task alignment central to the analysis.

**Definition 2.1** *Consider two tasks $\mathcal{T}_1$ and $\mathcal{T}_2$ defined by $\boldsymbol{\Sigma}_{1o}$ and $\boldsymbol{\Sigma}_{2o}$ and corresponding SVDs (equation 7). The overlap matrix $\mathbf{O}_{12}$ is defined as,*

$$\mathbf{O}_{12} = \mathbf{V}_1^T \mathbf{V}_2, \tag{8}$$

*and has singular values $(\alpha_1 \geq \alpha_2 \geq \cdots \geq \alpha_r \geq 0)$. Two tasks are said to be: (i) Fully Aligned if $\alpha_i = 1, \forall i$; (ii) Fully Orthogonal if $\alpha_i = 0 \ \forall i$ and, (iii) Fully Conflicting when $\forall i$ with $0 < \alpha_i < 1$.*

In Section 3, we show that the MTL gradient flow equation (Equation 5) can be reformulated and solved in terms of the combined task space $\xi$ and its singular value decomposition,

$$\xi = (\mathbf{\Sigma}_{1o} \quad \mathbf{\Sigma}_{2o})^T = \mathbf{U}_\xi \mathbf{S}_\xi \mathbf{V}_\xi^T. \tag{9}$$

The diagonal entries of $\mathbf{S}_\xi$ are given by the square roots of the eigenvalues of,

$$\mathbf{M} = \mathbf{V}_1 \mathbf{S}_1 \mathbf{V}_1^T + \mathbf{V}_2 \mathbf{S}_2 \mathbf{V}_2^T, \tag{10}$$

indicating that the spectrum $\mathbf{S}_\xi$ reflects the degree of (mis)-alignment between the singular modes of each task (Figure 2). For fully aligned tasks $\mathbf{V}_1 = \mathbf{V}_2 = \mathbf{V}$, such that $\mathbf{M} = \mathbf{V}(\mathbf{S}_1^2 + \mathbf{S}_2^2)\mathbf{V}^T$ and Rank($\mathbf{S}_\xi$)=Rank($\mathbf{S}_i$). In contrast, when the tasks are fully orthogonal $span(\mathbf{V}_1) \perp span(\mathbf{V}_2)$, such that the spectrum of eigenvalues corresponds to the ordered union of singular values from both tasks, i.e., Rank($\mathbf{S}_\xi$)=Rank($\mathbf{S}_1 \oplus \mathbf{S}_2$). For conflicting tasks, the singular modes are entangled in more complex ways, depending on the specific alignment between different singular directions. Note that task pairs in realistic datasets may have aligned, conflicting and orthogonal parts. As an example, we apply our analysis to the Multi-MNIST dataset Sabour et al. (2017) in Appendix D.5.

In addition to task alignment, we also analyze the effect of magnitude disparities across tasks. Specifically, we quantify relative scaling by computing ratios of Frobenius norms across task-specific dimensions. For dimensions $i$ and $j$ of two tasks $\mathcal{T}_1$ and $\mathcal{T}_2$, this is given by,

$$\Delta M_{12}^{(i,j)} = \frac{\|\mathbf{\Sigma}_{10}^i\|_F}{\|\mathbf{\Sigma}_{20}^j\|_F}. \tag{11}$$

Two tasks have *balanced magnitudes* when $\forall i \in \{0, N_{o1}\}, \forall j \in \{0, N_{o2}\} : \Delta M_{12}^{ij} = 1$.

## 3 RICCATI EQUATIONS AND INTER-TASK DYNAMICS

In this section, we adapt the Riccati-based formulation of gradient flow from Fukumizu (1998), extending the single-task analyses of Tarmoun et al. (2021); Braun et al. (2022) to the multi-task setting. Specifically, we write the first-order gradient flow dynamics (Equation 5) as coupled Riccati equations for two tasks ($N_T = 2$) and discuss the $N_T > 2$ case in Appendix C.

This formulation allows for an explicit computation of the evolution of the network's shared and task-specific representations. These quantities will be used in sections 4 and 5 to analyse how task alignment (Section 2.2) affects both learning speed and the structure of the learned representations.

In order to write the MTL dynamics as a Riccati matrix equation, define, $\mathbf{Q}_{MTL} = \begin{pmatrix} \mathbf{W}_S^T & \mathbf{W}_1 & \mathbf{W}_2 \end{pmatrix}^T$ and consider,

$$\mathbf{Q}_{MTL}\mathbf{Q}_{MTL}^T(t) = \begin{pmatrix} \mathbf{W}_S^T\mathbf{W}_S(t) & \mathbf{W}_S^T\mathbf{W}_1^T(t) & \mathbf{W}_S^T\mathbf{W}_2^T(t) \\ \mathbf{W}_1\mathbf{W}_S(t) & \mathbf{W}_1\mathbf{W}_1^T(t) & \mathbf{W}_1\mathbf{W}_2^T(t) \\ \mathbf{W}_2\mathbf{W}_S(t) & \mathbf{W}_2\mathbf{W}_1^T(t) & \mathbf{W}_2\mathbf{W}_2^T(t) \end{pmatrix} , \ \mathbf{R}_{MTL} = \begin{pmatrix} 0 & \mathbf{\Sigma}_{1o}^T & \mathbf{\Sigma}_{2o}^T \\ \mathbf{\Sigma}_{1o} & 0 & 0 \\ \mathbf{\Sigma}_{2o} & 0 & 0 \end{pmatrix}, \tag{12}$$

such that, $\mathbf{Q}_{MTL}\mathbf{Q}_{MTL}^T(t)$ contains all necessary information to compute task-specific network functions $\mathbf{W}_i\mathbf{W}_S(t)$. Lastly, define $\mathbf{\Gamma} = \gamma\mathbf{I}$, such that $\mathbf{\Gamma}$ is a constant matrix describing the difference in covariance between the shared and task-specific weights.

The derivation of the Ricatti equation and its analytical solution rely on the standard assumptions made in single-task learning (Braun et al., 2022; Dominé et al., 2025; Tarmoun et al., 2021; Fukumizu, 1998):

A1 The inputs are *whitened*, i.e., $\mathbf{\Sigma}_{11} = \mathbf{I}$.

A2 The shared and task-specific weights are $\mathbf{\Gamma}$-*balanced* at initialization such that $\mathbf{W}_S(0)\mathbf{W}_S(0)^T - \mathbf{W}_1^T(0)\mathbf{W}_1(0) - \mathbf{W}_2(0)^T\mathbf{W}_2(0)=\mathbf{\Gamma}$ or *Zero-balanced* at initialization such that $\mathbf{W}_S(0)\mathbf{W}_S(0)^T = \mathbf{W}_1^T(0)\mathbf{W}_1(0) + \mathbf{W}_2(0)^T\mathbf{W}_2(0)$.

A3 No bottleneck: $\forall i \in \{1, 2\} :$ Rank($\mathbf{\Sigma}_{io}$)=$min(N_1, N_{io})$, ensuring the shared space can hold all the task-relevant input-output modes for both tasks $N_s \geq min(N_1, N_{1o} + N_{2o})$.

Under these assumptions, the MTL dynamics governed by $\mathbf{Q}_{MTL}\mathbf{Q}_{MTL}^T$ and $\mathbf{R}_{MTL}$ can be described by a system of Matrix Riccati equations, as formalized in Lemma 1. A detailed discussion of these assumptions, along with possible relaxations, is provided in Dominé et al. (2025).

**Lemma 1** *Under assumptions A1 (whitened inputs) and A2 ($\Gamma$-balanced weights) the continuous-time dynamics of the matrix $\mathbf{Q}_{MTL}\mathbf{Q}_{MTL}^T(t)$ from an initial state $\mathbf{Q}_{MTL}\mathbf{Q}_{MTL}^T(0)$ are captured by the matrix Riccati Equation,*

$$\tau\frac{d(\mathbf{Q}_{MTL}\mathbf{Q}_{MTL}^T)}{dt} = \mathbf{R}_{MTL}\mathbf{Q}_{MTL}\mathbf{Q}_{MTL}^T + \mathbf{Q}_{MTL}\mathbf{Q}_{MTL}^T\mathbf{R}_{MTL} - (\mathbf{Q}_{MTL}\mathbf{Q}_{MTL}^T)^2 \quad (13)$$

$$- \mathbf{Q}_{MTL}^T\mathbf{\Gamma}\mathbf{Q}_{MTL}, \quad (14)$$

The proof of Lemma 1 is presented in Appendix C.2.

In order to obtain an interpretable solution of the Riccati equation (14), we define,

$$\xi = \begin{pmatrix} \mathbf{\Sigma}_{1o} & \mathbf{\Sigma}_{2o} \end{pmatrix}^T \quad \text{and} \quad \mathbf{W}_{MTL}^0 = \begin{pmatrix} \mathbf{W}_1(0)\mathbf{W}_S(0) & \mathbf{W}_2(0)\mathbf{W}_S(0) \end{pmatrix}^T \quad (15)$$

such that $\xi$ and $\mathbf{W}_{MTL}^0 \in \mathcal{R}^{N_{1o}+N_{2o}\times N_1}$, allows us to consider the compact singular value decomposition of the initial network and the input-output correlation matrix of the combined task space,

$$\xi = \mathbf{U}_\xi\mathbf{S}_\xi\mathbf{V}_\xi^T \quad \mathbf{W}_{MTL}^0 = \mathbf{U}_0\mathbf{S}_0\mathbf{V}_0. \quad (16)$$

In general, when $N_{1o} + N_{2o} \neq N_1$, $\mathbf{U}_\xi$ and $\mathbf{V}_\xi^T$ are not square and orthogonal. For cases where $N_{1o}+N_{2o} \neq N_1$, we use the approach in Braun et al. (2022) and construct $\mathbf{U}_\xi^\perp$ and $\mathbf{V}_\xi^\perp$ to complete the basis as explained in Appendix C.3. Finally introducing $\mathbf{A}_\xi$ and $\mathbf{B}_\xi$,

$$\mathbf{A}_\xi = \mathbf{U}_0^T\mathbf{U}_\xi + \mathbf{V}_0^T\mathbf{V}_\xi \quad \mathbf{B}_\xi = \mathbf{U}_0^T\mathbf{U}_\xi - \mathbf{V}_0^T\mathbf{V}_\xi, \quad (17)$$

allows a solution for $\mathbf{Q}_{MTL}\mathbf{Q}_{MTL}^T(t)$ as formulated in the next Theorem 2.

**Theorem 2** *Under assumptions A1 (whitened inputs), A2 (zero-balanced weights), A3 (full-rank) and non-singular $\mathbf{A}_\xi$, the MTL Riccati equation (14) has a unique solution $\forall t > 0$, given by,*

$$\mathbf{Q}_{MTL}\mathbf{Q}_{MTL}^T(t) \quad (18)$$

$$= \mathbf{Z}_{MTL}\Bigg[ 4\,e^{-\mathbf{S}_\xi\frac{t}{\tau}}\,\mathbf{A}_\xi^{-1}\,\mathbf{S}_\xi^{-1}\,(\mathbf{A}_\xi^T)^{-1}\,e^{-\mathbf{S}_\xi\frac{t}{\tau}} + \left(\mathcal{I} - e^{-2\mathbf{S}_\xi\frac{t}{\tau}}\right)\mathbf{S}_\xi^{-1} \quad (19)$$

$$- e^{-\mathbf{S}_\xi\frac{t}{\tau}}\,\mathbf{A}_\xi^{-1}\,\mathbf{B}\left(e^{-2\mathbf{S}_\xi\frac{t}{\tau}} - \mathcal{I}\right)\mathbf{S}_\xi^{-1}\mathbf{B}^T\,(\mathbf{A}_\xi^T)^{-1}\,e^{-\mathbf{S}_\xi\frac{t}{\tau}} \quad (20)$$

$$+ 4\frac{t}{\tau}e^{-\mathbf{S}_\xi\frac{t}{\tau}}\,\mathbf{A}_\xi^{-1}\left(\mathbf{V}_\xi^T\,\mathbf{V}_\xi^\perp(\mathbf{V}_\xi^\perp)^T\,\mathbf{V}_\xi + \mathbf{U}_\xi^T\,\mathbf{U}_\xi^\perp(\mathbf{U}_\xi^\perp)^T\,\mathbf{U}_\xi\right)(\mathbf{A}_\xi^T)^{-1}\,e^{-\mathbf{S}_\xi\frac{t}{\tau}}\Bigg]^{-1}\mathbf{Z}_{MTL}^T, \quad (21)$$

$$\mathbf{Z}_{MTL} = \begin{pmatrix} \mathbf{V}_\xi\left(\mathcal{I} - e^{-\mathbf{S}_\xi\frac{t}{\tau}}\mathbf{B}_\xi^T(\mathbf{A}_\xi^T)^{-1}e^{-\mathbf{S}_\xi\frac{t}{\tau}}\right) + 2\mathbf{V}_\xi^\perp(\mathbf{V}_\xi^\perp)^T\mathbf{V}_\xi(\mathbf{A}_\xi^T)^{-1}e^{-\mathbf{S}_\xi\frac{t}{\tau}} \\ \mathbf{U}_\xi\left(\mathcal{I} + e^{-\mathbf{S}_\xi\frac{t}{\tau}}\mathbf{B}_\xi^T(\mathbf{A}_\xi^T)^{-1}e^{-\mathbf{S}_\xi\frac{t}{\tau}}\right) + 2\mathbf{U}_\xi^\perp(\mathbf{U}_\xi^\perp)^T\mathbf{U}_\xi(\mathbf{A}_\xi^T)^{-1}e^{-\mathbf{S}_\xi\frac{t}{\tau}} \end{pmatrix}. \quad (22)$$

The proof of Theorem 2 is provided in Braun et al. (2022), and the necessary steps to incorporate multiple tasks are outlined in Appendix C.3.

In the following sections, we apply the result from Theorem 2 to analyse how task alignment and magnitude imbalances (section 2.2) impact learning dynamics (section 4), and the neural representations at convergence (section 5) in linear MTL networks. All experimental details are provided in Appendix D, and the corresponding code is included in the supplemental material and will be made available online[1].

## 4 TASK ALIGNMENT, MAGNITUDE DIFFERENCES AND LEARNING DYNAMICS

From single-task literature (Braun et al., 2022; Saxe et al., 2014), we know that a linear network's input-output map sequentially learns the task-specific singular modes. The speed at which each

---

[1]A link to the GitHub repository will be shared here upon acceptance.

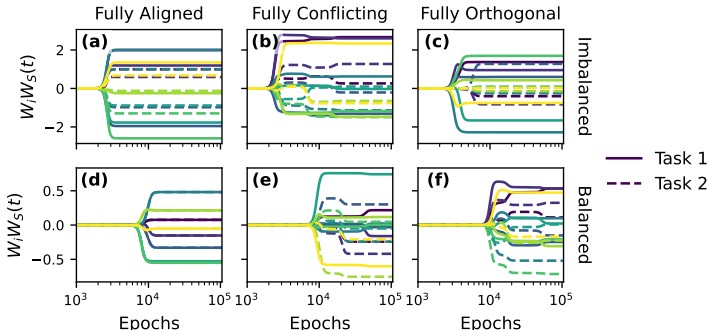

Figure 3: Simulations of the task-specific network functions $\mathbf{W}_i\mathbf{W}_S(t)$ based on Theorem 2 of a two-layer linear network with $N_1 = 4, N_{io} = 2, N_S = 4$ and $\alpha = 10^{-3}$ trained on Fully Aligned (**a**)-(**d**), Conflicting (**b**)-(**e**) and Orthogonal tasks (**c**)-(**d**). Comparison of the components of $\mathbf{W}_i\mathbf{W}_S(t)$ for Task 1 (full line) and Task 2 (dotted line) for pairs without (**top**) and with balanced (**bottom**) magnitudes. Task alignment determines whether all (**a**)-(**d**) or some (**b**) components converge simultaneously, or sequentially. (**c**)Imbalance in magnitudes further amplify these effects.

singular mode $\alpha$ is learned is dependent on the strength of the mode, which is reflected by the corresponding singular value $s_\alpha$. Theorem 2 shows that in MTL, these singular values are effectively captured by the diagonal matrix $\mathbf{S}_\xi$ of the shared MTL spectrum. As a result, the learning speed of the singular values and the evolution of the task-specific network functions are determined by the alignment of the tasks (section 2.2). In addition, prior MTL work indicates that unequal task scaling can degrade performance (Chen et al., 2018). We therefore examine whether magnitude disparities between tasks (i.e., $\Delta M_{1,2} \neq 1$) bias the training trajectory toward the higher-magnitude task, causing it to dominate learning and how this effect interferes with alignment.

To study alignment and magnitude effects, we use the analytical solution of Theorem 2 to simulate learning dynamics (Figure 3). We construct task pairs with controlled alignment and fixed singular values, analyzing dynamics with (top) and without imbalanced magnitudes (bottom). Predictions are validated by task-specific losses from training a linear MTL network on imbalanced tasks with varying alignment (Figure 4). More experiments and details are included in Appendix D.

**Perfect alignment enables simultaneous convergence.** Since each singular value in $\mathbf{S}_\xi$ is the result of the sum of two task-specific singular values $s_i^\alpha$, each mode equally contributes to the network components of both tasks. In Figure 3, plots (**a-d**), we see that as a result, both task 1 (full line) and 2 (dotted line) start learning and converge at the same time, even when there is a difference in the magnitudes (Figure 3**a**). As seen on Figure 4**a**, this results in a perfectly simultaneous evolution of the task-specific losses in a linear neural network.

**Orthogonality implies task-by-task mode learning.** Because singular modes of orthogonal tasks do not couple, each singular mode in $\mathbf{S}_\xi$ is only picked up by one task's network function while the other remains unchanged. If all singular modes of one task project more strongly into the shared space, that task converges first (Figure 3**c**), delaying learning of the other. As shown in Figure 4**f**, this produces a clear gap between the convergence of the two task-specific losses. When task magnitudes and projected values are similar, this delay becomes negligible, and the modes are learned in rapid succession with minimal delay (Figure 3**f**).

**Conflicting tasks might confuse network functions.** For fully conflicting tasks, all singular modes are shared, which allows both task functions to begin evolving simultaneously (Figure 4**b**) and initially follow a common trajectory before diverging (Figure 3**b**-3**e**). Large differences in how these modes project onto the shared spectrum $\mathbf{S}_\xi$ yield imbalanced contributions to the task-specific functions, producing phases in which one task learns faster and temporarily dominates (Figures 3**b**-4**b**).

**Theoretical motivation for weighting tasks dynamically.** Based on the previous discussions, we learn that for orthogonal and conflicting tasks the network might focus one task for a while and then turn back to another task. As a result, some tasks may be temporarily neglected or dominate learning at different stages of training. Over time, a task that was initially harmless may later interfere

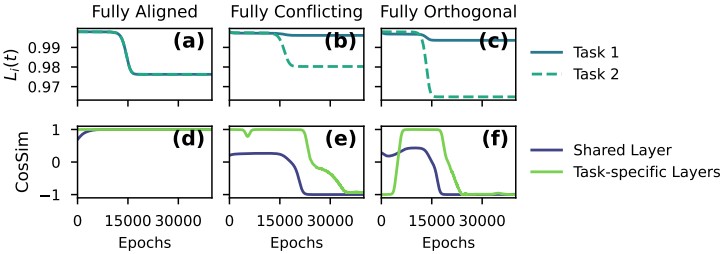

Figure 4: Comparison of the evolution of the task-specific losses (top) and cosine similarities (bottom) resulting from training a two-layer linear MTL network with $N_1 = 20$, $N_{o1} = N_{o2} = 1$, $N_S = 2$ on fully Aligned (**a**, **d**), Conflicting (**b**, **e** ) and Orthogonal Regression tasks (**c**, **f**) with a fixed difference in magnitude between for all task pairs $\Delta M_{12}^{ij} = 0.25$. $\mathcal{L}_2(t)$-values are rescaled after training ($\div 16$) for clarity. (**a**) For aligned tasks, both losses converge together and the cosine similarity remains positive, unlike in the conflicting (**e**) and orthogonal (**f**) cases. (**b**) For conflicting tasks, both losses begin together, but Task 1 stalls before converging later. (**c**) For orthogonal tasks, Task 1 begins converging only after Task 2 has already converged.

with or suppress the learning of others. This indicates that weighting task-specific losses (equation 1) should happen in a dynamical way when tasks are not aligned. Consistent with this view, the cosine similarities between task gradients with respect to the shared and task-specific network layers (Figure 4, bottom) remain high and stable for aligned tasks, but vary strongly, and even become negative, for conflicting (**e**) and orthogonal (**f**) tasks, reflecting the negative interference widely reported in MTL literature Liu et al. (2021); Zhang & Yang (2021).

## 5 SHARED AND TASK SPECIFIC REPRESENATIONS AT CONVERGENCE

In this section, we examine the impact of task-alignment on the MTL network representations at convergence. To this end, we use the following convergence Theorem 3.

**Theorem 3** *Under the assumptions of Theorem 2, $\mathbf{Q}_{MTL}\mathbf{Q}_{MTL}^T(t)$ converges to a steady state given by,*

$$\mathbf{Q}_{MTL}\mathbf{Q}_{MTL}^T(t_{steady}) = \begin{pmatrix} \mathbf{V}_\xi \mathbf{S}_\xi \mathbf{V}_\xi^T & \mathbf{V}_\xi \mathbf{S}_\xi \mathbf{U}_\xi^T \\ \mathbf{U}_\xi \mathbf{S}_\xi \mathbf{V}_\xi^T & \mathbf{U}_\xi \mathbf{S}_\xi \mathbf{U}_\xi^T \end{pmatrix}. \tag{23}$$

As explained in Braun et al. (2022), this result is a direct consequence of Theorem 2. For completeness, the proof is included in Appendix C.4.

Theorem 3 states that the Gram Matrices of the shared and task-specific weights ($\mathbf{W}_S\mathbf{W}_S^T$ and $\mathbf{W}_i\mathbf{W}_i^T$ respectively), converge to the Representation Similarity Matrices (RSM) of the combined task data $\xi$, i.e.,

$$\lim_{t \to t_{steady}} \mathbf{W}_S^T\mathbf{W}_S(t) = \left(\mathbf{V}_\xi \mathbf{S}_\xi \mathbf{V}_\xi^T\right) , \lim_{t \to t_{steady}} \mathbf{W}_i\mathbf{W}_i^T(t) = \left(\mathbf{U}_\xi \mathbf{S}_\xi \mathbf{U}_\xi^T\right)_{[N_{(i-1)0}:N_{i0}, N_{(i-1)0}:N_{i0}]}. \tag{24}$$

Here, $\mathbf{V}_\xi \mathbf{S}_\xi \mathbf{V}_\xi^T$ and $\mathbf{U}_\xi \mathbf{S}_\xi \mathbf{U}_\xi^T$, are the RSMs of the combined input and output data respectively. These matrices encode the similarities between different input and output dimensions and hence represent structural relationships in the input or output data. In Figure 5, we show that task alignment and magnitude imbalance affect the RSMs of the combined input and output data. As a consequence of Theorem 3, these factors also impact the way the network's weights reflect structural relationships from the different tasks.

Theorem 3 also enables a direct theoretical comparison between the task-specific and shared representations learned in MTL and those learned when training each task independently without requiring model training. Namely, from the convergence theorem in Braun et al. (2022), it follows that the hidden and output weights in a STL network initialized with zero-balanced weights converge to the

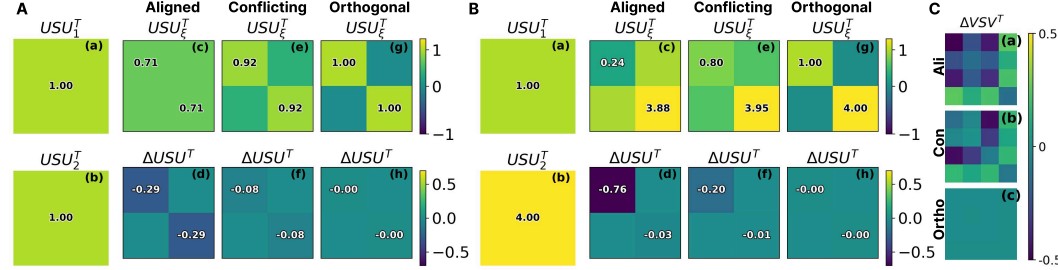

Figure 5: Comparison of RSMs for regression task pairs with different alignment. **A**. Balanced tasks, where single task RSMs $\mathbf{U}_i\mathbf{S}_i\mathbf{U}_i^T$ combine into $\mathbf{U}_\xi\mathbf{S}_\xi\mathbf{U}_\xi^T$. Their corresponding difference $\Delta\mathbf{USU}^T$ shows weaker structural relationships for fully aligned and conflicting tasks, but no effect from fully orthogonal tasks as compared to the single task case. **B**. These effects are amplified by a magnitude difference $\Delta M_{12} = 0.25$. **C**. $\Delta\mathbf{VSV}^T$. Aligned tasks interfere and create different structural relationships, conflicting tasks show intermediate effects, while for orthogonal tasks, this yields exactly the direct sum between the task specific RSMs.

task-i-specific RSMs of the input $(\mathbf{V}_i\mathbf{S}_i\mathbf{V}_i^T)$ and the output $(\mathbf{U}_i\mathbf{S}_i\mathbf{U}_i^T)$,

$$\lim_{t\to t_{steady}} \mathbf{W}_S^{T,STL}\mathbf{W}_S^{STL}(t) = \left(\mathbf{V}_i\mathbf{S}_i\mathbf{V}_i^T\right) \quad \lim_{t\to t_{steady}} \mathbf{W}_i^{STL}\mathbf{W}_i^{T,STL}(t) = \left(\mathbf{U}_i\mathbf{S}_i\mathbf{U}_i^T\right). \quad (25)$$

As a consequence, Equations (24) and (25) can be used to compare the way a linear MTL neural network reflects structural relationships at convergence, when it is trained on several tasks simultaneously, as compared to the situation where it is trained on each task separately. In Figures **5A** and **5B**, we visualize the difference between the RSMs of the output data by defining $\Delta\mathbf{USU}^T = (\mathbf{U}_\xi\mathbf{S}_\xi\mathbf{U}_\xi^T)_{[:N_{o1},:N_{o1}]} - \mathbf{U}_1\mathbf{S}_1\mathbf{U}_1^T$. To quantify how strong the combined structure deviates from the direct sum of the task specific RSMs of the input data, we define $\Delta\mathbf{VSV}^T = \mathbf{VSV}_\xi^T - \mathbf{V}_1\mathbf{S}_1\mathbf{V}_1^T - \mathbf{V}_2\mathbf{S}_2\mathbf{V}_2^T$ (Figure **5C**).

**Impact of task alignment on neural representations.** From Figure **5A** we learn that, relative to the single taks case, the RSMs of the task-specific layers $\mathbf{W}_i\mathbf{W}_i^T$ reflect weaker correlations $\Delta\mathbf{USU}_{i,j}^T < 0$ for aligned and conflicting tasks, and are identical $\Delta\mathbf{USU}_{i,j}^T = 0$ to the single-task case for orthogonal tasks. Figure **5C** shows that, relative to the representations learned when each task is trained in isolation (Equation 25), the shared hidden representation $\mathbf{W}_S^T\mathbf{W}_S$ reflects new or stronger structural relationships $\Delta\mathbf{VSV}_{i,j}^T \neq 0$ for aligned tasks pairs (a), similar ($\Delta\mathbf{VSV}_{i,j}^T \approx 0$) or new $\Delta\mathbf{VSV}_{i,j}^T \neq 0$ relationships for conflicting tasks (b). For orthogonal tasks, the combined spectrum is exactly equal to the sum of the task-i specific input RSMs ($\Delta\mathbf{VSV}_{i,j}^T = 0$), indicating the absence of new structural relationships.

**Impact of scale differences.** Comparing Figures **5A** and **5B** shows that task-magnitude imbalance biases task-specific representations for aligned or conflicting tasks. When the dimensions of task 2 have a larger magnitude than task 1 ($\Delta M_{12} < 1$), correlations in the task-1–specific block of the joint RSM (top-left) are further attenuated relative to the equal-scale case (Figure **5A**), even though task 1 is unchanged (Figure **5B**). By contrast, for orthogonal tasks, this block remains invariant to magnitude changes. Appendix D examines how the loss-weighting coefficients in equation (1) modulate this effect.

## 6 APPLICATION TO THE MULTI-MNIST DATASET

In this section, we compare the theoretical predictions from the previous sections with the behaviour of a two-layer linear network trained on two variants of the Multi-MNIST dataset. Multi-MNIST is a multi-task extension of MNIST Lecun et al. (1998), introduced by Sabour et al. (2017) and widely used in MTL benchmarks. It is constructed by overlaying pairs of MNIST digits, with the left and right digit classifications forming two separate tasks. To study the impact of alignment, we use the original dataset and a permuted variant designed to modify the task alignment, referred to as the *Baseline* and *Permuted* Multi-MNIST datasets, respectively. Further details on both variants are provided in the Appendix. D.5.

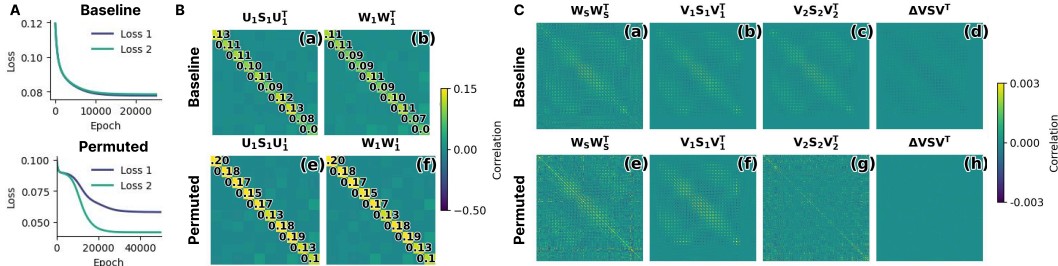

Figure 6: Learning dynamics and RSMs for a linear two-layer MTL network ($N_s = 20, \alpha = 10^{-2}$) trained on the Baseline (top) and Permuted (bottom) Multi-MNIST datasets. **A.** Learning curves for two tasks, showing simultaneous learning in the Baseline configuration and sequential learning in the Permuted configuration. **B.** Comparison of single-task RSMs $\mathbf{U}_i\mathbf{S}_i\mathbf{U}_i^T$ (a) and (c) with the RSMs of the task-specific network weights at convergence (b) and (d), revealing weaker structural relationships for the Baseline and matching STL structure for the Permuted case. **C.** RSMs of shared neural representations (a) and (e), showing deviations from single-task RSMs in the Baseline configuration (b)-(c), and matching the sum of STL RSMs in the Permuted configuration (f)-(g).

**Alignment in Real-World Datasets.**    In realistic settings, task relationships rarely fall into the strictly aligned, conflicting, or orthogonal cases (Definition 2.1). Instead, the placement of the singular values $\alpha_i$ of $\mathbf{O}_{12}$ near 1, 0.5, or 0 indicates whether tasks are mostly aligned, conflicting, or orthogonal. To illustrate how the previous analyses apply, we study the learning dynamics and convergence behaviour of the Baseline Multi-MNIST configuration ($\alpha_i \in [0.92, 1]$), corresponding to a predominantly aligned setting, and the Permuted Multi-MNIST configuration ($\alpha_i \in [0.040, 1]$, with most values in $[0.254, 0.37]$), representing a regime between conflicting and orthogonal directions.

**Learning Dynamics.**    Figure 6**A** shows the learning dynamics after training a two-layer linear MTL network on the Baseline (top) and Permuted (bottom) Multi-MNIST datasets. As predicted from their high alignment, the Baseline tasks converge nearly simultaneously (Figure 6**A**, top). In contrast, tasks in the Permuted configuration start learning together due to their shared aligned mode ($\alpha_1 = 1$), but then transition to a mix of simultaneous and sequential learning. As shown in Figure 6**A**(bottom), the losses initially decrease together before separating, with the second task converging first. A similar pattern is observed with a ReLU network (Appendix D.5.1).

**Convergence Behaviour.**    In Figure 6**B-C**, we compare the predictions of Section 5 with the RSMs of the neural representations for both the task-specific weights (Figure 6**B**) and the shared weights (Figure 6**C**). For the Baseline configuration (Figure 6**B**, top), the network (b) captures weaker structural relationships than the STL case (a) would (i.e., $\Delta\mathbf{USU}_{i,j}^T < 0$), whereas for the Permuted configuration (Figure 6**B**, bottom), it recovers the STL relationships (c) as expected for orthogonal tasks ($\Delta\mathbf{USU}_{i,j}^T = 0$). Figure 6**C** shows that the Baseline tasks induce stronger or additional structural relationships in the MTL network (a), while for the Permuted variant, the shared representation RSM (e) matches the sum of the single-task RSMs (f-g) (i.e., $\Delta\mathbf{VSV}_{i,j}^T = 0$) further confirming the behaviours predicted for aligned and orthogonal tasks.

# 7 DISCUSSION

This work extends the Riccati framework to multi-task learning, offering a principled analysis of the dynamics and task interactions in deep linear MTL. We show that task alignment and magnitude imbalance shape how tasks interact: orthogonal or conflicting components create time-dependent shifts in task importance through learning lags, whereas aligned components evolve simultaneously even under magnitude differences. At the same time, the way the structural relationships are represented by the neural network remains robust to imbalance for orthogonal tasks, while aligned and conflicting tasks are strongly affected. The analysis is restricted to deep linear networks and relies on simplifying assumptions (1-3), which may not fully capture the complexity of real-world non-linear systems. Despite these simplifications, we believe our framework offers a first step toward a data-driven understanding of task interactions and can guide task grouping and dynamic weighting beyond the linear setting.

**Reproducibility Statement.** We aim to make our results fully reproducible. All mathematical derivations and proofs are given in full in Appendix C, and we provide additional related work (Appendix A) together with concise summaries of prior results we build on. For every experiment, we detail the data-generation pipeline, model architectures, training procedures, and hyperparameters in Appendix D and present the specific configurations used to generate the figures of the main paper in Appendix E. Upon acceptance, we will release the complete codebase and an executable notebook that regenerates all figures on Github. For the review phase, we include an anonymized version of the code and notebook as supplementary material. Instructions to reproduce synthetic data (and to access public datasets, where used) are included in a Readme file.

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

## A ADDITIONAL RELATED WORK

In this section, we review additional related work in theoretical single-task and multi-task learning and compare it to our contributions.

### A.1 LEARNING DYNAMICS OF DEEP LINEAR NETWORKS

Our work extends several foundational theoretical analyses of the continuous-time limit of gradient descent, known as *gradient flow*, to the setting where multiple tasks are learned simultaneously, i.e., multi-task learning (MTL). We build on prior theoretical studies of deep linear networks, including those by Saxe et al. (2014), Fukumizu (1998), and Braun et al. (2022). As a key starting point, Saxe et al. (2014) derived nonlinear, coupled differential equations that describe the learning dynamics of network weights in terms of input-output statistics. They provided exact analytical solutions under *tabula rasa* (i.e., small random weights) initializations and whitened input data. Later work Saxe et al. (2019) extended these results to both shallow and deep networks, offering insights into the semantic development of representations in terms of effective singular values and the temporal evolution of hidden layers. However, whereas their analysis focuses on how different classes are learned over time within a shared multi-output setting, our framework addresses a distinct scenario: multiple tasks are learned in parallel, each with its task-specific gradients and dedicated layers. While the analytical expressions in Saxe et al. (2014; 2019) rely on the assumptions of linear networks, specific initialization schemes, and whitened inputs, Pennington et al. (2017) demonstrated how theoretical insights from linear networks can inform improved initialization strategies for nonlinear models. Similarly, Gidel et al. (2019) extended aspects of the gradient flow analysis to discrete-time gradient descent, narrowing the gap between theory and practice. As such, these theoretical insights are still believed to offer important insights for more general architectures and initializations.

Building on the foundational analyses from Saxe et al. (2014), subsequent work Lampinen & Ganguli (2018); Advani et al. (2020); Goldt et al. (2019) developed an analytical theory for generalization in deep linear neural networks in a teacher-student setup Seung et al. (1992); Engel (2001). This setup has also been applied to analyse continual learning Lee et al. (2021) (including the problem of *catastrophical forgetting* Goodfellow et al. (2013) ) and transfer learning Lampinen & Ganguli (2018). Additionally, Saxe et al. (2022) introduced the Gated Deep Linear Network (GDLN) framework to allow for non-linear functions of inputs to learn more about the implications of architectural choices on the learning dynamics for spectral initializations. While they allow for output-specific nodes, they only consider models updated with one overall gradient and no task-specific weights. Finally, Zhang et al. (2024) analyses the dynamics of multi-modal learning, where different modalities (different inputs) are used to learn a common task (same output). They obtain analytical solutions for uncorrelated modalities and analyse unimodal bias for early and late fusion networks through their loss landscape. However, their setup should not be confused with our work, which considers multiple outputs from a similar input.

Our work is mostly related to the recent line of work Tarmoun et al. (2021); Braun et al. (2022); Dominé et al. (2025) that builds from the matrix Riccati formulation of gradient descent Fukumizu (1998) and matrix factorization. In Tarmoun et al. (2021); Braun et al. (2022), gradient flow equations are written for single-task learning as matrix Riccati equations which have finite solutions under some assumptions Sasagawa (1982) that hold for a richer class of initial conditions than the *Tabula Rasa initialization* as encountered in the analytical solutions obtained in Saxe et al. (2014) and subsequent work. In particular Braun et al. (2022) uses this formalism and describes alignment dynamics and convergence behaviour that do not occur in the tabula rasa setting for zero-balanced weight initialization, corresponding to rich dynamics, and Dominé et al. (2025) extends the obtained solution to more general non-zero-balanced initializations. Both Braun et al. (2022); Dominé et al. (2025) apply their formalism to transfer and/or continual learning, where different tasks are learned sequentially, but do not mention multi-task learning, where tasks are learned simultaneously.

### A.2 THEORETICAL WORK ON MULTI-TASK LEARNING

Early theoretical work in multi-task learning Ben-David & Schuller (2003); Xue et al. (2007) focused on Kernel Learning and states that multi-task works well on similar tasks, where the notion of similarity is quantified based on the similarity of the corresponding single-task models (e.g. decision

boundaries are close). In Wu et al. (2020), this notion of similarity is extended to neural networks where task data similarity is considered as a second-order factor that impacts interference between multiple tasks in hard-parameter sharing MTL networks. Based on that observation, they derive bounds for the maximal capacity of the shared layer and the number of datapoints of the source task needed to enforce interference between tasks. Accordingly, they provide a static, geometry-based characterization of task similarity and transfer conditions in MTL neural networks, without accounting for the temporal dynamics or describing how these interactions evolve. The main findings of Wu et al. (2020) are: a too high capacity of the shared module eliminates any kind of information transfer between tasks, changing the number of data points, and a notion of STL covariance similarity impacts the amount of transfer between tasks. In contrast to their work, we show that, next to the training losses, transfer can be studied by considering the representations and output functions. Furthermore, this work considers how the evolution of the differences in alignment affects the learning dynamics (i.e., the learning speed and internal representations) while Wu et al. (2020) defines transfer as a static notion related to performance on the validation set.

Particularly relevant to our work is the analysis presented in Lampinen & Ganguli (2018), which introduced the concept of task alignment through a similarity matrix constructed from the right singular vectors associated with different tasks. Their study is situated within a student-teacher framework and focuses on the derivation of generalization dynamics in single-task networks. By deriving analytical expressions for the generalization error at early stopping, they define a measure of transfer benefit as the difference in generalization error when tasks are trained jointly versus separately. In the specific setting of two low-rank teacher models, they demonstrate that transfer in multi-task learning is primarily determined by the initial alignment between tasks and the magnitudes of their respective singular values, while being independent of the output singular vectors. Building on this work, we also utilize the similarity matrix to define various classes of alignment. However, rather than focusing on generalization error, our analysis is concerned with how task alignment shapes the learning dynamics, in a way that is independent of the student-teacher paradigm and accommodates more general initializations and model architectures.

While Lampinen & Ganguli (2018) and Wu et al. (2020) primarily quantify transfer in terms of its effect on generalization, our work goes further by characterizing both how transfer evolves throughout training and what forms it can take. Specifically, we examine not only how tasks influence each other's final performance, but also how they affect the trajectory of learning, the speed of convergence, and the structure of intermediate representations. In contrast to prior analyses, we aim to provide a more detailed account of the dynamics of inter-task interaction, offering a richer understanding of transfer beyond end-point generalization metrics.

### A.3 PRACTICAL ALGORITHMS FOR TASK GROUPING AND DYNAMIC TASK LOSS WEIGHTING

Recent work in MTL is often focused on the design of dynamic weighting algorithms with the goal of eliminating negative task interference/transfer He et al. (2024b); Grégoire & Verboven (2023). At the base of the design of these different algorithms are some common paradigms concerning the source of negative interference, i.e., *gradient imbalance* Chen et al. (2018); Leang et al. (2020) which occurs when the gradients of different tasks have large differences in magnitudes and *conflicting gradient directions* Du et al. (2020); Liu et al. (2021; 2023) which occurs when task-specific gradients have negative cosine similarities. Recently, these paradigms have been questioned and critically analyzed Elich et al. (2023). To address the mentioned problems, we aim to understand the learning dynamics and propose a theoretical framework to effectively study task interactions from a new perspective.

**Critical Studies on Dynamic Weighting Algorithms.** While it is clear that in many cases, dynamic weighting algorithms outperform Equal Weighting algorithms where tasks are given the same, static, task specific weight, Lin et al. (2022) showed that random dynamical weighting yields similar performance as computationally expensive SOTA dynamic weighting algorithms which are designed on task-conflicting heuristics. Similarly, Xin et al. (2022) performed an extensive empirical search and found that most of the performance differences between MTL models is much more sensitive to the choice of hyperparameters than the choice of the dynamical weighting algorithms, highlighting the point of Elich et al. (2023) that a better theoretical understanding is necessary to complement the popular gradient based task-weighting algorithms and rethink the common paradigms in MTL literature. According to Grégoire et al. (2024), a necessary step is to disentangle interference that is

not related to differences between tasks as a whole with interference coming from different samples within the same task, since gradient interference is not necessarily unique to MTL setups Elich et al. (2023).

**Task Grouping.** Another line of research in MTL literature is concerned with the fundamental question of which tasks should be learned together in a multi-task model to maximize cooperation and minimize competition. Since the search space of possible task groupings grows exponentially with the number of tasks, exhaustive evaluation of different groups is infeasible even for moderate task counts. Recent works have approached this challenge from several angles, e.g., regularization-based methods that learn task relationships as part of the training objective Zhang & Yeung (2014) heuristic or empirical methods that evaluate candidate groupings (often via training on subsets or proxy metrics) Standley et al. (2020); Sherif et al. (2024), and meta-learning approaches that predict optimal groupings or task affinities Song et al. (2022); Fifty et al. (2021).

A.4 PRACTICAL IMPLICATIONS OF THIS WORK

There are two main practical implications from our results. Concerning task grouping, we show that there must be some signal in the data that could guide task grouping algorithms: since the degree of alignment between tasks implies how tasks interact, we propose to take into account the degree of alignment when grouping tasks. As such, our results open a new perspective, one that moves away from computationally expensive techniques that require repeatedly training full MTL models. Instead of proposing yet another heuristic method for task grouping or dynamic weighting, our work provides theoretical insights that highlight the role of the data structure itself in shaping multitask interactions.

With respect to dynamic weighting algorithms specifically, our findings show that task alignment directly influences the learning dynamics: aligned tasks tend to learn more simultaneously and- as we now illustrate with additional plots-produce far fewer conflicting gradients than conflicting or orthogonal tasks. This indicates that the choice or design of a dynamic weighting algorithm should be task- and alignment-dependent, rather than one-size-fits-all. For example, our insights suggest that complex dynamic weighting algorithms are likely to offer larger benefits—relative to simple uniform weighting—for conflicting and orthogonal task pairs than for aligned ones. While determining the exact form that such weighting strategies should take is beyond the scope of this paper, we believe that our theoretical framework provides a principled basis that can inform and inspire future work on more effective and efficient MTL algorithms.

# B MULTI-TASK LEARNING: PRELIMINARIES

This work considers a Multi-Task Learning (MTL) Ruder (2017) setup in which the objective is to simultaneously learn $N_T$ related tasks $\{\mathcal{T}_1, \ldots, \mathcal{T}_{N_T}\}$. All tasks share a common input vector $\mathbf{x} \in \mathbb{R}^{N_1}$, where $N_1$ is the input dimensionality, but each task $\mathcal{T}_i$ is associated with task-specific output vectors $\mathbf{y}_i \in \mathbb{R}^{N_{io(l)}}$, where $N_{io}$ denotes the output dimensionality of task $\mathcal{T}_i$. To this end, a standard hard-parameter sharing architecture is adopted Zhang & Yang (2021). In this setup, the tasks are trained jointly using shared weight matrices $\mathbf{W}_S = \prod_{\ell=1}^{L_S} \mathbf{W}_S^{(\ell)}$ followed by $\mathcal{T}_i$-specific weight matrices $\mathbf{W}_i = \prod_{\ell=1}^{L_i} \mathbf{W}_i^{(\ell)}$. The difference with a single-task two-layer linear network is visualised in Figure 1.

The MTL network is trained by minimizing a composite loss function $\mathcal{L}_{\text{Tot}}$, defined as a weighted sum of task-specific losses $\mathcal{L}_i$,

$$\mathcal{L}_{\text{Tot}}(t) = \sum_{i=1}^{N_T} \omega_i(t)\mathcal{L}_i(\mathbf{W}_i, \mathbf{W}_S, t), \qquad (26)$$

where $\omega_i(t)$ denotes the time-dependent loss weighting coefficient for task $\mathcal{T}_i$, controlling its relative contribution to the overall loss at training iteration $t$. Here, $t$ corresponds to the number of epochs. We assume uniform static weighting $\omega_i = 1$ but discuss the impact of $\omega_i \neq 1$ in Appendix D.

In this work, the neural network is trained with full-batch gradient descent with a mean squared error (MSE) task-specific loss function $\mathcal{L}_i^{\text{MSE}}$ for each task,

$$\mathcal{L}_i^{\text{MSE}} = \frac{1}{N} \sum_{\mu=1}^{N_D} ||\mathbf{y}_i^\mu - \mathbf{W}_i \mathbf{W}_S \mathbf{x}^\mu||^2, \tag{27}$$

where $N_D$ is the number of input samples and $\mathbf{x}^\mu, \mathbf{y}_i^\mu$ denote the $\mu$-th input and corresponding target output for task $\mathcal{T}_i$. While the shared weights $\mathbf{W}_S$ are updated based on the gradient computed on the total loss $\mathcal{L}_{\text{Tot}}(t)$ (equation 26), the task-specific weights are updated through task-specific gradients computed on the task-specific losses $\mathcal{L}_i$ and learning rate $\alpha$ according to the update rule,

such that $\forall i \neq j : \nabla_{\mathbf{W}_i} \mathcal{L}_j = 0$.

### B.1 Two-Layer Linear Two-Task Networks

In line with prior studies on the learning dynamics in single-task learning, we consider the setup of a two-layer linear network. Specifically, we use MTL networks with hard-parameter sharing Ruder (2017) where the shared weights $\mathbf{W}_S$ are updated through the gradient computed on the composite loss of both tasks, $\mathcal{L}_{MTL}$,

$$\begin{aligned}
\mathbf{W}_S^{(t+1)} &= \mathbf{W}_S^{(t)} - \alpha \nabla_{\mathbf{W}_S^{(t)}} \mathcal{L}_{MSE} \\
&= \mathbf{W}_S^{(t)} - \frac{1}{N_D} \alpha \nabla_{\mathbf{W}_S^{(t)}} \sum_i^{N_T} \sum_{\mu=1}^{N_D} \|\mathbf{y}_i^\mu - \mathbf{W}_i \mathbf{W}_S \mathbf{x}^\mu\|^2,
\end{aligned} \tag{28}$$

The gradient with respect the shared layer $\mathbf{W}_S$, is given by

$$\begin{aligned}
\nabla_{\mathbf{W}_S^{(t)}} \mathcal{L}_{MSE} &= -\sum_{i=1}^{N_T} \sum_{\mu=1}^{N_D} \mathbf{W}_i^T \left( \mathbf{y}_i^\mu - \mathbf{W}_i \mathbf{W}_S \mathbf{x}^\mu \right) \mathbf{x}^{\mu T} \\
&= -\sum_{i=1}^{N_T} \sum_{\mu=1}^{N_D} \left( \mathbf{W}_i \right)^\top \left( \mathbf{y}_i^\mu \mathbf{x}^{\mu T} - \mathbf{W}_i \mathbf{W}_S \mathbf{x}^\mu \mathbf{x}^{\mu T} \right)
\end{aligned} \tag{29}$$

In the limit of small learning rates $\alpha \to 0$, the updates of the weights are sufficiently small such that $\Delta W^t \to dW^t$ and the gradient descent dynamics are well approximated by the continuous time differential equation Saxe et al. (2014),

$$\tau \frac{d\mathbf{W}_S}{dt} = \sum_{i=1}^{N_T} \mathbf{W}_i^\top (\mathbf{\Sigma}_{io} - \mathbf{W}_i \mathbf{W}_S \mathbf{\Sigma}_{11}), \tag{30}$$

which we refer to as *Gradient Flow*. Where we defined, the input-input and input-output correlation matrices as,

$$\begin{aligned}
\mathbf{\Sigma}_{11} &\equiv \sum_{\mu=1}^{N_D} \mathbf{x}^\mu (\mathbf{x}^\mu)^T \\
\mathbf{\Sigma}_{io} &\equiv \sum_{\mu=1}^{N_D} \mathbf{y}_i^\mu (\mathbf{x}^\mu)^T.
\end{aligned} \tag{31}$$

In contrast, the task-specific gradient signals are explicitly separated for the task-specific weights $\mathbf{W}_i$, which get updated through,

$$\mathbf{W}_i^{(t+1)} = = \mathbf{W}_i^{(t)} - \frac{1}{N_D} \alpha \nabla_{\mathbf{W}^{(t)}} \sum_{\mu=1}^{N_D} \|\mathbf{y}_i^\mu - \mathbf{W}_i \mathbf{W}_S \mathbf{x}^\mu\|^2 \tag{32}$$

yielding,

$$\tau \frac{d\mathbf{W}_i}{dt} = \left(\mathbf{\Sigma}_{io} - \mathbf{W}_i \mathbf{W}_S \mathbf{\Sigma}_{11}\right) \mathbf{W}_S^\top, \tag{33}$$

in the continuous limit. Assuming whitened wheights ($\mathbf{\Sigma}_{11} = \mathbf{1}$) the gradient flow equations for learning two tasks simultaneously are thus given by,

$$\tau \frac{d\mathbf{W}_S}{dt} = \mathbf{W}_1^\top \left(\mathbf{\Sigma}_{1o} - \mathbf{W}_1 \mathbf{W}_S\right) + \mathbf{W}_2^\top \left(\mathbf{\Sigma}_{2o} - \mathbf{W}_2 \mathbf{W}_S\right) \tag{34}$$

$$\tau \frac{d\mathbf{W}_1}{dt} = \left(\mathbf{\Sigma}_{1o} - \mathbf{W}_1 \mathbf{W}_S\right) \mathbf{W}_S^\top \quad \frac{d\mathbf{W}_2}{dt} = \left(\mathbf{\Sigma}_{2o} - \mathbf{W}_2 \mathbf{W}_S\right) \mathbf{W}_S^\top \tag{35}$$

For $N_T > 2$, the derivation of the equations is completely analogous. Note that we can write these equations differently by stacking the task labels and task-specific weights into larger matrices,

$$Y_{MTL} = \begin{pmatrix} y_1 & y_2 \end{pmatrix}^T \quad \mathbf{W}_{MTL} = \begin{pmatrix} \mathbf{W}_1 & \mathbf{W}_2 \end{pmatrix}^T \quad \xi = \begin{pmatrix} \mathbf{\Sigma}_{1o} & \mathbf{\Sigma}_{2o} \end{pmatrix}^T \tag{36}$$

With that notation, we can write the MTL gradient flow equations as,

$$\tau \frac{d\mathbf{W}_S}{dt} = \mathbf{W}_{MTL}^T (\xi - \mathbf{W}_{MTL} \mathbf{W}_S) \tag{37}$$

$$\tau \frac{d\mathbf{W}_i}{dt} = (\xi_{[N_{(i-1)o}:N_i, N_{(i-1)}o:N_i]} - \mathbf{W}_{MTL[N_{(i-1)o}:N_i, N_{(i-1)}o:N_i]} \mathbf{W}_S) \mathbf{W}_S^T. \tag{38}$$

These are just like the single-task equations solved for a family of initializations in Saxe et al. (2014). In the main paper, we discuss how the solutions to these equations serve as a way to describe inter-task dynamics. In Appendix B.3 we discuss solutions from *Tabula Rasa* initializations and the special case of similar statistical tasks. In Appendix C we discuss how to rewrite these equations into the Riccati formalism which enables us to find an expression for the output functions and internal representations for a broader family of initializations as compared to the Tabula Rasa case.

## B.2 Differences and Application to Single-Task Learning

In contrast to single-task learning (STL), our framework accommodates heterogeneous tasks (e.g., semantic segmentation and depth estimation Liu et al. (2019)) with task-specific losses and gradient updates that separate the task-signals in parts of the network. Furthermore, each task's influence can be controlled through task-specific loss weights (equation 26), while supporting both shared and task-specific layers. Certain single-task problems, such as multi-class classification, can easily be reframed as multi-task problems by defining subgroups of tasks and task-specific layers. As such, our analysis can also shed light on how class-specific modes interact during training, and compare it to learning each class/subgroup of tasks in a separate model. Moreover, this architecture allows for different weighting between these groups or a different kind of activation/loss function, which will typically occur when combining regression and classification tasks (e.g., simultaneous depth and segmentation tasks Liu et al. (2021)). As such, we can see this work as a generalization of earlier solutions proposed in Braun et al. (2022) but with new insights concerning the alignment of subgroups of tasks.

## B.3 Solving the equations in the Total MTL space: Tabula Rasa

In this section, we extend the methodology proposed in Saxe et al. (2014; 2019) to implement the notion of multiple tasks into the analytical solutions to the learning dynamics of gradient flow. Decomposing $\xi = \mathbf{U}_\xi \mathbf{S}_\xi \mathbf{V}_\xi^T$ and performing the change of variables $\mathbf{W}_S = \bar{\mathbf{W}}_S \mathbf{V}_\xi^T$ and $\mathbf{W}_{MTL} = \mathbf{U}_\xi \bar{\mathbf{W}}_{MTL}$ yields the following equations,

$$\tau \frac{d\bar{\mathbf{W}}_S}{dt} = \bar{\mathbf{W}}_{MTL}^T (\mathbf{S}_\xi - \bar{\mathbf{W}}_{MTL} \bar{\mathbf{W}}_S) \tag{39}$$

$$\tau \frac{d\bar{\mathbf{W}}_{MTL}}{dt} = (\mathbf{S}_\xi - \bar{\mathbf{W}}_{MTL} \bar{\mathbf{W}}_S) \bar{\mathbf{W}}_S^T. \tag{40}$$

Hence, these formally look like the STL equations which are solved in Saxe et al. (2014) for a special class of initializations (often referred to as spectral or Tabula Rasa (random and small) initializations). Unlike STL analyses, which treat each task's gradient flow in isolation, our approach

solves a single coupled dynamical system where one shared hidden matrix is pulled simultaneously by the residuals of all tasks, letting us capture transfer and interference effects that STL cannot reveal. By diagonalizing the joint teacher and deriving a closed-form solution to the dynamics, we obtain an expression of waht every task shares and the task-specific read-outs which represent how much each tasks restracts from a certain mode. To recover the the latter, we construct,

$$\mathbf{U}_\xi = (\mathbf{U}_{\xi 1} \quad \mathbf{U}_{\xi 2})^T, \tag{41}$$

where $\mathbf{U}_{\xi 1}$ and $\mathbf{U}_{\xi 2}$ are obtained by slicing the correct number of rows for task 1 and task 2. As such, the task-specific output maps are given by,

$$\mathbf{W}_1 \mathbf{W}_S = \mathbf{U}_1 \mathbf{H}(t) \mathbf{V}_\xi \quad \mathbf{W}_2 \mathbf{W}_S = \mathbf{U}_2 \mathbf{H}(t) \mathbf{V}_\xi, \tag{42}$$

where $\mathbf{H}(\mathbf{t})$ is the time-dependent diagonal effective singular value matrix, with diagonal values $h^\alpha$ corresponding to a mode in the shared space $\xi$, which are found by solving equations (40),

$$h^\alpha(t) = \frac{e^{\frac{2s_\xi t}{\tau}} s_\xi}{\frac{s_\xi}{h_0^2} + e^{\frac{2s_\xi t}{\tau}} - 1}, \tag{43}$$

as shown in Saxe et al. (2014). Since the rank and the values of the singular value matrix $\mathbf{S}_\xi$ are determined by the alignment of the two tasks, the MTL learning dynamics are as well. While both tasks feel the same amplitude of the effective singular value $h^\alpha$, they react to it in a different way, which is determined by the left singular values $\mathbf{U}_{\xi,i}$. The way these task-specific layers react to the same singular mode $\alpha$ is quantified through gamma, $\gamma_\alpha^{(j)}$,

$$\gamma_\alpha^{(j)} = \|u_\alpha^{(j)}\| = \sqrt{\sum_{k=1}^{N_{j,o}} \left[U_\xi\right]_{k,\alpha}^2}, \quad j = 1, 2, \tag{44}$$

such that for for each mode $\alpha$ the task-specific "effective" singular values $a_\alpha(t)$, $b_\alpha(t)$ evolve according to,

$$a_\alpha(t) = \gamma_\alpha^{(1)} h_\alpha(t), \qquad b_\alpha(t) = \gamma_\alpha^{(2)} h_\alpha(t). \tag{45}$$

In equation (45), the scalars $\gamma_i^j$ are constant prefactors that measure how strongly each task j "loads" onto mode-i, e.g., a mode with $\gamma_i^1 >> \gamma_i^2$ is mostly used by task one while a mode with $\gamma_i^1 \approx \gamma_i^2$ is shared by both tasks equally. Since both tasks feel the same amplitude of the effective singular value $h^\alpha$, they read out the same modes at the same rate, determined by the task that perceives the highest strength for this mode.

### B.4 SOLVING THE EQUATIONS FOR TABULA RASA INITIALIZATIONS

For completeness, we show how to obtain the solution in equation (43). The equations in (40) are typically solved for the columns of the synaptic matrices $\bar{\mathbf{W}}_S$ and $\bar{\mathbf{W}}_{MTL}$. Let $h^\alpha$ be the $\alpha$-th column of the shared synaptic weight matrix $\bar{\mathbf{W}}_S$, representing a column vector of $N_S$ synaptic weights connecting input mode $\alpha$ to the hidden layer, $m^\alpha$ a column vector of $N_s$ synaptic weights connecting shared neurons to the $\xi$-output mode $\alpha$. The equations (40) in terms of these columns vectors are given by,

$$\tau \frac{d}{dt} h^\alpha = [(s_\xi^\alpha - h^\alpha.m^{\alpha,\top})m^{\alpha,\top} - \sum_{\alpha \neq \gamma} m^{\gamma,\top}(h^\alpha.m^{\gamma,\top})] \tag{46}$$

$$\tau \frac{d}{dt} m^\alpha = (s_\xi^\alpha - h^\alpha.m^{\alpha,\top})m^\alpha - \sum_{\alpha \neq \gamma} h^\gamma(m^\alpha.m^\gamma). \tag{47}$$

To further simplify the vector equations (47), we assume a special class of initializations (i.e., Tabula Rasa) in which the equations the cross-mode contributions in (47) disappear. Namely, $h^\alpha, a^\alpha, b^\alpha \sim r^\alpha \forall \alpha \in \{1, ..., N_2\}$ such that $r^\alpha.r^\beta = \delta^{\alpha\beta}$ and all other connectivity modes set to zero. Under these assumptions, the modes evolve independently from each other. Projecting everything along the $r^\alpha$ direction ($h^\alpha.r^\alpha = h, m^\alpha.r^\alpha = a, s_\xi^\alpha.r^\alpha = s_\xi$) turns the equations in (47) to the coupled scalar equations for each mode $\alpha$,

$$\tau \frac{dh}{dt} = m(s_\xi - mh) \tag{48}$$

$$\tau \frac{da}{dt} = m(s_\xi - mh). \tag{49}$$

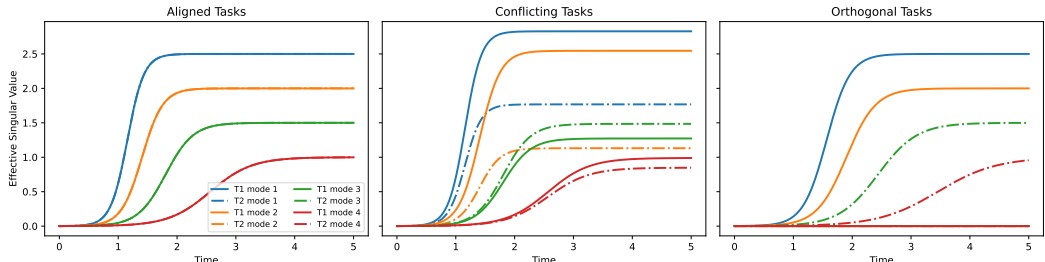

Figure 7: Simulation of the learning dynamics for aligned, conflicting, and orthogonal tasks based on the analytical solution defined by equation (43).

Since $h \sim m$ for our initializations, one can solve the equation (49) in terms of the variable $u = hm$, which allows us to capture the dynamics with one equation,

$$\tau \frac{du}{dt} = 2u(s_\xi - u), \tag{50}$$

which is separable and yields the following expression in terms of t, which can be reverted to obtain (43). In subsection B.5, we solve the equations (35) in another task basis for similar tasks. Note that zero-balanced initializations, as assumed in the main paper, are more general than the Tabula Rasa initialization assumed here. In Appendix C we discuss the derivation of the solutions under this more general assumption.

### B.5 SOLVING THE EQUATIONS FOR SIMILAR STATISTICAL STRUCTURE TASKS: TABULA RASA

To get a better understanding of the interpretation of the solution (43), we consider the case of tasks with a similar statistical structure, with similar and imbalanced singular values. To this end, we consider the singular value decomposition (SVD) of the first task's input-output correlation matrix,

$$\mathbf{\Sigma}_{1o} = \mathbf{U}_1 \mathbf{S}_1 \mathbf{V}_1^T, \tag{51}$$

where $\mathbf{V}_1$ is a $N_1 \times N_1$ orthogonal matrix whose columns contain input-analyzing singular vectors $v_\alpha$ that reflect independent modes of variation in the input, $\mathbf{U}_1$ is an $N_{31} \times N_{31}$ orthogonal matrix whose coluluns contain output-analyzing vectors $u_\alpha$ that reflect independent modes of variation in the output and $\mathbf{S}_1$ is an $N_{31} \times N_1$ diagonal matrix containing the singular values $s_\alpha^1$, ordered such that $s_1^1 > s_2^1 ... > s_{N_1}$.

We transform the variables in equation 35 to the task 1 synaptic weight space according to $\mathbf{W}_S = \bar{\mathbf{W}}_S V_A^\top$, $\mathbf{W}_1 = \mathbf{U}_1 \bar{\mathbf{W}}_1$, $\mathbf{W}_2 = \mathbf{U}_1 \bar{\mathbf{W}}_2$ (where we assume both tasks have the same dimension for a moment so we can project $\mathbf{W}_2$ along the $\mathbf{U}_1$ basis). By studying the evolution of $\bar{\mathbf{W}}_2$ we learn how $\mathbf{W}_2$ evolves along the singular directions of task's 1 data structure. Note that we can find the original weights back once we found solutions for $\bar{\mathbf{W}}_i$ and $\bar{\mathbf{W}}_S$.

This transforms the equations (5) to,

$$\tau \dot{\bar{\mathbf{W}}}_S = \bar{\mathbf{W}}_1^T (\mathbf{S}_1 - \bar{\mathbf{W}}_1 \bar{\mathbf{W}}_S) + \bar{\mathbf{W}}_2^T (\mathbf{U}_1^T \mathbf{\Sigma}_{2o} \mathbf{V}_1 - \bar{\mathbf{W}}_2 \bar{\mathbf{W}}_S) \tag{52}$$

$$= \bar{\mathbf{W}}_1^T (\mathbf{S}_1 - \bar{\mathbf{W}}_1 \bar{\mathbf{W}}_S) + \bar{\mathbf{W}}_2^T (\mathbf{S}_2^{proj} - \bar{\mathbf{W}}_2 \bar{\mathbf{W}}_S) \tag{53}$$

$$\tau \dot{\bar{\mathbf{W}}}_1 = (\mathbf{S}_1 - \bar{\mathbf{W}}_1 \bar{\mathbf{W}}_S) \bar{\mathbf{W}}_S^T \tag{54}$$

$$\tau \dot{\bar{\mathbf{W}}}_2 = (\mathbf{S}_2^{proj} - \bar{\mathbf{W}}_2 \bar{\mathbf{W}}_S) \bar{\mathbf{W}}_S^T, \tag{55}$$

where we defined $\mathbf{S}_2^{proj} \equiv \mathbf{U}_1^T \mathbf{\Sigma}_{2o} \mathbf{V}_1$, which is not necessary diagonal (in contrast to $\mathbf{S}_1$). $\mathbf{S}_2^{proj}$ quantifies how much the second task aligns with the latent structure extracted from the first task. If $\mathbf{S}_2^{proj}$ is nearly diagonal, it suggests that both tasks rely on similar input-output relationships. Conversely, if there is significant off-diagonal structure, it indicates that task 2 involves a different or more mixed combination of the underlying features captured by task 1. By studying the evolutions

of the weights in the basis of the singular directions of task 1 we learn how this initial alignment or dis-alignment between the tasks evolves throughout training updates.

Assuming similar tasks, we obtain a nearly diagonal $\mathbf{S}_{proj}$. Similar to the total-space case, we solve the resulting equations for the columns of these synaptic matrices. Let $h^\alpha$ be the $\alpha$-th column of the shared synaptic weight matrix, representing a column vector of $N_2$ synaptic weights connecting input mode $\alpha$ to the hidden layer, $a^\alpha$ a column vector of $N_2$ synaptic weights connecting shared neurons to the task-1 output mode $\alpha$ and $b^\alpha$ as a vector containing weights which connect hidden layers to the task-2 output mode $\alpha$ as projected in the task-1 weight space. The equations (55) in terms of these columns vectors are given by,

$$\tau \frac{d}{dt} h^\alpha = [(s_1^\alpha - h^\alpha . a^{\alpha,\top}) a^{\alpha,\top} - \sum_{\alpha \neq \gamma} a^{\gamma,\top} (h^\alpha . a^{\gamma,\top})] \tag{56}$$

$$+ [(s_{2,proj}^\alpha - h^\alpha . b^{\alpha,\top}) b^{\alpha,\top} - \sum_{\alpha \neq \gamma} b^{\gamma,\top} (h^\alpha . b^{\gamma,\top})] \tag{57}$$

$$\tau \frac{d}{dt} a^\alpha = (s_1^\alpha - h^\alpha . a^{\alpha,\top}) h^\alpha - \sum_{\alpha \neq \gamma} h^\gamma (a^\alpha . h^\gamma) \tag{58}$$

$$\tau \frac{d}{dt} b^\alpha = (s_2^{\alpha,proj} - h^\alpha . b^{\alpha,\top}) b^\alpha - \sum_{\alpha \neq \gamma} h^\gamma (b^\alpha . h^\gamma). \tag{59}$$

Similar to the case in single-task learning, the summations $\sum_{\alpha \neq \gamma}$ represent inter-mode interactions between different layers. The other terms represent the cooperation of the different connectivity modes $h^\alpha, a^\alpha, b^\alpha$ to drive each other to larger magnitudes and similar directions to reflect the strength of the singular values $s_1^\alpha$ and $s_{2,proj}^\alpha$ in the shared weight space. We discuss how the ratio of the singular values $s_1^\alpha$ and $s_{2,proj}^\alpha$ defines the learning dynamics of the connectivity modes and hence how the different tasks are influenced by differences in magnitudes. Projecting everything along the $r^\alpha$ direction ($h^\alpha . r^\alpha = h, a^\alpha . r^\alpha = a, b^\alpha . r^\alpha = b, s_1^\alpha . r^\alpha = s_1$ and $s_{2,proj}^\alpha . r^\alpha = s_{2,proj}$) turns the equations in (59) to the coupled scalar equations,

$$\tau \frac{dh}{dt} = [a(s_1 - ah) + b(s_{2,proj} - bh)] \tag{60}$$

$$\tau \frac{da}{dt} = h(s_1 - ah) \tag{61}$$

$$\tau \frac{db}{dt} = h(s_1^{proj} - bh). \tag{62}$$

For similar tasks, the $\gamma$ factor occurring in (45) corresponds to a scalar factor. As such we can write $h = Kb(t)$ and $a = Lb(t)$, where $L$ and $K$ are scalar scaling factors which depend on the ratio $s_{proj}/s_1$. Inserting $h = Kb(t)$ and $a = Lb(t)$ into (60) yields,

$$\frac{db}{b\left(s_1(L + \frac{s_{proj}}{s_1}) - Kb^2(L^2 + 1)\right)} = \frac{1}{\tau K} dt; \tag{63}$$

or, setting, $K' = K(L^2 + 1), s' = s_1(L + \frac{s_{proj}}{s_1})$,

$$\frac{db}{b(s' - K'b^2)} = \frac{1}{\tau K} dt. \tag{64}$$

Subsituting $u = b^2$ and using partial fractions: $\frac{1}{u(s-uK)} = \frac{1}{su} + \frac{K'}{s(s-uK)}$ yields,

$$\frac{1}{2}\left(\int_{u_0}^{u_f} \frac{du}{s'u} + \int_{u_0}^{u_f} \frac{K'du}{s'(s' - K'u)}\right) = \frac{1}{\tau K'} \int_0^t dt, \tag{65}$$

or after substituting $p = s' - uK'$,

$$\frac{1}{2}\left(\frac{1}{s_{proj}} \ln|u||_{u_0}^{u_f} - \int_{p_0}^{p_f} \frac{dp}{s_{proj}p}\right) = \frac{t}{\tau K} \tag{66}$$

and finally,

$$\frac{K'\tau}{2} \frac{1}{s_{proj}} \left( \ln |\frac{b_f^2}{b_0^2}| + \ln \frac{|(s - K'b_0^2)|}{|(s - K'b_f^2)|} \right) = t. \tag{67}$$

$$\frac{K'\tau}{2s'} \ln \frac{|b_f^2(s' - K'b_0^2)|}{|b_0^2((s' - K'b_f^2)|} = t \tag{68}$$

Inverting this to an expression in terms of b and going back to K and $s_{proj}$ one obtains,

$$b(t) = \sqrt{\frac{e^{\frac{2(s_1 L + s_{proj})t}{\tau K}}(s_1 L + s_{proj})}{\frac{(s_1 L + s_{proj})}{b_0^2} + e^{\frac{2(s_1 L + s_{proj})t}{\tau K}} K(L^2 + 1) - K(L^2 + 1)}} \tag{69}$$

$$h(t) = Kb(t) \qquad a(t) = Lb(t), \tag{70}$$

where, $K$ and $L$ are scaling constants,

$$K = \frac{a_K}{\frac{s_{proj}}{s_1} + b_K} \qquad L = \frac{a_L}{\frac{s_{proj}}{s_1} + b_L} \approx \frac{s_1}{s_{proj}}, \tag{71}$$

where the last approximation is a result of noticing that $a_L = 1.006 \approx 1$ and $b_L = 0.006 \approx 0$.

**Imbalance in magnitude:** $s_1/s_{2,proj} << 1$. When the singular values associated to the same mode are different, in such a way that it is much stronger for task 2 then task 1, the solution in equation (70) can be written as,

$$b_{s_{proj} > s_1}(t) = \sqrt{\frac{e^{\frac{2s_{2,proj}t}{\tau K}} s_{2,proj}}{\frac{s_{2,proj}}{b_0^2} + e^{\frac{2s_{2,proj}t}{\tau K}} K - K}} \tag{72}$$

$$h_{s_{proj} > s_1}(t) = Kb_{s_{proj} > s_1}(t) \qquad a_{s_{proj} > s_1}(t) = Lb_{s_{proj} > s_1}(t) \tag{73}$$

Hence, in this case, the magnitude of the shared connectivity mode of the multi-task system are determined fully by the second task and $s_{2,proj}$. Furthermore, for $K \to 1$, this correponds to the solution derived in Saxe et al. (2014; 2019) for single task learning with $h \approx b$. In this regime, the hidden layer and task-specific layer corresponding to the "leading task" act as if they were in a single-task network. The smaller the relative difference between $s_{2,proj}$ and $s_1$, the bigger $K$ and hence the difference with the single-task behaviour. Interestingly, $a(t)$ converges at the same speed as $h$ and $b$, which can be explained by the rise of $h$ that influences $a$. As can be seen from the $s_1/s_{2,proj}$ factor appearing in $a(t)$, $a$ stays relatively small for big values of $s_{2,proj}$ but the change rate is similar or equal to that of $h$. Of course, a similar solution holds for the case where $s_1/s_{2,proj} > 1$ by interchanging the roles of $a$ and $b$.

**Similar magnitude singular values** $s_1 \approx s_{2,proj}$. When a mode is equally strong for both tasks, i.e. $s_1 = s_{2,proj}$ the solution in (70) can be written as,

$$b_\sim(t) = \sqrt{\frac{e^{\frac{4s_{2,proj}t}{\tau K}} s_{2,proj}}{\frac{s_{2,proj}}{b_0^2} + e^{\frac{4s_{2,proj}t}{\tau K}} K - K}} \tag{74}$$

$$h_\sim(t) = Kb(t) \qquad a_\sim(t) = b(t), \tag{75}$$

In this case, the perceived strength is doubled for both tasks and $L = 1$ so both modes evolve in the same way.

### B.6 LEARNING SPEED COMPARED TO SINGLE TASK LEARNING

By inverting the solution in equation (70), we find that the time to learn modes of strengths $s_{proj}$ in the regime when for $s_{proj} \geq s_1$ is (and therefore also $s_1$ since $a$ converges at same speed in MTL),

$$t = \frac{K\tau}{2(s_1 L + s_{proj})} \ln \frac{b_f^2 \left((s_1 L + s_{proj}) - (K(L^2 + 1))b_0^2\right)}{b_0^2 \left((s_1 L + s_{proj}) - (K(L^2 + 1))b_f^2\right)} \tag{76}$$

When the difference in strength is significant, $K \approx 1$ and $L << 1$, such that the time corresponds to

$$t_{s_{proj} > \mathbf{s}_1} = \frac{K\tau}{2s_{proj}} \ln \frac{|b_f^2(s_{proj} - Kb_0^2)|}{|b_0^2((s_{proj} - Kb_f^2)|} \tag{77}$$

which for $K = 1$ is equal to the time needed to learn a mode of $s_{proj}$ in a single task network, a shown in Saxe et al. (2019). However, if $K > 1$ the time needed to learn that same mode is longer in MTL than in STL (but keep in mind we are in fact learning two tasks and two modes). In contrast, in the regime $s_1 \approx s_{proj}$ we learn that the speed is equal to

$$t_{s_{proj} \sim s_1} = \frac{K\tau}{4s_{proj}} \ln \frac{|b_f^2(s_{proj} - Kb_0^2)|}{|b_0^2((s_{proj} - Kb_f^2)|} \tag{78}$$

so for $K \approx 1$, this yields a speed-up of a factor of two as compared to single-task learning. Furthermore, note that in both regimes, the learning speed is determined by the magnitude of the strongest mode.

### B.7 SOLVING EQUATIONS FOR NON-UNIFORM WEIGHTS

To see what happens to the solutions when $\omega_i \neq 1$, first go back to the equation (68) and substitute $K' = K(w1L**2 + w2)$ and $s' = s_1(w1*L + w2*\frac{sp}{s1})$. This yields,

$$\frac{K'\tau}{2} \frac{1}{s_{proj}} \left( \ln|\frac{b_f^2}{b_0^2}| + \ln \frac{|(s - K'b_0^2)|}{|(s - K'b_f^2)|} \right) = t. \tag{79}$$

$$\frac{K'\tau}{2s'} \ln \frac{|b_f^2(s' - K'b_0^2)|}{|b_0^2((s' - K'b_f^2)|} = t \tag{80}$$

### B.8 TASK ALIGNMENT AND SINGULAR MODES

In the main paper, we connect these insights to the learning speed in deep linear networks.

Consider $\xi\xi^T = \mathbf{\Sigma}_{1o}^T\mathbf{\Sigma}_{1o} + \mathbf{\Sigma}_{2o}^T\mathbf{\Sigma}_{2o}$ and the ordered SVD's $\xi = \mathbf{U}_\xi\mathbf{S}_\xi\mathbf{V}_\xi$, $\mathbf{\Sigma}_{1o} = \mathbf{U}_1\mathbf{S}_1\mathbf{V}_1^T$ and $\mathbf{\Sigma}_{2o} = U_2\mathbf{S}_1\mathbf{V}_1^T$. Comparing these to each other yields

$$\xi\xi^T = \mathbf{\Sigma}_{1o}^T\mathbf{\Sigma}_{1o} + \mathbf{\Sigma}_{2o}^T\mathbf{\Sigma}_{2o} \tag{81}$$

$$V_\xi S_\xi^2 \mathbf{V}_\xi^T = \mathbf{V}_1^2\mathbf{S}_1^2\mathbf{V}_1^T + \mathbf{V}_2^2\mathbf{S}_2^2\mathbf{V}_2^T \tag{82}$$

such that $\forall i s_\xi^i = \sqrt{\lambda_i}$ where $\lambda_i$ is a singular value of $\mathbf{V}_1^2\mathbf{S}_1^2\mathbf{V}_1^T + \mathbf{V}_2^2\mathbf{S}_2^2\mathbf{V}_2^T$. Now for perfectly aligned tasks, take $V = \mathbf{V}_1 = \mathbf{V}_2$, such that

$$\xi\xi^T = \mathbf{V}(\mathbf{S}_1^2 + \mathbf{S}_2^2)\mathbf{V}^T \tag{83}$$

and $\lambda_i = s_{i,1}^2 + s_{i,2}^2$. In contrast, when the tasks are completely orthogonal, we have $\mathbf{V}_1\mathbf{V}_2^T = 0$, or $span(\mathbf{V}_1) \perp span(\mathbf{V}_2)$ such that

$$X^TX = (\mathbf{V}_1\mathbf{V}_2) \begin{pmatrix} \mathbf{S}_1^2 & 0 \\ 0 & \mathbf{S}_2^2 \end{pmatrix} (\mathbf{V}_1\mathbf{V}_2)^T \tag{84}$$

such that $\{\lambda_i\} = \{s_{1,1}^2, s_{2,1}^2, \ldots, s_{r1,1}^2, s_{1,2}^2, \ldots, s_{r2,2}^2\}$, e.g., the ordered union of singular values from both tasks.

When tasks are conflicting, the way the singular values decompose into the initial task-specific singular values is more complex.

Next, we show this also tells us something about the rank of the matrix $\mathbf{S}_\xi$.

**Lemma 4** *Consider tasks $\mathcal{T}_i$ with rank $r_i$, a shared input-dimension $N_1$ and output-i-specific dimension $N_{io}$. Consider the reduced singular value decompositions of the task-i-specific input-output*

*correlation matrices* $\boldsymbol{\Sigma}_{io} = \mathbf{U}_i \mathbf{S}_i \mathbf{V}_i^T \in \mathcal{R}^{N_{oi} \times N_1}$ *and Let* $\xi = \begin{pmatrix} \boldsymbol{\Sigma}_{1o} \\ \dots \\ \boldsymbol{\Sigma}_{N_T o} \end{pmatrix} \in \mathcal{R}^{\sum_i^{N_T} N_{oi} \times N_1}$. *Then,*

*the following holds. If tasks* $\mathbf{T}_i$ *are aligned,*

$$Rank(\mathbf{S}_\xi) \leq max(\{r_1, \dots, r_{N_T}\}) < \sum_i^{N_T} r_i \tag{85}$$

*while for orthogonal tasks,*

$$Rank(\mathbf{S}_\xi) = \sum_i^{N_T} Rank(\mathbf{S}_i). \tag{86}$$

**Proof 1** *First note that the fact that the input is shared among all tasks implies that the number of columns in* $\boldsymbol{\Sigma}_{io}$ *is the same for all tasks and equal to* $N_1$*, so,* $\text{rank}(\xi) = \text{rank}(\mathbf{S}_\xi) = \dim(\text{Col}(\xi))$. *Then, using a standard Linear Algebra Theorem, we have that,*

$$dim(\sum \boldsymbol{\Sigma}_{io}) = \sum_i^{N_T} dim(\boldsymbol{\Sigma}_{i0}) - dim(\boldsymbol{\Sigma}_{1o} \cap \dots \boldsymbol{\Sigma}_{N_T o}). \tag{87}$$

*which proves the lemma.*

This corresponds to the intuition presented in Wu et al. (2020) concerning the necessary capacity of the shared network to induce interactions between tasks.

### B.9 EFFECT OF DEPTH

As shown in Saxe et al. (2014; 2019), the main qualitative difference in learning dynamics for deeper linear networks is observed between the case of a shallow network (only one layer and no hidden layer) and deeper networks. The extra layer impacts the learning speed but keeps the stage-like transitions and formatting of internal representations Saxe et al. (2019). In principle, there is no reason why this should be different in the case of multiple tasks. We confirm this in Appendix D, where we present results on deeper networks.

## C MULTI-TASK RICCATI EQUATIONS

In this section, we describe another way of solving the MTL gradient flow equations corresponding to the Riccati formalism discussed in the main paper. This formalism allows solutions for a broader family of initial conditions.

Defining,

$$\mathbf{Q}_{MTL} = \begin{pmatrix} \mathbf{W}_S^T \\ \mathbf{W}_1 \\ \mathbf{W}_2 \end{pmatrix} \qquad \mathbf{R}_{MTL} = \begin{pmatrix} 0 & \boldsymbol{\Sigma}_{1o}^T & \boldsymbol{\Sigma}_{2o}^T \\ \boldsymbol{\Sigma}_{1o} & 0 & 0 \\ \boldsymbol{\Sigma}_{2o} & 0 & 0 \end{pmatrix}, \tag{88}$$

allows for writing the gradient flow equations as Riccati equations and, under some assumptions, obtaining analytical expressions of the learning dynamics of the task-specific network functions, $\mathbf{W}_i \mathbf{W}_S(t)$ and task-specific training losses $\mathcal{L}_i$ and the representational similarity matrices (RSM).

### C.1 ASSUMPTIONS & DISCUSSION

The derivations discussed in the main paper, which are an extension of Braun et al. (2022) to the case of MTL, depend on the following assumptions,

**Assumption 1** *The inputs are whitened, i.e.,* $\boldsymbol{\Sigma}_{11} = \boldsymbol{I}$.

**Assumption 2** *The shared and task-specific weights are* $\boldsymbol{\Gamma}$-*balanced at initialization such that* $\mathbf{W}_S(0)\mathbf{W}_S(0)^T - \mathbf{W}_1^T(0)\mathbf{W}_1(0) - \mathbf{W}_2(0)^T\mathbf{W}_2(0) = \boldsymbol{\Gamma}$. *2bis. Zero-balanced at initialization such that* $\mathbf{W}_S(0)\mathbf{W}_S(0)^T = \mathbf{W}_1^T(0)\mathbf{W}_1(0) + \mathbf{W}_2(0)^T\mathbf{W}_2(0)$.

**Assumption 3** *The input-output correlations of the tasks and the initial state of the network functions have full rank, i.e., $\forall i \in \{1,2\} : rank(\boldsymbol{\Sigma}_{io}) = min(N_1, N_{io})$, ensuring the shared space can hold all the task-relevant input-output modes for both tasks $N_s \geq min(N_1, N_{1o} + N_{2o})$ (i.e., no bottleneck).*

These are not different from the assumptions in STL literature (apart from Assumption II). A detailed discussion about these assumptions and possible relaxations is presented in Dominé et al. (2025); Tarmoun et al. (2021). Furthermore, Dominé et al. (2025) shows that, Under the assumption of whitened inputs, the charge $\boldsymbol{\Gamma}$ is constant. As such, zero-balanced weights at initialization assure zero-balanced weight throughout learning, which enables us to use the Riccati solution for late times as well.

## C.2 DERIVATION OF THE MTL RICATI EQUATIONS

**Lemma 5** *Under assumption I, the continuous-time dynamics of the matrix $\mathbf{Q}_{MTL}\mathbf{Q}_{MTL}^T$ from an initial state $\mathbf{Q}_{MTL}(0)$ determined by an initialization $\mathbf{W}_S(0), \mathbf{W}_1(0), \mathbf{W}_2(0)$,*

$$\tau \frac{d(\mathbf{Q}_{MTL}\mathbf{Q}_{MTL}^T)}{dt} = \mathbf{R}_{MTL}\mathbf{Q}_{MTL}\mathbf{Q}_{MTL}^T + \mathbf{Q}_{MTL}\mathbf{Q}_{MTL}^T\mathbf{R}_{MTL} - (\mathbf{Q}_{MTL}\mathbf{Q}_{MTL}^T)^2 \quad (89)$$

$$- \mathbf{Q}_{MTL}^T\boldsymbol{\Gamma}(t)\mathbf{Q}_{MTL}, \quad (90)$$

*where $\boldsymbol{\Gamma}_0$ is a constant matrix describing the difference in covariance between the shared and task-specific weights.*

**Proof 2** *First note that,*

$$\mathbf{Q}_{MTL}\mathbf{Q}_{MTL}^T = \begin{pmatrix} \mathbf{W}_S^T\mathbf{W}_S & \mathbf{W}_S^T\mathbf{W}_1^T & \mathbf{W}_S^T\mathbf{W}_2^T \\ \mathbf{W}_1\mathbf{W}_S & \mathbf{W}_1\mathbf{W}_1^T & \mathbf{W}_1\mathbf{W}_2^T \\ \mathbf{W}_2\mathbf{W}_S & \mathbf{W}_2\mathbf{W}_1^T & \mathbf{W}_2\mathbf{W}_2^T \end{pmatrix}, \quad (91)$$

*such that,*

$$\tau \frac{d(\mathbf{Q}_{MTL}\mathbf{Q}_{MTL}^T)}{dt} = \tau \begin{pmatrix} \frac{d\mathbf{W}_S^T}{dt}\mathbf{W}_S + \mathbf{W}_S^T\frac{d\mathbf{W}_S}{dt} & \frac{d\mathbf{W}_S^T}{dt}\mathbf{W}_1^T + \mathbf{W}_S^T\frac{d\mathbf{W}_1^T}{dt} & \frac{d\mathbf{W}_S^T}{dt}\mathbf{W}_2^T + \mathbf{W}_S^T\frac{d\mathbf{W}_2^T}{dt} \\ \frac{d\mathbf{W}_1}{dt}\mathbf{W}_S + \mathbf{W}_1\frac{d\mathbf{W}_S}{dt} & \frac{d\mathbf{W}_1}{dt}\mathbf{W}_1^T + \mathbf{W}_1\frac{d\mathbf{W}_1^T}{dt} & \frac{d\mathbf{W}_1}{dt}\mathbf{W}_2^T + \mathbf{W}_1\frac{d\mathbf{W}_2^T}{dt} \\ \frac{d\mathbf{W}_2}{dt}\mathbf{W}_S + \mathbf{W}_2\frac{d\mathbf{W}_S}{dt} & \frac{d\mathbf{W}_2}{dt}\mathbf{W}_1^T + \mathbf{W}_2\frac{d\mathbf{W}_1^T}{dt} & \frac{d\mathbf{W}_2}{dt}\mathbf{W}_2^T + \mathbf{W}_2\frac{d\mathbf{W}_2^T}{dt} \end{pmatrix}$$

$$\quad (92)$$

$$\equiv \begin{pmatrix} A & B & C \\ D & E & F \\ G & H & I \end{pmatrix}. \quad (93)$$

Assuming Whitened Inputs (Assumption **I**) allows us to use the equations (35). Then, one obtains the explicit expression of the components A-I in $\frac{d(\mathbf{Q}_{MTL}\mathbf{Q}_{MTL}^T)}{dt}$,

$$A = (\boldsymbol{\Sigma}_1^T\mathbf{W}_1 + \boldsymbol{\Sigma}_2^T\mathbf{W}_2)\mathbf{W}_S + \mathbf{W}_S^T(\mathbf{W}_1^T\boldsymbol{\Sigma}_1 + \mathbf{W}_2^T\boldsymbol{\Sigma}_2) \tag{94}$$

$$- \mathbf{W}_S^T(\mathbf{W}_1^T\mathbf{W}_1 + \mathbf{W}_2^T\mathbf{W}_2)\mathbf{W}_S - \mathbf{W}_S^T(\mathbf{W}_1^T\mathbf{W}_1 + \mathbf{W}_2^T\mathbf{W}_2)\mathbf{W}_S \tag{95}$$

$$B = (\boldsymbol{\Sigma}_1^T\mathbf{W}_1 + \boldsymbol{\Sigma}_2^T\mathbf{W}_2)\mathbf{W}_1^T + \mathbf{W}_S^T\mathbf{W}_S\boldsymbol{\Sigma}_1^T - \mathbf{W}_S^T(\mathbf{W}_1^T\mathbf{W}_1 + \mathbf{W}_2^T\mathbf{W}_2)\mathbf{W}_1^T \tag{96}$$

$$- \mathbf{W}_S^T\mathbf{W}_S\mathbf{W}_S^T\mathbf{W}_1^T \tag{97}$$

$$C = (\boldsymbol{\Sigma}_1^T\mathbf{W}_1 + \boldsymbol{\Sigma}_2^T\mathbf{W}_2)\mathbf{W}_2^T + \mathbf{W}_S^T\mathbf{W}_S\boldsymbol{\Sigma}_2^T - \mathbf{W}_S^T(\mathbf{W}_1^T\mathbf{W}_1 + \mathbf{W}_2^T\mathbf{W}_2)\mathbf{W}_2^T \tag{98}$$

$$- \mathbf{W}_S^T\mathbf{W}_S\mathbf{W}_S^T\mathbf{W}_2^T \tag{99}$$

$$D = \boldsymbol{\Sigma}_1\mathbf{W}_S^T\mathbf{W}_S + \mathbf{W}_1(\mathbf{W}_1^T\boldsymbol{\Sigma}_1^T + \mathbf{W}_2^T\boldsymbol{\Sigma}_2^T) - \mathbf{W}_1\mathbf{W}_S\mathbf{W}_S^T\mathbf{W}_S \tag{100}$$

$$- \mathbf{W}_1(\mathbf{W}_1^T\mathbf{W}_1 + \mathbf{W}_2^T\mathbf{W}_2)\mathbf{W}_S \tag{101}$$

$$E = \boldsymbol{\Sigma}_1\mathbf{W}_S^T\mathbf{W}_1^T + \mathbf{W}_1\mathbf{W}_S\boldsymbol{\Sigma}_1^T - \mathbf{W}_1\mathbf{W}_S\mathbf{W}_S^T\mathbf{W}_1^T - \mathbf{W}_1\mathbf{W}_S\mathbf{W}_S^T\mathbf{W}_1^T \tag{102}$$

$$F = \boldsymbol{\Sigma}_1\mathbf{W}_S^T\mathbf{W}_2^T + \mathbf{W}_1\mathbf{W}_S\boldsymbol{\Sigma}_2^T - \mathbf{W}_1\mathbf{W}_S\mathbf{W}_S^T\mathbf{W}_2^T - \mathbf{W}_1\mathbf{W}_S\mathbf{W}_S^T\mathbf{W}_2^T \tag{103}$$

$$G = \boldsymbol{\Sigma}_2\mathbf{W}_S^T\mathbf{W}_S + \mathbf{W}_2(\mathbf{W}_1^T\boldsymbol{\Sigma}_1^T + \mathbf{W}_2^T\boldsymbol{\Sigma}_2^T) - \mathbf{W}_2\mathbf{W}_S\mathbf{W}_S^T\mathbf{W}_S \tag{104}$$

$$- \mathbf{W}_2(\mathbf{W}_1^T\mathbf{W}_1 + \mathbf{W}_2^T\mathbf{W}_2)\mathbf{W}_S \tag{105}$$

$$H = \boldsymbol{\Sigma}_2\mathbf{W}_S^T\mathbf{W}_1^T + \mathbf{W}_2\mathbf{W}_S\boldsymbol{\Sigma}_1^T - \mathbf{W}_S\mathbf{W}_S\mathbf{W}_S^T\mathbf{W}_1^T - \mathbf{W}_2\mathbf{W}_S\mathbf{W}_S^T\mathbf{W}_1^T \tag{106}$$

$$I = \boldsymbol{\Sigma}_2\mathbf{W}_S^T\mathbf{W}_S + \mathbf{W}_2\mathbf{W}_S\boldsymbol{\Sigma}_1^T - \mathbf{W}_2\mathbf{W}_S\mathbf{W}_S^T\mathbf{W}_2^T - \mathbf{W}_2\mathbf{W}_S\mathbf{W}_S^T\mathbf{W}_2^T \tag{107}$$

For completeness, we show that the RHS in equation (90). Essentially, we have to show that the components of the right-hand side of equation (14),

$$\mathbf{R}_{MTL}\mathbf{Q}_{MTL}\mathbf{Q}_{MTL}^T + \mathbf{Q}_{MTL}\mathbf{Q}_{MTL}^T - (\mathbf{Q}_{MTL}\mathbf{Q}_{MTL}^T)^2 - \mathbf{Q}_{MTL}\boldsymbol{\Gamma}_0\mathbf{Q}_{MTL} \equiv \begin{pmatrix} A' & B' & C' \\ D' & E' & F \\ G' & H' & I' \end{pmatrix}, \tag{108}$$

is equal to the left-hand side. In particular, we show that components (A'-I') are equal to components (A-I). To do so, note that

$$\mathbf{R}_{MTL}\mathbf{Q}_{MTL}\mathbf{Q}_{MTL}^T = \begin{pmatrix} \boldsymbol{\Sigma}_1^T\mathbf{W}_1\mathbf{W}_S + \boldsymbol{\Sigma}_2^T\mathbf{W}_2\mathbf{W}_S & \boldsymbol{\Sigma}_1^T\mathbf{W}_1\mathbf{W}_1^T + \boldsymbol{\Sigma}_2^T\mathbf{W}_2\mathbf{W}_1^T & \boldsymbol{\Sigma}_1^T\mathbf{W}_1\mathbf{W}_1^T + \boldsymbol{\Sigma}_2^T\mathbf{W}_2\mathbf{W}_2^T \\ \boldsymbol{\Sigma}_1\mathbf{W}_S^T\mathbf{W}_S & \boldsymbol{\Sigma}_1\mathbf{W}_S^T\mathbf{W}_1^T & \boldsymbol{\Sigma}_1\mathbf{W}_S^T\mathbf{W}_2^T \\ \boldsymbol{\Sigma}_2\mathbf{W}_S^T\mathbf{W}_S & \boldsymbol{\Sigma}_2\mathbf{W}_S^T\mathbf{W}_1^T & \boldsymbol{\Sigma}_2\mathbf{W}_S^T\mathbf{W}_2^T \end{pmatrix} \tag{109}$$

$$\mathbf{Q}_{MTL}\mathbf{Q}_{MTL}^T\mathbf{R}_{MTL} = \begin{pmatrix} \mathbf{W}_S^T\mathbf{W}_1^T\boldsymbol{\Sigma}_1 + \mathbf{W}_S^T\mathbf{W}_2^T\boldsymbol{\Sigma}_2 & \mathbf{W}_S^T\mathbf{W}_S\boldsymbol{\Sigma}_1^T & \mathbf{W}_S^T\mathbf{W}_S\boldsymbol{\Sigma}_2^T \\ \mathbf{W}_1\mathbf{W}_1^T\boldsymbol{\Sigma}_1 + \mathbf{W}_1\mathbf{W}_2^T\boldsymbol{\Sigma}_2 & \mathbf{W}_1\mathbf{W}_S\boldsymbol{\Sigma}_1^T & \mathbf{W}_1\mathbf{W}_S\boldsymbol{\Sigma}_2^T \\ \mathbf{W}_2\mathbf{W}_1^T\boldsymbol{\Sigma}_1 + \mathbf{W}_2\mathbf{W}_2^T\boldsymbol{\Sigma}_2 & \mathbf{W}_2\mathbf{W}_S\boldsymbol{\Sigma}_1^T & \mathbf{W}_2\mathbf{W}_S\boldsymbol{\Sigma}_2^T \end{pmatrix} \tag{110}$$

$$(\mathbf{Q}_{MTL}\mathbf{Q}_{MTL}^T)^2 \equiv \begin{pmatrix} A'' & B'' & C'' \\ D'' & E'' & F'' \\ G'' & H'' & I'' \end{pmatrix} \tag{111}$$

with $\tag{112}$

$$A'' = \mathbf{W}_S^T\mathbf{W}_S\mathbf{W}_S^T\mathbf{W}_S + \mathbf{W}_S^T(\mathbf{W}_1^T\mathbf{W}_1 + \mathbf{W}_2^T\mathbf{W}_2)\mathbf{W}_S \tag{113}$$

$$B'' = \mathbf{W}_S^T\mathbf{W}_S\mathbf{W}_S^TW_1^T + \mathbf{W}_S^T(\mathbf{W}_1^T\mathbf{W}_1 + \mathbf{W}_S^T\mathbf{W}_2)\mathbf{W}_1^T \tag{114}$$

$$C'' = \mathbf{W}_S^T\mathbf{W}_S\mathbf{W}_S^TW_2^T + \mathbf{W}_S^T(\mathbf{W}_1^T\mathbf{W}_1 + \mathbf{W}_S^T\mathbf{W}_2)\mathbf{W}_2^T \tag{115}$$

$$D'' = \mathbf{W}_1\mathbf{W}_S\mathbf{W}_S^T\mathbf{W}_S + \mathbf{W}_1(\mathbf{W}_1^T\mathbf{W}_1 + \mathbf{W}_S^T\mathbf{W}_2)\mathbf{W}_S \tag{116}$$

$$E'' = \mathbf{W}_1\mathbf{W}_S\mathbf{W}_S^T\mathbf{W}_1^T + \mathbf{W}_1(\mathbf{W}_1^T\mathbf{W}_1 + \mathbf{W}_S^T\mathbf{W}_2)\mathbf{W}_1^T \tag{117}$$

$$F'' = \mathbf{W}_1\mathbf{W}_S\mathbf{W}_S^T\mathbf{W}_2^T + \mathbf{W}_1(\mathbf{W}_1^T\mathbf{W}_1 + \mathbf{W}_S^T\mathbf{W}_2)\mathbf{W}_2^T \tag{118}$$

$$G'' = \mathbf{W}_2\mathbf{W}_S\mathbf{W}_S^T\mathbf{W}_S + \mathbf{W}_2\mathbf{W}_1^T\mathbf{W}_1\mathbf{W}_S + \mathbf{W}_2\mathbf{W}_2^T\mathbf{W}_2\mathbf{W}_S \tag{119}$$

$$H'' = \mathbf{W}_2\mathbf{W}_S\mathbf{W}_S^T\mathbf{W}_1^T + \mathbf{W}_2\mathbf{W}_1^T\mathbf{W}_1\mathbf{W}_1^T + \mathbf{W}_2\mathbf{W}_2^T\mathbf{W}_2\mathbf{W}_1^T \tag{120}$$

$$I'' = \mathbf{W}_2\mathbf{W}_S\mathbf{W}_S^T\mathbf{W}_2^T + \mathbf{W}_2\mathbf{W}_1^T\mathbf{W}_1\mathbf{W}_2^T + \mathbf{W}_2\mathbf{W}_2^T\mathbf{W}_2\mathbf{W}_2^T \tag{121}$$

*Such that,*

$$A' = (\mathbf{\Sigma}_1^T \mathbf{W}_1 + \mathbf{\Sigma}_2^T \mathbf{W}_2)\mathbf{W}_S + \mathbf{W}_S^T(\mathbf{W}_1^T \mathbf{\Sigma}_1 + \mathbf{W}_2^T \mathbf{\Sigma}_2) \tag{122}$$
$$- \mathbf{W}_S^T \mathbf{W}_S \mathbf{W}_S^T \mathbf{W}_S - \mathbf{W}_S^T(\mathbf{W}_1^T \mathbf{W}_1 + \mathbf{W}_S^T \mathbf{W}_2)\mathbf{W}_S \tag{123}$$
$$B' = (\mathbf{\Sigma}_1^T \mathbf{W}_1 + \mathbf{\Sigma}_2 \mathbf{W}_2)\mathbf{W}_1^T + \mathbf{W}_S^T \mathbf{W}_S \mathbf{\Sigma}_1^T - \mathbf{W}_S^T(\mathbf{W}_1^T \mathbf{W}_1 + \mathbf{W}_S^T \mathbf{W}_2)\mathbf{W}_1^T \tag{124}$$
$$- \mathbf{W}_S^T \mathbf{W}_S \mathbf{W}_S^T \mathbf{W}_1^T \tag{125}$$
$$C' = (\mathbf{\Sigma}_1^T \mathbf{W}_1 + \mathbf{\Sigma}_2 \mathbf{W}_2)\mathbf{W}_2^T + \mathbf{W}_S^T \mathbf{W}_S \mathbf{\Sigma}_2^T - \mathbf{W}_S^T(\mathbf{W}_1^T \mathbf{W}_1 + \mathbf{W}_S^T \mathbf{W}_2)\mathbf{W}_2^T \tag{126}$$
$$- \mathbf{W}_S^T \mathbf{W}_S \mathbf{W}_S^T \mathbf{W}_2^T \tag{127}$$
$$D' = \mathbf{\Sigma}_1 \mathbf{W}_S^T \mathbf{W}_S + \mathbf{W}_1(\mathbf{W}_1^T \mathbf{\Sigma}_1^T + \mathbf{W}_2^T \mathbf{\Sigma}_2^T) - \mathbf{W}_1 \mathbf{W}_S \mathbf{W}_S^T \mathbf{W}_S \tag{128}$$
$$- \mathbf{W}_1(\mathbf{W}_1^T \mathbf{W}_1 + \mathbf{W}_S^T \mathbf{W}_2)\mathbf{W}_S \tag{129}$$
$$E' = \mathbf{\Sigma}_1 \mathbf{W}_S^T \mathbf{W}_1^T + \mathbf{W}_1 \mathbf{W}_S \mathbf{\Sigma}_1^T - \mathbf{W}_1 \mathbf{W}_S \mathbf{W}_S^T \mathbf{W}_1^T - \mathbf{W}_1(\mathbf{W}_1^T \mathbf{W}_1 + \mathbf{W}_2^T \mathbf{W}_2)\mathbf{W}_1^T \tag{130}$$
$$F' = \mathbf{\Sigma}_1 \mathbf{W}_S^T \mathbf{W}_2^T + \mathbf{W}_1 \mathbf{W}_S \mathbf{\Sigma}_2^T - \mathbf{W}_1 \mathbf{W}_S \mathbf{W}_S^T \mathbf{W}_2^T - \mathbf{W}_1(\mathbf{W}_1^T \mathbf{W}_1 - \mathbf{W}_2^T \mathbf{W}_2)\mathbf{W}_2^T \tag{131}$$
$$G' = \mathbf{\Sigma}_2 \mathbf{W}_S^T \mathbf{W}_S + \mathbf{W}_2(\mathbf{W}_1^T \mathbf{\Sigma}_1 + \mathbf{W}_2^T \mathbf{\Sigma}_2) - \mathbf{W}_2 \mathbf{W}_S \mathbf{W}_S^T \mathbf{W}_S \tag{132}$$
$$- \mathbf{W}_2(\mathbf{W}_1^T \mathbf{W}_1 + \mathbf{W}_2^T \mathbf{W}_2)\mathbf{W}_S \tag{133}$$
$$H' = \mathbf{\Sigma}_2 \mathbf{W}_S^T \mathbf{W}_1^T + \mathbf{W}_2 \mathbf{W}_S \mathbf{\Sigma}_1^T - \mathbf{W}_S \mathbf{W}_S \mathbf{W}_S^T \mathbf{W}_1^T - \mathbf{W}_2(\mathbf{W}_1^T \mathbf{W}_1 + \mathbf{W}_2^T \mathbf{W}_2)\mathbf{W}_1^T \tag{134}$$
$$I' = \mathbf{\Sigma}_2 \mathbf{W}_S^T \mathbf{W}_2^T + \mathbf{W}_S \mathbf{W}_S \mathbf{\Sigma}_2^T - \mathbf{W}_2 \mathbf{W}_S \mathbf{W}_S^T \mathbf{W}_2^T - \mathbf{W}_2(\mathbf{W}_1^T \mathbf{W}_1 + \mathbf{W}_2^T \mathbf{W}_2)\mathbf{W}_2^T \tag{135}$$

*For components $BC, D, G$ of equation (90) the equality to $B', C', D', G'$ of the right hand side is trivial without any further assumption. In contrast, the other components $(A, E, F, H, I)$ differ from their counterparts $(A', E', F', H', I')$ by the difference in covariance between the shared weights and task-specific weights, which, under assumption, we can write as*

$$\mathbf{W}_S \mathbf{W}_S^T - \mathbf{W}_1^T \mathbf{W}_1 - \mathbf{W}_2^T \mathbf{W}_2 \equiv \mathbf{\Gamma}(t). \tag{136}$$

*Using equation (136) in the remaining components $(A', E', F', H', I')$ proves the Lemma.*

**Lemma 6** *Assuming whitened inputs (assumption I) and balanced weights at initialization (assumption II) with a charge equal to $\mathbf{\Gamma}_0$, the gradient flow equations of $\mathbf{Q}_{MTL}\mathbf{Q}_{MTL}$ can be written as,*

$$\tau \frac{d(\mathbf{Q}_{MTL}\mathbf{Q}_{MTL}^T)}{dt} = \mathbf{R}_{MTL}\mathbf{Q}_{MTL}\mathbf{Q}_{MTL}^T + \mathbf{Q}_{MTL}\mathbf{Q}_{MTL}^T \mathbf{R}_{MTL} - (\mathbf{Q}_{MTL}\mathbf{Q}_{MTL}^T)^2 \tag{137}$$
$$- \mathbf{Q}_{MTL}^T \mathbf{\Gamma}_0 \mathbf{Q}_{MTL}, \tag{138}$$

**Proof 3** *As shown in Dominé et al. (2025), $\mathbf{\Gamma} \in \mathcal{R}^{N_s \times N_s}$ is constant, i.e., so the imbalance is determined by the initial imbalance $\mathbf{\Gamma}_0$. Using this in the proof of Lemma 5 proves equation (138).*

As shown in Dominé et al. (2025), the charge $\mathbf{\Gamma}_0$ controls the behavior between the Lazy and Rich Regime. In this work, we focus on the inter-task dynamics of zero-balanced weights (i.e., $\mathbf{\Gamma}_0 = 0$) which constraints the model to the Rich Regime.

**Lemma 7** *Assuming whitened inputs (assumption **I**) and zero-balanced weights at initialization (assumption **IIbis**), the gradient flow equations of $\mathbf{Q}_{MTL}\mathbf{Q}_{MTL}$ can be written as,*

$$\tau \frac{d(\mathbf{Q}_{MTL}\mathbf{Q}_{MTL}^T)}{dt} = \mathbf{R}_{MTL}\mathbf{Q}_{MTL}\mathbf{Q}_{MTL}^T + \mathbf{Q}_{MTL}\mathbf{Q}_{MTL}^T \mathbf{R}_{MTL} - (\mathbf{Q}_{MTL}\mathbf{Q}_{MTL}^T)^2. \tag{139}$$

**Proof 4** *Replacing $\mathbf{\Gamma}_0$ with a matrix filled with zero's in equation (138) proves the statement.*

C.3 SOLUTION TO THE MTL RICATI EQUATIONS: ZERO-BALANCED WEIGHTS

As is the case in Fukumizu (1998); Braun et al. (2022), this work primarily focuses on the setting where the shared and task-specific weights are zero-balanced ($\mathbf{\Gamma}_0 = 0_{N_S \times N_S}$).

**Lemma 8** *Under assumption 1 (whitened inputs), assumption 2bis (zero-balanced weights) and assumption 3 (full-rank), the MTL Ricati equation (14) has a unique solution $\forall t > 0$ for the case*

*where $N_{1o} + N_{2o} = N_1$, given by*

$$\mathbf{Q}_{MTL}\mathbf{Q}_{MTL}^T(t) \tag{140}$$

$$= e^{\mathbf{R}_{MTL}\frac{t}{\tau}}\mathbf{Q}_{MTL}(0)\left[\mathcal{I} + \frac{1}{2}\mathbf{Q}_{MTL}(0)^T(e^{\mathbf{R}_{MTL}\frac{t}{\tau}}\mathbf{R}_{MTL}^{-1}e^{\mathbf{R}_{MTL}\frac{t}{\tau}} - \mathbf{R}_{MTL}^{-1})\mathbf{Q}_{MTL}(0)\right]^{-1} \tag{141}$$

$$\times \mathbf{Q}_{MTL}(0)^T e^{\mathbf{R}_{MTL}\frac{t}{\tau}}, \tag{142}$$

*where $Q(0)$ is assumed to be full rank and corresponds to the initial state.*

**Proof 5** *As shown in Fukumizu (1998), equation (139) has a unique solution under assumptions I-II-III-IV. This can be shown explicitly by inserting equation (142) into (139).*

Note that assumption IV can be relaxed for the theorem, where the approach in Braun et al. (2022) is used to rewrite the solution (142) in a more interpretable way. Namely, defining,

$$\xi = \begin{pmatrix} \mathbf{\Sigma}_{1o} \\ \mathbf{\Sigma}_{2o} \end{pmatrix} \qquad \mathbf{W}_{MTL}^0 = \begin{pmatrix} \mathbf{W}_1(0)\mathbf{W}_S(0) \\ \mathbf{W}_2(0)\mathbf{W}_S(0) \end{pmatrix}, \tag{143}$$

such that $\xi, \mathbf{W}_{MTL}^0 \in \mathcal{R}^{N_{1o}+N_{2o}\times N_1}$, allows us to consider the compact singular value decomposition of the initial network and the input-output correlation matrix of the shared-task space,

$$\xi = U_\xi \mathbf{S}_\xi V_\xi^T \qquad W_{MTL}^0 = U_0 S_0 V_0. \tag{144}$$

Then $\mathbf{S}_\xi, \mathbf{S}_0 \in \mathcal{R}^{m\times m}$ with $m = min(N_{1o}+N_{2o}, N_1)$ represents the combined strength of the correlations contained in the two input-output matrices $\mathbf{\Sigma}_{1o}, \mathbf{\Sigma}_{2o}$, they are not simply the singular values of the individual matrices but rather reflect their joint contribution. Furthermore, $\mathbf{U}_\xi \in \mathcal{R}^{N_{1o}+N_{2o}\times m}$ and $V_\xi^T \in \mathcal{R}^{m\times N_1}$.

Similar to the single-task case discussed in Braun et al. (2022), if $N_1 \neq N_{1o}+N_{2o}$, then, in general, $\mathbf{U}_\xi$ and $\mathbf{V}_\xi$ are not square and orthonormal. Namely, if $N_1 < N_{1o}+N_{2o}$, then $m = N_1$ such that $\mathbf{U}_\xi^T\mathbf{U}_\xi = \mathbf{V}_\xi^T\mathbf{V}_\xi = \mathbf{V}_\xi\mathbf{V}_\xi^T = \mathcal{I} \in \mathcal{R}^{N_1\times N_1}$ while $\mathbf{U}_\xi\mathbf{U}_\xi^T \neq \mathcal{I}$ in $\mathcal{R}^{N_{1o}+N_{2o}\times N_{1o}+N_{2o}}$. In that case, we can define $\mathbf{U}_\xi^\perp \in \mathcal{R}^{N_{1o}+N_{2o}\times N_{1o}+N_{2o}-N_1}$ to denote the matrix that contains orthogonal column vectors such that concatenation $[U_\xi U_\xi^\perp] \in \mathcal{R}^{N_{1o}+N_{2o},N_{1o}+N_{2o}}$ is orthonormal. In that case, $V_\xi^\perp \in \mathcal{R}^{N_1\times N_{1o}+N_{2o}-N_i}$ denotes a matrix of zeros.

Converserly, for the situation: $N_1 > N_{1o}+N_{2o}$, in that case, $m = N_{1o}+N_{2o}$ such that $U_\xi U_\xi^T = U_\xi^T U_\xi = \mathbf{V}_\xi^T V = \mathcal{I} \in \mathcal{R}^{N_{1o}+N_{2o}\times N_{1o}+N_{2o}}$ but $\mathbf{V}_\xi\mathbf{V}_\xi^T \neq I \in \mathcal{R}^{N_1\times N_1}$. Defining $V_\xi^\perp \in \mathcal{R}^{N_1\times(N_1-N_{1o}+N_{2o})}$ such that concat $[\mathbf{V}_\xi\mathbf{V}_\xi^\perp] \in \mathcal{R}^{N_1\times N_1}$ is orthonormal and $U_{\xi\perp} \in \mathcal{R}^{N_{1o}+N_{2o}\times N_{1o}+N_{2o}-N_1}$ denotes a matrix of zeros.

Finally introducing $\mathbf{A}_\xi$ and $\mathbf{B}_\xi$,

$$\mathbf{A}_\xi = \mathbf{U}_0^T\mathbf{U}_\xi + \mathbf{V}_0^T\mathbf{V}_\xi \qquad \mathbf{B}_\xi = \mathbf{U}_0^T\mathbf{U}_\xi - \mathbf{V}_0^T\mathbf{V}_\xi, \tag{145}$$

allows an interpretable solution for $\mathbf{Q}_{MTL}\mathbf{Q}_{MTL}^T(t)$, as formulated in the next Theorem 2.

**Theorem 9** *Under assumption 1 (whitened inputs), assumption 2bis (zero-balanced weights), assumption 3 (full-rank) and non-singular $\mathbf{A}_\xi$, the MTL Riccati equation (14) has a unique solution $\forall t > 0$, given by,*

$$\mathbf{Q}_{MTL}\mathbf{Q}_{MTL}^T(t) \tag{146}$$

$$= \mathbf{Z}_{MTL}\left[4\,e^{-\mathbf{S}_\xi\frac{t}{\tau}}\mathbf{A}_\xi^{-1}\mathbf{S}_\xi^{-1}\left(\mathbf{A}_\xi^T\right)^{-1}e^{-\mathbf{S}_\xi\frac{t}{\tau}} + \left(\mathcal{I} - e^{-2\mathbf{S}_\xi\frac{t}{\tau}}\right)\mathbf{S}_\xi^{-1}\right. \tag{147}$$

$$- e^{-\mathbf{S}_\xi\frac{t}{\tau}}\mathbf{A}_\xi^{-1}\mathbf{B}\left(e^{-2\mathbf{S}_\xi\frac{t}{\tau}} - \mathcal{I}\right)\mathbf{S}_\xi^{-1}\mathbf{B}^T\left(\mathbf{A}_\xi^T\right)^{-1}e^{-\mathbf{S}_\xi\frac{t}{\tau}} \tag{148}$$

$$\left. + 4\frac{t}{\tau}e^{-\mathbf{S}_\xi\frac{t}{\tau}}\mathbf{A}_\xi^{-1}\left(\mathbf{V}_\xi^T\mathbf{V}_\xi^\perp(\mathbf{V}_\xi^\perp)^T\mathbf{V}_\xi + \mathbf{U}_\xi^T\mathbf{U}_\xi^\perp(\mathbf{U}_\xi^\perp)^T\mathbf{U}_\xi\right)\left(\mathbf{A}_\xi^T\right)^{-1}e^{-\mathbf{S}_\xi\frac{t}{\tau}}\right]^{-1}\mathbf{Z}_{MTL}^T, \tag{149}$$

$$\mathbf{Z}_{MTL} = \begin{pmatrix} \mathbf{V}_\xi\left(\mathcal{I} - e^{-\mathbf{S}_\xi\frac{t}{\tau}}\mathbf{B}_\xi^T(\mathbf{A}_\xi^T)^{-1}e^{-\mathbf{S}_\xi\frac{t}{\tau}}\right) + 2\mathbf{V}_\xi^\perp(\mathbf{V}_\xi^\perp)^T\mathbf{V}_\xi(\mathbf{A}_\xi^T)^{-1}e^{-\mathbf{S}_\xi\frac{t}{\tau}} \\ \mathbf{U}_\xi\left(\mathcal{I} + e^{-\mathbf{S}_\xi\frac{t}{\tau}}\mathbf{B}_\xi^T(\mathbf{A}_\xi^T)^{-1}e^{-\mathbf{S}_\xi\frac{t}{\tau}}\right) + 2\mathbf{U}_\xi^\perp(\mathbf{U}_\xi^\perp)^T\mathbf{U}_\xi(\mathbf{A}_\xi^T)^{-1}e^{-\mathbf{S}_\xi\frac{t}{\tau}} \end{pmatrix}. \tag{150}$$

**Proof 6** *Essentially, the proof consists of using the singular value decomposition of $\mathbf{R}_{MTL}$ to obtain explicit expressions for $\mathbf{R}_{MTL}^{-1}$ and $e^{\mathbf{R}_{MTL}}$ in terms of $\mathbf{S}_\xi$, yielding a more interpretable and stable analytical solution.*

*To do so, rewrite $\mathbf{R}_{MTL}$ as a $2 \times 2$ block-matrix. By completing the basis of the resulting block-matrix, one can take the same approach as used in Appendix C in Braun et al. (2022) to diagonalize $F$,*

$$\mathbf{R}_{MTL} = \begin{pmatrix} 0 & \mathbf{\Sigma}_{1o}^T & \mathbf{\Sigma}_{2o}^T \\ \mathbf{\Sigma}_{1o} & 0 & 0 \\ \mathbf{\Sigma}_{2o} & 0 & 0 \end{pmatrix} \tag{151}$$

$$= \begin{pmatrix} 0 & \xi^T \\ \xi & 0 \end{pmatrix} \tag{152}$$

$$= \frac{1}{\sqrt{2}} \begin{pmatrix} \mathbf{V}_\xi & \mathbf{V}_\xi & \sqrt{2}\mathbf{V}_\xi^\perp \\ \mathbf{U}_\xi & -\mathbf{U}_\xi & \sqrt{2}\mathbf{U}_\xi^\perp \end{pmatrix} \begin{pmatrix} \mathbf{S}_\xi & 0 & 0 \\ 0 & \mathbf{S}_\xi & 0 \\ 0 & 0 & 0 \end{pmatrix} \frac{1}{\sqrt{2}} \begin{pmatrix} \mathbf{V}_\xi & \mathbf{V}_\xi & \sqrt{2}\mathbf{V}_\xi^\perp \\ \mathbf{U}_\xi & -\mathbf{U}_\xi & \sqrt{2}\mathbf{U}_\xi^\perp \end{pmatrix}^T \tag{153}$$

$$= \mathbf{P}_{MTL}\Gamma_{MTL}\mathbf{P}_{MTL}^T \tag{154}$$

*Then, $\mathbf{P}_{MTL}\mathbf{P}_{MTL}^T = \mathcal{I}$ and therefore $\mathbf{P}_{MTL}^T = \mathbf{P}_{MTL}^{-1}$, enabling the computation of $\mathbf{R}_{MTL}^{-1}$ and $e^{R\frac{t}{\tau}}$ in equation (142). The remaining part of the proof is analogous to the proof presented in Appendix C2 of Braun et al. (2022) without any further assumptions.*

### C.4 PROOF OF THE CONVERGENCE BEHAVIOUR

**Theorem 10** *Under the assumptions of Theorem 2, $\mathbf{Q}_{MTL}\mathbf{Q}_{MTL}^T(t)$ converges to a steady state given by,*

$$\mathbf{Q}_{MTL}\mathbf{Q}_{MTL}^T(t_{steady}) = \begin{pmatrix} \mathbf{V}_\xi\mathbf{S}_\xi\mathbf{V}_\xi^T & \mathbf{V}_\xi\mathbf{S}_\xi\mathbf{U}_\xi^T \\ \mathbf{U}_\xi\mathbf{S}_\xi\mathbf{V}_\xi^T & \mathbf{U}_\xi\mathbf{S}_\xi\mathbf{U}_\xi^T \end{pmatrix}. \tag{155}$$

**Proof 7** *For $t \to \infty$ the negative exponentials appearing in equation (150) converge to zero $\lim_{t\to\infty} e^{-\mathbf{S}_\xi\frac{t}{\tau}} = 0$. Using this in equation (150) yields,*

$$\lim_{t\to\infty} \mathbf{Q}_{MTL}\mathbf{Q}_{MTL}^T(t) = \begin{pmatrix} \mathbf{V}_\xi \\ \mathbf{U}_\xi \end{pmatrix} S^{-1} \begin{pmatrix} \mathbf{V}_\xi^T & \mathbf{U}_\xi^T \end{pmatrix}, \tag{156}$$

*which proves the lemma.*

### C.5 EXTENSION TOWARDS MORE TASKS

In principle, the derivations also hold for more than two tasks ($N_T > 2$). In the case of an arbitrary number of tasks, one replaces $Q_{MTL}$ and $R_{MTL}$ to

$$\mathbf{Q}_{MTL} = \begin{pmatrix} \mathbf{W}_S^T \\ \mathbf{W}_1 \\ \mathbf{W}_2 \\ \vdots \\ \mathbf{W}_{N_T} \end{pmatrix} \qquad \mathbf{R}_{MTL} = \begin{pmatrix} 0 & \mathbf{\Sigma}_{1o}^T & \mathbf{\Sigma}_{2o}^T & \dots & \mathbf{\Sigma}_{N_To}^T \\ \mathbf{\Sigma}_{1o} & 0 & 0 & \dots & 0 \\ \mathbf{\Sigma}_{2o} & 0 & 0 & \dots & 0 \\ \vdots & \vdots & \vdots & \vdots & 0 \\ \mathbf{\Sigma}_{N_To} & \dots & \dots & \dots & 0 \end{pmatrix}, \tag{157}$$

and change $\xi$ and $\mathbf{W}_{MTL}^0$ accordingly.

## D EXPERIMENTS

All experiments were conducted using Google Colab and a local machine. Computations were run exclusively on CPUs, without the use of GPUs. The local system used an Intel Core i7-1165G7 processor (11th Gen, 4 cores, 8 threads, 2.80 GHz base frequency). The code was implemented in Python using the PyTorch framework. Simulations with toy data can be run ¡ 10 minutes runtime.

For Multi-MNIST we ran experiments on a subset of the data to decrease learning time. The corresponding code repository is available in the Supplemental Material and will be available on Github upon acceptance. The repository contains a README.md file with additional information about the functions and Python files. For the experiment with Multi-MNIST (Figure 4) we additionaly provide text files containing the tracked statistics of the analytical simulation and Neural Network. These files can be imported into the corresponding Mnist-notebook to reproduce the Figure.

### D.1 FULLY ALIGNED, CONFLICTING AND ORTHOGONAL TASKS

We construct two input-output correlation matrices to simulate the inter-task dynamics for perfectly aligned and orthogonal tasks. Note that, in this setup we need $d = N_1 = N_{o1} = N_{o2}$ and $d \geq 2r$ to match dimensions in the creation of the tasks and initial weights. To make sure the corresponding magnitudes are balanced, one can compute $\Delta M_{12}$ for the different dimensions of the tasks. If the tasks have to be aligned, one can normalize the created input-output correlation matrices. For the experiments in the paper we manually choose the values of the eigenvalues (equation 159) instead of randomly selecting them, to clearly distinguish between different modes in the visualization.

**Fully aligned tasks.** The input-output matrices of the perfectly aligned tasks are generated by choosing a random dimension $d$ and rank $r$ and the construction of an orthonormal basis $\mathbf{U}_{\text{shared}} \in \mathbb{R}^{d \times r}$, via QR decomposition:

$$\mathbf{U}_{\text{shared}, \_} = \text{QR}(\mathcal{N}(0, 1)^{d \times r}). \tag{158}$$

Next, random eigenvalue vectors are sampled for each task, e.g., for rank r=2,

$$\lambda^{(1)} = \begin{bmatrix} \lambda^1_{11} \\ \lambda^1_{12} \end{bmatrix}, \quad \lambda^{(2)} = \begin{bmatrix} \lambda^2_{11} \\ \lambda^2_{12} \end{bmatrix} \tag{159}$$

such that the covariance matrices can be constructed like,

$$\mathbf{\Sigma}_{1o} = \mathbf{U}_{\text{shared}} \, \text{diag}(\lambda^{(1)}) \, \mathbf{U}^\top_{\text{shared}}, \quad \mathbf{\Sigma}_{2o} = \mathbf{U}_{\text{shared}} \, \text{diag}(\lambda^{(2)}) \, \mathbf{U}^\top_{\text{shared}} \tag{160}$$

to acquire two fully aligned tasks.

**Fully orthogonal Tasks.** This time a higher dimensional orthonormal basis $Q_{\text{full}} \in \mathbb{R}^{d \times d}$ is generated via QR decomposition,

$$\mathbf{Q}_{\text{full},L, \_} = \text{QR}(\mathcal{N}(0, 1)^{N_{1o} + N_{20} \times 2r}) \quad \mathbf{Q}_{\text{full},R, \_} = \text{QR}(\mathcal{N}(0, 1)^{2r \times N_1}). \tag{161}$$

This basis $\mathbf{Q}_{\text{full}}$ is then split into two orthogonal subspaces for both the left and right singular vectors,

$$\mathbf{U}_1 = \mathbf{Q}_{\text{full,L}}[: N_{1o}, 1{:}r], \quad \mathbf{U}_2 = \mathbf{Q}_{\text{full,L}}[N_{1o} : N_{2o}, r{+}1{:}2r] \tag{162}$$

and similarly for $\mathbf{V}_1$ and $\mathbf{V}_2$ bu using $Q_{full,R}$. The eigenvalue vectors are defined as in the aligned case (159) to construct the input-output correlation matrices as

$$\mathbf{\Sigma}_{io} = \mathbf{U}_i \text{diag}(\lambda^{(i)}) \mathbf{V}^T_i \tag{163}$$

**Fully Conflicting Tasks** For the fully conflicting tasks, we construct two independent orthonormal bases for each task,

$$\mathbf{U}_{i, \_} = \text{QR}(\mathcal{N}(0, 1)^{N_{io} \times r}) \quad \mathbf{V}_{i, \_} = \text{QR}(\mathcal{N}(0, 1)^{2r \times N_1}) \tag{164}$$

and construct the input-output correlation matrices according to equation 163.

The specific configuration used to create Figure 3 in the main paper are presented in section E. Below we additionally plot the results for higher ranks and dimensions on Figures 8, 9, **??**.

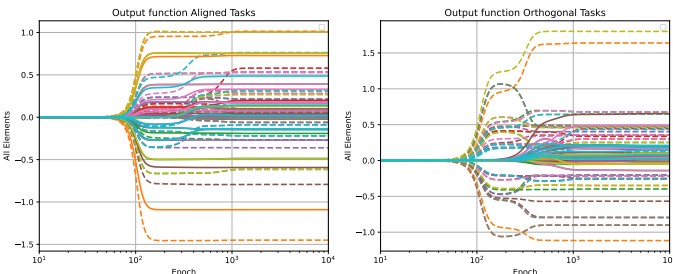

Figure 8: Analytical simulation of the output dynamics for perfectly aligned (left) and orthogonal (right) task pairs (d=7, r=3). In the aligned case, singular modes are learned simultaneously for both tasks (dotted vs. solid). In the orthogonal case, dotted trajectories depart from zero before any solid trajectory, indicating a learning lag between tasks.

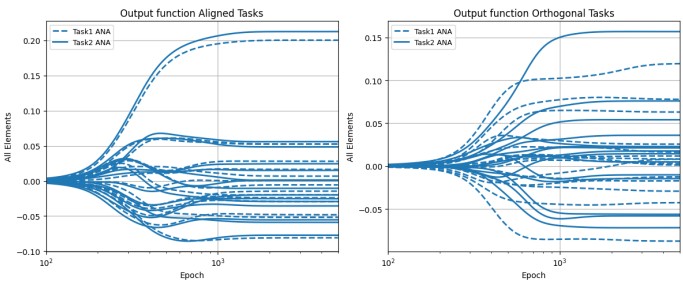

Figure 9: Analytical simulation of the output dynamoics for perfectly aligned (left) and orthogonal (right) task pairs (d=50, r=20); behaviour matches Fig. 8..

## D.2 TEACHER-STUDENT MTL REGRESSION TASK

For this setup, we generalise the teacher-student task in Braun et al. (2022) to an MTL setup. We sample the input $X \sim \mathcal{N}(0, \sigma_X)$ with size $N_D \times N_i$ and whiten it such that $\frac{1}{N_D} X X^T = \mathcal{I}$. To make sure we create tasks that are fully learnable with a linear neural network, we generate,

$$\mathbf{Y}_1 = \mathbf{W}_1(0)\mathbf{W}_S(0)\mathbf{X} \quad \mathbf{Y}_2 = \mathbf{W}_2(0)\mathbf{W}_S(0)\mathbf{X}, \tag{165}$$

where $\mathbf{W}_1 \sim \mathcal{N}(0, \sigma_{TS1})$ (size $N_1 \times N_s$), $\mathbf{W}_2 \sim \mathcal{N}(0, \sigma_{TS2})$ (size $N_2 \times N_s$) are randomly sampled from a gaussian distribution . Then $\mathbf{W}_S$ (size $N_s \times N_i$ is chosen such that the zero-balanced initialization condition is satisfied.

The tasks in equation (165) are learned by neural network with input dim $N_i$, shared dim $N_s$, and task-specific layers $N_1$, $N_2$ with uniform weighting. Note that the scale of the initial weights of the neural network, $\sigma_{NN}$ is not necessarily the same as $\sigma_{TS}$. The weights are also not necessarily the same. In the experiments, we play around with different $\sigma_{TSi}$ to vary the similarity between the tasks. To test the robustness of the analytical solution, we also change $\sigma_{NN}$, learning rate and $N_i, N_s, N_1, N_2$.

Experimental details for Figure 10 **Plot A** Simulation of analytical expression of the output functions of perfectly Aligned tasks (Appendix D) with input dimension d=4 and rank r=2. Eigenvalues are set to task 1=[4,1] and task 2=[3,0.5] resulting in $\mathbf{S}_\xi = [5, 1.118, 1.810^{-16}, 1.261.810^{-16}]$ which is approximately equal to $\sqrt{Tr(P\mathbf{S}_1) + Tr(P\mathbf{S}_2)}$. **Plot B** Simulation of analytical expression of the output functions of perfectly Orthogonal tasks (Appendix D) with input dimension d=4 and rank r=2. Eigenvalues are set to task 1=[4,1.5] and task 2=[1,0.5]. The singular value matrix of the combined task space is $\mathbf{S}_\xi = [4, 1.5, 1, 0.5]$ which is $\sqrt{Tr(P\mathbf{S}_1) \bigoplus Tr(P\mathbf{S}_2)}$. **Plots C-D-E-F** Comparison of the analytical expressions of the output functions and losses with the functions and losses of linear MTL network trained on Teacher-Student Regression tasks (Appendix D) with $N_1 = 10, N_{1o} = N_{2o} = 2, N_S = 5, \alpha = 0.01, \Sigma - TS = 0.025$ and zero balanced weights initialized with $\Sigma_W = 0.01$. To validate our results from section 5, we perform similar experiments to compare the RSMs of target values.

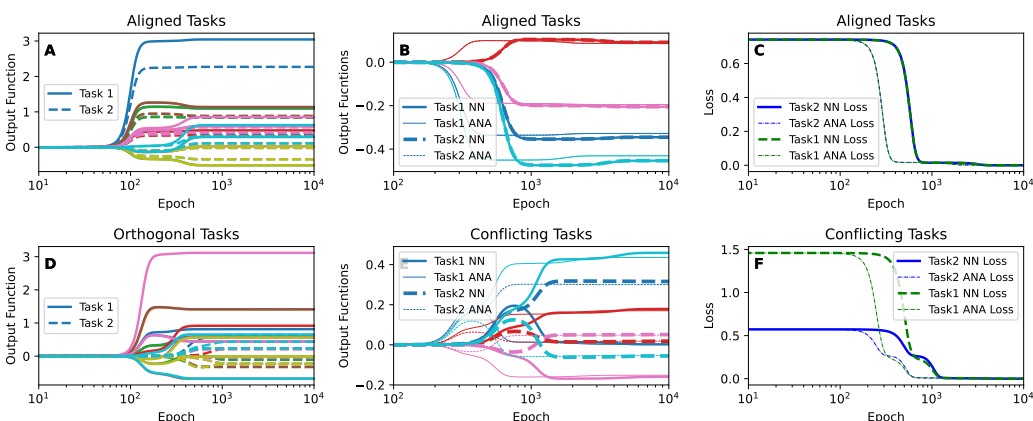

Figure 10: Plots (**A**) and (**D**) show the analytically derived output trajectories of two perfectly aligned and orthogonal handcrafted tasks. Plots (**B**)-(**E**) compare the analytical solution (ANA) to the empirical output dynamics of an MTL neural network (NN) trained on aligned (top row) and conflicting (bottom row) Teacher-Student Regression tasks. Plots (**B**) and (**E**) show the first four output dimensions; plots (**C**) and (**F**) show the corresponding loss trajectories. The MTL networks have $N_1$=10,$N_{1o}$=$N_{2o}$=2,$N_S$=5. All experiments use a learning rate ($\alpha = 0.01$) and zero-balanced weights. Colours correspond to distinct singular modes from $\mathbf{S}_\xi$.

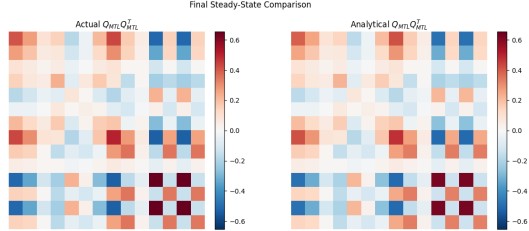

Figure 11: Comparison of $\mathbf{Q}\mathbf{Q}^T$ at convergence after training on Aligned Teacher Student tasks. Left plot: $\mathbf{Q}\mathbf{Q}^T$ obtained from two-layer linear MTL with $N_1 = 10$, $N_{1o} = N_{2o} = 2$ and $N_S = 5$.

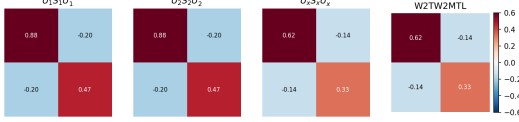

Figure 12: Comparison of RSMs of target values of aligned Teacher-Student regression tasks and the corresponding RSMs of internal layers obtained from a two-layer linear MTL with $N_1 = 10$, $N_{1o} = N_{2o} = 2$ and $N_S = 5$.

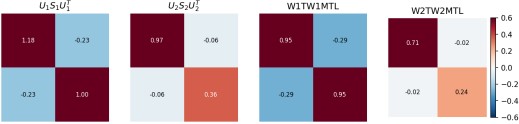

Figure 13: Comparison of RSMs of target values of conflicting Teacher-Student regression tasks and the corresponding RSMs of internal layers obtained from a two-layer linear MTL with $N_1 = 10$, $N_{1o} = N_{2o} = 2$ and $N_S = 5$.

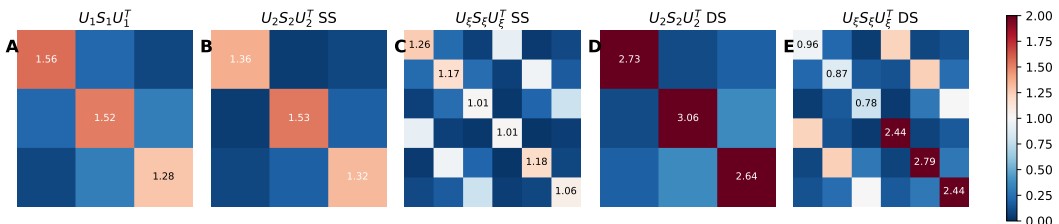

Figure 14: Representational similarity matrices (RSMs) of target values from random regression tasks defined by $\mathbf{Y}_i = \alpha_i \mathbf{B}_i$. Plots (**A**) and (**B**) show the RSMs of tasks 1 and 2, respectively, under equal scaling $\alpha_1 = \alpha_2$ (SS), resulting in the combined task structure shown in (**C**). In contrast, (**D**) shows the RSM of task 2 with a different scale (DS) and (**E**) presents the resulting combined task-structure RSM when task 1 remains identical to (**A**).

### D.3 MULTI-TASK RANDOM REGRESSION TASKS

In this setup, we generalize the Random Regression tasks in Braun et al. (2022) for STL to an MTL setup similar to Chen et al. (2018); Grégoire et al. (2024). We sample the input $X \sim \mathcal{N}(0, \sigma_X)$ with size $N_D \times N_i$ and whiten it such that $\frac{1}{N_D} X X^T = \mathcal{I}$. In this setup, the outputs of each task are also sampled according to,

$$\mathbf{Y}_i = \alpha_i[\mathbf{B} + \epsilon_i], \tag{166}$$

where $\alpha_i$ is a task-specific scalar determining the scale of the $\mathcal{T}_i$, $\mathbf{B} \sim \mathcal{N}(0, 1/\sqrt{N_{oi}})$ is a shared vector sampled from with a variance scaled to the number of $\mathcal{T}_i$ output nodes and $\epsilon_i$ is a task-specific vector $\mathcal{N}(0, \sigma_i)$. The (dis-)alignment between the tasks is fully controlled by $\epsilon_i$ and $\alpha_i$. Additionally, we will also study the effect of scale $\alpha_i$ differences between tasks.

**Impact of scale differences.** We use this setup to study the impact of scale differences on the representation in Figure 14. We present a comparison of the representational similarity matrices (RSMs) between the single-task target and input data with the combined-task space RSM for three scaling conditions: equal task scales (Figure 15), task 2 scaled by a factor of two (Figure 16), and task 2 scaled by a factor of one-tenth (Figure 17). In Figure 14, we compare the RSMs of target values from random regression tasks generated through $\mathbf{Y}_i = \alpha_i \mathbf{B}_i$ Figure 14 :Representational similarity matrices (RSMs) of target values from random regression tasks defined by $\mathbf{Y}_i = \alpha_i(\mathbf{B} + \epsilon_i)$ (Appendix D). The Regression tasks have dimensions defined with $N_1 = 20, N_{1o} = N_{2o} = 3$, $N_D = 100$. For the similar scale tasks $\alpha_1 = \alpha_2 = 1$ while for the different scale tasks= $\alpha_1 = \alpha_2 = 2, \sigma_W = 0.001$.

**Additional shared layers: impact on $W_i W_i^T$.** We perform experiments for the case where there are several hidden shared layers while having one task-specific layer. We perform the experiments with networks initialized with small Gaussian weights. In Figure **??**, we compare the final $\mathbf{W}_1 \mathbf{W}_1(t)$ of neural networks with different amounts of shared layers and one task-specific layer, each trained on the same Random Regression tasks. To this end, we use the last shared layer to obtain $\mathbf{W}_S \mathbf{W}_S^T$. In Figure 18, we show the difference in training losses. Adding more shared layers makes the convergence a bit slower, but the same optimum is reached as stated earlier

**The impact of scalar weighting** In Figure 19,we compare the shared representation of a linear MTL trained on Random Regression tasks with scale difference $\alpha_2 = 2 * \alpha_1$ trained with different task-specific weights $\omega_2$=1, 0.5, 0.05, 0.005.

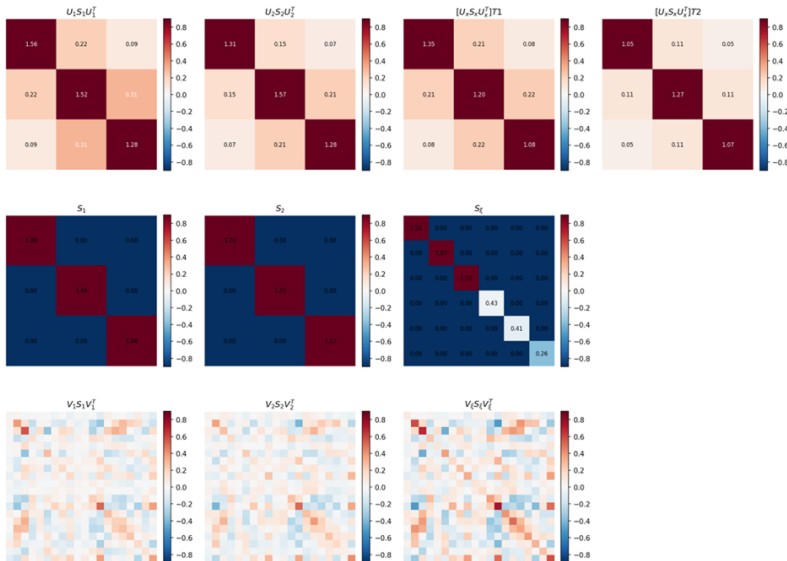

Figure 15: Comparison of RSMs of input and target data of Random Regression Tasks with $\alpha_1 = \alpha_2 = 1$ (no scale difference) RSMs of input and target combined task space $\xi$.

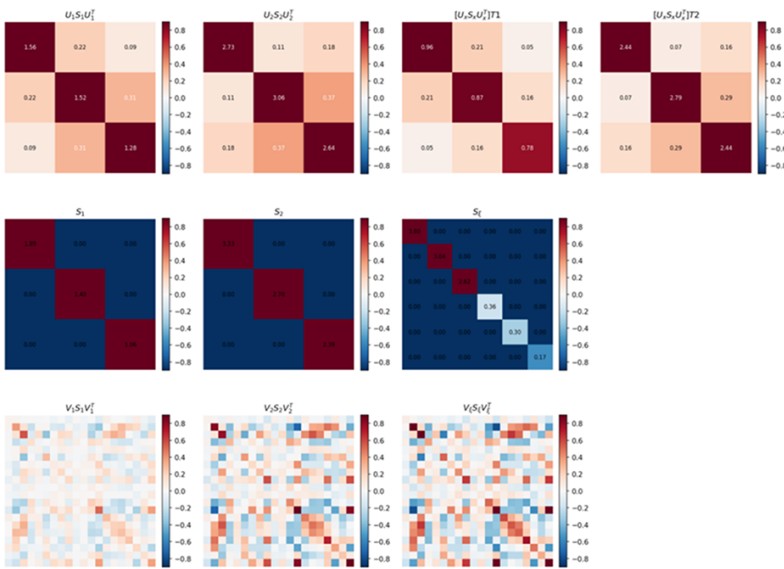

Figure 16: Comparison of RSMs of input and target data of Random Regression Tasks with $\alpha_2 = 2 * \alpha_1$ (scale difference) to RSMs of input and target combined task space $\xi$.

### D.4  $\gamma$-MTL TASKS

Generate an orthonormal basis $\mathbf{R}$ and choose $B_1$ as the first $N_1$ rows, $B_1^\perp$ as the remaining rows. Then we can construct two tasks like,

$$\mathbf{Y}_1 = \alpha_1 \mathbf{B}_1 \mathbf{X} \qquad (167)$$

and construct

$$\mathbf{Y}_1 = \alpha_2 [(\gamma * \mathbf{B}_1 + (1 - \gamma)\mathbf{B}_1^\perp]\mathbf{X}. \qquad (168)$$

As such $\gamma$ controls the alignment of the two tasks ($\gamma = 1$, perfect alignment, $\gamma = 0$, perfect orthogonality). While $\alpha_i$ only alters the relative magnitude between the tasks.

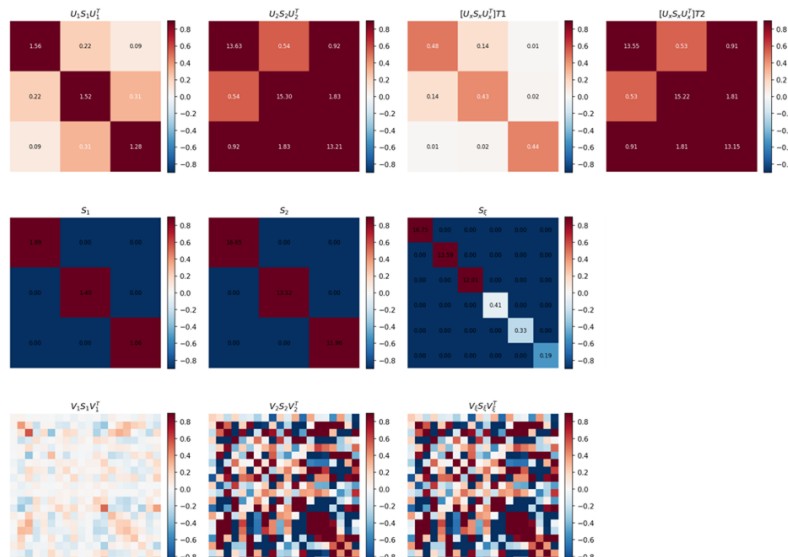

Figure 17: Comparison of RSMs of input and target data of Random Regression Tasks with $\alpha_2 = 10 * \alpha_1$ (scale difference) to RSMs of input and target combined task space $\xi$.

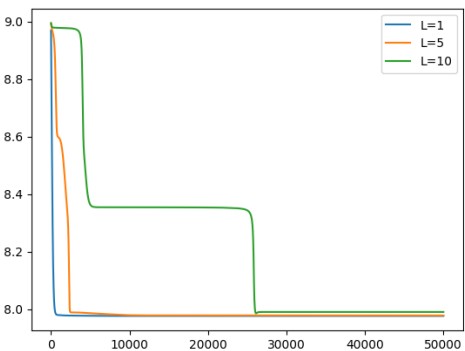

Figure 18: Comparison of training losses for linear MTL networks with 1,5 and 10 shared layers and 1 task-specific layer for each task. The hyperparameters: $N_1 = 10, N_S = 5, N_{1o} = N_{2o} = 2, \alpha = 0.01$, Initialised with small Gaussian Weights $\sigma_W = 0.001$.

### D.5 MULTI-MNIST

Multi-MNIST is a MTL version of the original MNIST dataset Lecun et al. (1998) introduced by Sabour et al. (2017) and commonly used for benchmarking in the MTL literature. The dataset is generated by overlaying digits from the MNIST dataset on top of other digits of the same set (training or test) but from another class. As the digits are shifted up to 4 pixels in each direction, the dataset contains 36x36 pixel images and is then flattened to 1296-dimensional input vectors, which are whitened later. We use this dataset in the MTL setup by defining the classification of the left and right digits on each image as different tasks. The data is openly available on `https://paperswithcode.com/paper/dynamic-routing-between-capsules` and openly licensed under CC-BY-SA. We perform experiments on different adaptations of Multi-MNIST where we alter how much the digits overlap and how much the representations are aligned. The shift parameter controls the overlap: shift=0 is perfect overlap, 1 almost perfect, 4, medium, 8 can be disjoint. Throughout this section we consider the prediction of the left-digit as Task 1 and the prediction of the right digit as Task 2.

In this section, we discuss the classic Multi-MNIST baseline and a permuted multi-MNIST version in order to compare the discussion in the main paper with the results from a real-world dataset.

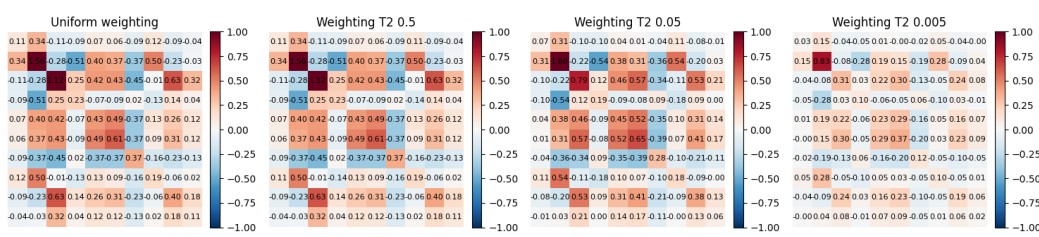

Figure 19: Comparison of final $\mathbf{W}_S^T \mathbf{W}_S$ of the same linear MTL with $N_1 = 10, N_{o1} = N_{o2} = 2, N_S = 100$ trained on 100 samples during 10000 epochs with different task-2-specific loss weights $\omega_2$. Scalar weighting impacts the influence of task 2 on the shared representation and might reduce the impact caused by imbalanced magnitudes.

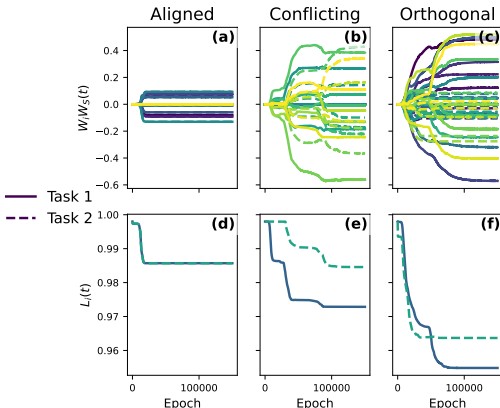

Figure 20: Similar experiment as Figure 4 but with one RELU-layer and no magnitude difference. $N_1 = 10, N_{o1} = 2 = N_{o2}, N_S = 4$. In this case, we find a similar behaviour as in the linear case in the paper: aligned tasks converge simultaneously, orthogonal tasks sequentially and conflicting tasks converge piecewise but more or less simultaneously.

### D.5.1 BASELINE MULTI-MNIST

In this setup we do not alter the overlap between the left and right digit and keep the configuration is the one introduced by Sabour et al. (2017).

First of all, note that for this dataset, the singular values of the overlap matrix are given by: array([1. , 0.9911755 , 0.98780644, 0.9826204 , 0.98043734, 0.968582 , 0.9650397 , 0.95887566, 0.93100125, 0.9173822 ]) meaning that the tasks are partially aligned ($\alpha_1 = 1$) and mostly conflicting $0 < \alpha_i < 1$) but with a high agree of alignment ($\alpha_i \approx 1$). From the analysis in the main paper, we would expect the tasks to converge more or less simultaneously. This is also confirmed by the experiments presented on Figure 22a where we compare the task-specific losses obtained from training a linear MTL network and a non-linear MTL (ReLU) network.

Concerning the RSMs, Figure 23 shows a similar analysis to Figure 5 to compare the RSM's of the input and output data for the single tasks (i.e., only the left or right digit estimation) and find that our analyses generalize well to the real-world dataset. The combined RSMs of the input data yield weaker structural relationships and the combined data RSMs of the input data create new structural relationships. On Figure 24 compare these predictions to the actual neural representations after training a neural network with $N_S = 20$ during 25000 epochs. We see that the neural network captures weaker structural relationships for the output while finding new relationships (although weak)

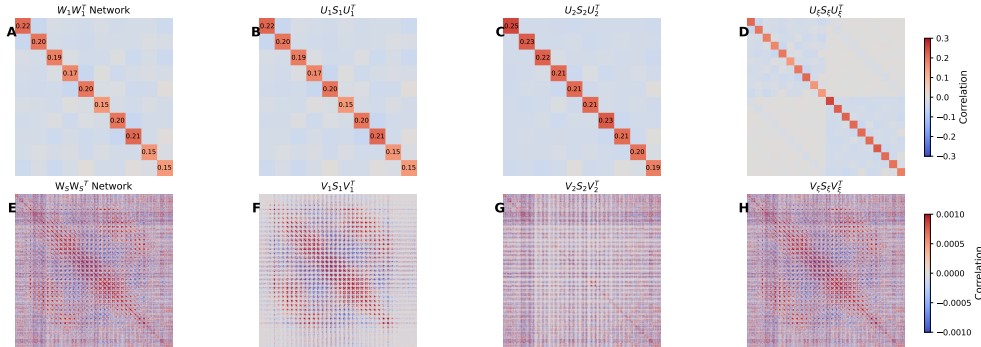

Figure 21: Representational similarity matrices (RSMs) of tasks target values (top) and inputs (bottom) compared to the RSM of the neural representations of a linear MTL NN trained on a batch (N=20000) of the permuted Multi-MNIST dataset ($\alpha = 0.01$, $N_S$=20) after convergence. Top row: (**A**) RSM$_{1o}$ from the network at convergence compared to the RSMs of the target values of (**B**) task 1,(**C**) task 2 and (**D**) the combined task space $\xi$. Bottom row: RSM$_I$ of the shared hidden representation of the network (**E**) compared to the RSMs of the input of (**F**) task 1, (**G**) task 2 and (**H**) the combined task space $\xi$.

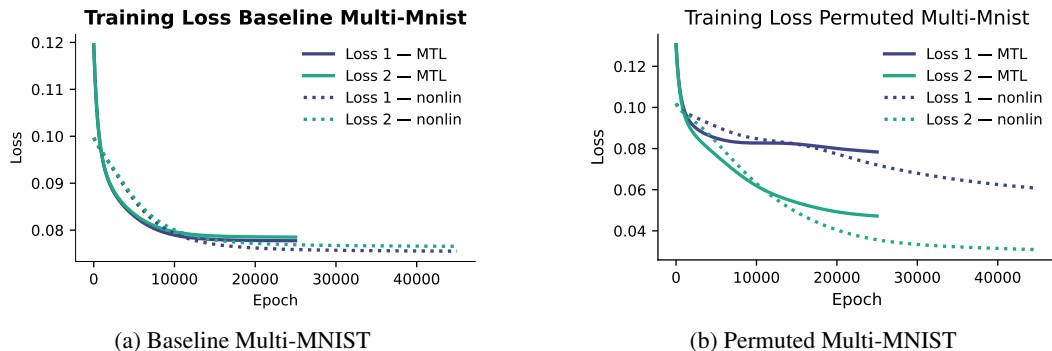

(a) Baseline Multi-MNIST        (b) Permuted Multi-MNIST

Figure 22: Dynamics of task-specific losses for linear (solid) and non-linear (dotted) MTL with $N_S = 20$ on 50K samples. (a) Baseline Multi-MNIST. (b) Permuted Multi-MNIST.

for input data. Nevertheless, the expected values do not exactly match the theoretical expectations, since the tasks did not probably converge.

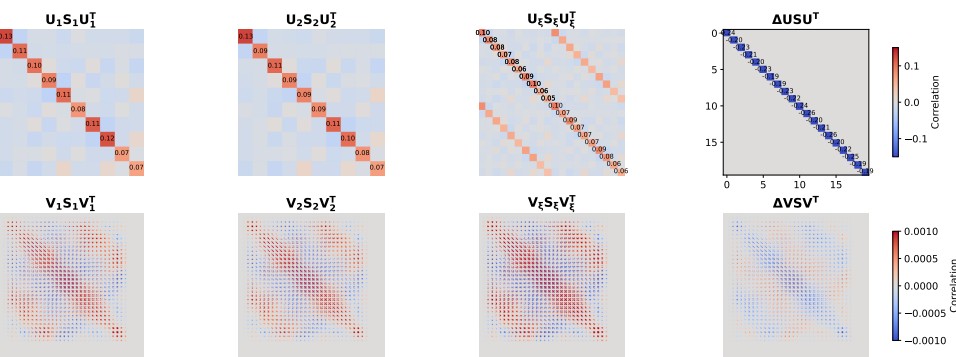

Figure 23: Baseline Multi-MNIST. RSM's of the input and output data are obtained from a batch of 50K samples. We see that compared to the single task RSMs, the combined RSM of the ouptut data yields weaker structural relationships, as expected for conflicting tasks.

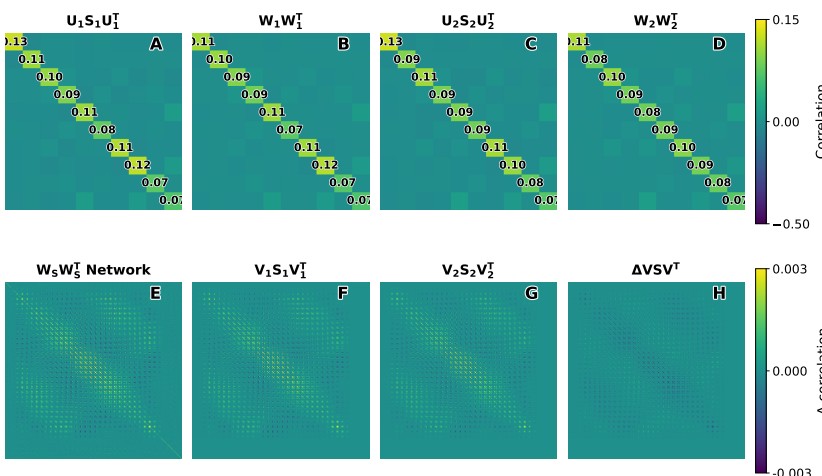

Figure 24: Comparison of the data RSMS with the observed neural representation after training a neural network with $N_S = 20$ on 50K examples of Baseline-Multi-MNIST for 25K epochs, with learning rate=0.01.

### D.5.2 Permuted MNIST

Task 1: left digit estimation. Task 2 random noise In order to compare the results obtained for the baseline multi-MNIST dataset with a similar dataset with different alignment between the tasks, we leave Task 1 unaltered but replace task 2 (right digit estimation) by a noisy task with the same dimensions. The corresponding eigenvalues of the overlap matrix are now equal to [0.9999999 , 0.37486207, 0.36476606, 0.33212408, 0.31399104,0.29427034, 0.28034934, 0.2724696 , 0.253729 , 0.04012636] indicating that the task has aligned ($\alpha_1 \approx 1$ and orthogonal parts ($\alpha_{10} \approx 0$) but is mostly conflicting ($0 < \alpha_i < 1$). As a result, we expect a bigger learning lag between the tasks as compared to the case with Baseline Multi-MNIST. This is also what we observe on Figure 22b for the case where the tasks are trained in a linear and a non-linear (ReLU) network. Although the tasks did not fully converge on the plot, we still see an initial simultaneous decrease related to the shared mode, after which task 1 stops converging and start again at a later point when task 2 is already closer to actual convergence.

We also extend the analysis of the RSMs to the case of permuted Mnist, where we expect the interference to become much weaker in the combined RSMs of both the input and output RSMs as compared to the case for the Baseline Multi-MNIST dataset. This is also what we observe when comparing Figure 23 with Figure 25. Finally, we compare the neural representations at convergence with the theoretical expectations in Figure 26.

### D.6 Multi-Task Networks

For all experiments described in the previous subsections, we adopted a hard-parameter sharing architecture Ruder (2017) as schematically represented in Figure 1 of the main text for one shared and one task-specific output layer for each task. Unless otherwise stated, we use this architecture (one shared and one task-specific linear layer) for all experiments. The specific hyperparameters (input, hidden, and task-specific dimensions, learning rates, initialization scale) are mentioned on each Figure and the corresponding text describing the experiment. The neural network and its optimization were implemented with PyTorch.

## E Figures

In this section we provide the experimental details concerning the Figures appearing in the main paper.

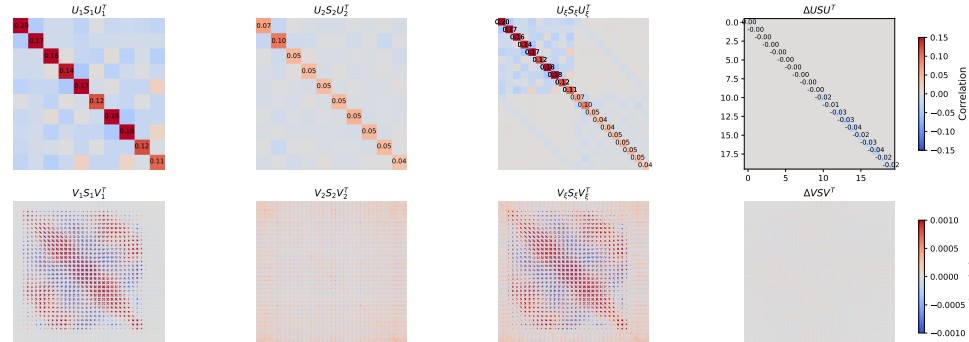

Figure 25: Permuted Multi-MNIST. RSMs of the input and output data are obtained from a batch of 50K samples. We see that compared to the single task RSMs, the combined RSM of the output data yields structural relationships which are very similar to the task-specific RSMS as expected for conflicting tasks (which are close to orthogonal).

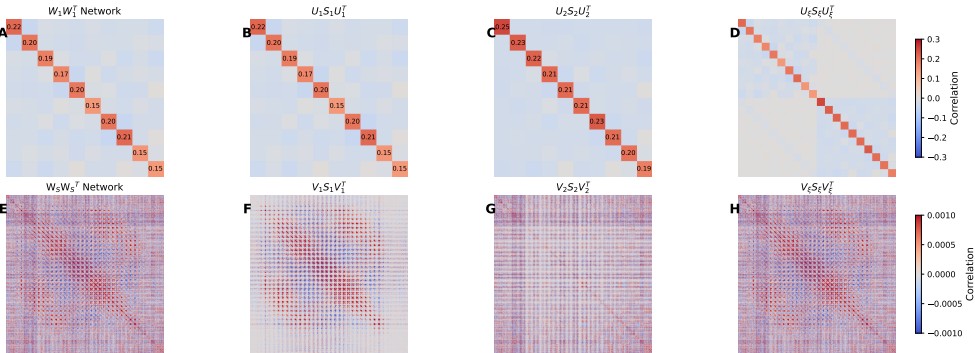

Figure 26: Representational similarity matrices (RSMs) of tasks target values (top) and inputs (bottom) compared to the RSM of the neural representations of a linear MTL NN trained on a batch (N=20000) of the Multi-MNIST dataset ($\alpha = 0.01$, $N_S$=20) after convergence. Top row: (**A**) RSM$_{1o}$ from the network at convergence compared to the RSMs of the target values of (**B**) task 1,(**C**) task 2 and (**D**) the combined task space $\xi$. Bottom row: RSM$_I$ of the shared hidden representation of the network (**E**) compared to the RSMs of the input of (**F**) task 1, (**G**) task 2 and (**H**) the combined task space $\xi$.

**Figure 1.** Representation of a two-layer linear multi-task network created with Figma software. Note that the we provide a notebook *Figures_ITD_MTL.ipynb* where all the code used to create the Figures is included.

**Figure 2.** Schematic example of how the singular value decompositions of two tasks are combined into the "total MTL" space $\xi$ in a non-trivial way, creating new modes which are related to the original tasks in complex ways. Figure created with Figma software.

**Figure 3.** Analytical simulations of the task-specific network functions $\mathbf{W}_i\mathbf{W}_S$ based on Theorem 2 of a two-layer linear network with $N_1 = 4, N_{io} = 2, N_S = 4$ and $\alpha = 10^{-3}$ trained on Fully Aligned (**a**)-(**d**), Conflicting (**b**)-(**e**) and Orthogonal tasks (**c**)-(**d**) created by generating orthonormal basis to create $V_1^T V_2$ basis with different amounts of alignment as explained in Appendix D.1. For the imbalanced examples (top row Figure 3) we used the same singular values for plots (**a**)-(**b**)-(**c**) such that only their alignment differs. For task 1 the singular values $\mathbf{S}_1 = diag([4,3])$ and $\mathbf{S}_2 = diag([2, 1.5])$. For perfectly aligned tasks depicted on plot (**A**) this resulted in $\mathbf{S}_\xi = diag([4.472, 3.354])$, i.e., Rank($S_\xi$) = 2. For the conflicting and orthogonal tasks Rank($S_\xi$) = 4. The eigenvalues of the overlap matrix were equal to [1, 1] for the fully aligned tasks plot (**A**), [0.923 0.277] for the fully conflicting tasks (plot **B**), and [1.90e-16 6.29e-17] for the orthogonal tasks. For the bottom row, we used the same singular values for task 1 and 2 but normalized the resulting

correlation matrices to correct for magnitude imbalances and capture only the effect of alignment. Similar analytical simulations are found in Appendix D.1.

**Figure 4.** Comparison of the evolution of the task-specific losses resulting from training a two-layer linear MTL network with $\Delta M_{12} = 4$. $N_1 = 20$, $N_{o1} = N_{o2} = 1$, $N_S = 2$ on Aligned (**a**)-(**d**), Conflicting (**b**)-(**e**) and Orthogonal Regression tasks (**c**)-(**d**) with a fixed difference in magnitude between for all task pairs $\Delta M_{12}^{ij} = 1/4$. The tasks are generated by an orthonormal basis $\mathbf{R}$ and choose $B_1$ as the first $N_1$ rows, $B_1^\perp$ as the remaining rows. As explained in Appendix D.4, the parameter $\gamma$ determines the amount of alignment between the tasks while $\alpha_i$ determines the magnitude (im)balance. In Figure 4, we set $\gamma = 1$ for the aligned tasks, $\gamma = 0.5$ for the conflicting tasks and $\gamma = 0$ for the orthogonal tasks. The scale difference is set by $\alpha_1 = 1$ and $\alpha_2 = 4$. $L_2$ values are rescaled after training ($\div 16$) to make the comparison between the evolution of the curves more easy, this however, does not change anything for the corresponding analysis.

**Figure 5.** Comparison of RSMs for regression task pairs ($N_1 = 4$, $N_o1 = N_o2$) with different alignment, generated by constructing by an orthonormal basis $\mathbf{R}$ and choosing $B_1$ as the first $N_1$ rows, $B_1^\perp$ as the remaining rows. As explained in Appendix D.4, the parameter $\gamma$ determines the amount of alignment between the tasks while $\alpha_i$ determines the magnitude (im)balance. For the balanced tasks Figure 5(**A**) and 5(**C**), we set $\alpha_1 = \alpha_2 = 1$ while we change $\alpha_2 = 4$ for the plot in Figure 5(**B**).

**LLM Usage.** LLMs were employed exclusively to refine the clarity of the writing and to enhance the code for improved visual presentation; the underlying data and analyses remain entirely unaffected.

