# OpenReview forum: "Inter-Task Learning Dynamics in Deep Linear Multi-Task Networks"
_ICLR.cc/2026/Conference — Submitted to ICLR 2026_

### Official Review · Reviewer_15xH · 2025-10-28

**Soundness:** 3
**Presentation:** 2
**Contribution:** 2
**Rating:** 6
**Confidence:** 3

**Summary:**

This work presents a theoretical analysis of learning dynamics in multi-task learning (MTL), focusing on the common hard parameter sharing setting.
The approach of the authors follows a series of works in the single-task setup, where two-layer deep linear networks are analyzed through the use of the Riccati formulation of gradient flow. These results are adapted and employed in the MTL setting.
A detailed analysis in the cases of aligned, conflicting and orthogonal tasks is presented, somewhat mirroring common intuitions in the MTL literature.
The appendix compares these theoretical results with a small dataset commonly employed in the MTL literature.

**Strengths:**

The MTL setting is still far from being understood, with a series of works in the area drawing conflicting conclusions for instance concerning the utility of multi-task optimizers.
This work is definitely a step towards providing a better understanding of MTL learning dynamics, and will hopefully pave the way for practical advances.
Most of the results also appear to be relatively intuitive, and in line with common intuitions in the area.
I appreciate the effort by the authors to provide intuition behind their results, and the extensive appendices.

**Weaknesses:**

- While the work is definitely interesting and novel, its methodology heavily builds on previous work on STL. This is however appropriately acknowledged by the authors.
- The employed assumptions appear to be extremely strict, and of course do not apply to deep MTL.
- Not a lot of emphasis is placed on real-world data. Multi-MNIST results are delegated to a fairly long appendix, however it would be quite important to more concisely and directly discuss the relationship between the theoretical results and real-world data in the main body of the paper.
- Some of the authors' conclusions are not immediately apparent from the figures. For instance, in Figure 3, it is honestly quite hard to discern any difference between the orthogonal and the conflicting tasks setups. This is fairly surprising, though. I am even more surprised by the fact, when considering the loss function in Figure 4, orthogonal tasks seem to make for a harder learning problem.
- As far as I understand, the provided theory will not easily take into account the effect of network capacity, which however appears to be a very important factor in MTL, especially considering whether any task orthogonality will effectively hinder the learning process.

**Questions:**

- How can network capacity be factored in the provided theory?
- The practice of Multi-MNIST appears to more or less mirror the provided theory. Would this happen on larger-scale MTL datasets too?
- Could you please share some more intuition as to how your results could influence the practical design of MTL techniques?

---

> ### Author Response · Authors · 2025-11-21
> **Response to the Reviewer’s Concerns**
>
> We thank the reviewer for the valuable feedback and positive assessment of our contributions. We address each concern below and respond to the questions in a separate comment. **In the revised manuscript, we moved the Multi-MNIST analysis into the main text and clarified the observations in Figure 3, as noted in the general comment (M1–M4)**.
>
> > W1.  While the work is definitely interesting and novel, its methodology heavily builds on previous work on STL. This is however appropriately acknowledged by the authors.
>
> We clarify that our theoretical contribution does not lie in the lemmas or equations inherited from Braun et al., **but in extending their framework to the MTL setting and deriving new structural relationships between task alignment and task interactions**. This provides a new perspective for understanding interaction dynamics in multi-task neural networks.
>
> **Concerning the mathematical machinery, the novelty consists of showing that the solutions of the Riccati equation presented by Braun et. al (2022) still hold** when drawing an explicit difference between task-specific output layers; to this end, we had to rederive the second assumption (the balancedness assumption) to see under which assumptions this solution could be extended to this setup. We saw that it is enough to change the STL version of the zero-balancedness assumption to:
> $W_S^TW_S=W_1W_1^T+W_2W_2^T$. Additionally, we introduced $\mathbf{\xi}$  and $W_{MTL}^0$ and their SVD, which allowed for the direct analysis of the impact of task alignment on the resulting dynamics where the combined spectrum plays a central role. Finally, we extended the convergence theorem to the case with multiple tasks, which allows for a comparison between learned representations obtained in MTL and STL without any training. Additionally, we derive solutions for Tabula Rasa initialisations in Appendix B3 and perform a theoretical analysis.
>
> > W2. The employed assumptions appear to be extremely strict, and of course do not apply to deep MTL.
>
> We would like to emphasize that these are **standard assumptions in the theoretical deep-learning literature** and are not central to our contribution. Next, we would like to highlight that **we showcase additional experiments in the Appendix on non-linear and deeper networks and discuss how this affects our predictions.** Furthermore, several of these assumptions (zero-balancedness, no-bottleneck) have recently been relaxed in the single-task setting. **The resulting insights indicate that relaxing them mainly leads to phenomena associated with transitions between the lazy and rich regimes, a well-known property of neural network training that is not specific to MTL.** There is no reason to expect that MTL networks would behave differently under the same conditions, and therefore, such relaxations would not produce qualitatively new insights for this contribution.  Recent work [1] has extended the Riccati framework to more general initialization schemes. In particular, the zero-balanced initialization assumption has been generalized to a $\lambda$-balanced setting, enabling smooth interpolation between the rich and lazy regimes and analyses the impact of architecture on the transition between these regimes.
>
> > W3. Not a lot of emphasis is placed on real-world data. Multi-MNIST results are delegated to a fairly long appendix, however it would be quite important to more concisely and directly discuss the relationship between the theoretical results and real-world data in the main body of the paper.
>
> **We agree this should be highlighted more and decided to add this to the main body of the paper (M1).**
>
> > W4. Some of the authors' conclusions are not immediately apparent from the figures. For instance, in Figure 3, it is honestly quite hard to discern any difference between the orthogonal and the conflicting tasks setups.
>
> When comparing Figures 3(b) and 3(c), the key difference is in how the curves evolve over time. For orthogonal tasks, the singular modes of Task 2 project only weakly onto the shared space, so the solid curves (Task 1) converge well before Task 2. This shows that learning one task can leave the other largely unaffected for a substantial period. With an additional magnitude imbalance, this effect becomes even more pronounced.
>
> When the magnitude imbalance is removed (Figure 3f), the network functions of the two tasks remain disentangled, but the time lag disappears because both tasks now possess singular modes of comparable strength. Although the resulting trajectories may superficially resemble those of conflicting tasks, the underlying mechanism is fundamentally different, as is also evident from the learned representations in Figure 5.
>
> > W5. As far as I understand, the provided theory will not easily take into account the effect of network capacity, which however appears to be a very important factor in MTL.
>
> We clarify this point in the comment concerning the reviewer's questions.

---

> ### Author Response · Authors · 2025-11-21
> **Response to the Reviewer's questions**
>
> We thank the reviewer again for the valuable feedback. Below, we address each question individually and respond to the concerns in a separate comment.
>
> > Q1: How can network capacity be factored in the provided theory?
>
> To better understand the impact of architecture, we refer to [1], where the zero-balanced assumption is relaxed to
> $\lambda$-balanced initializations and the architectural effects under this setup are analysed (see their Fig. 5). They show that different $\lambda$ values yield different learning regimes of the network, and enable a transition between the rich and lazy regimes. Furthermore, their analysis shows that the architecture directly affects how the sign of the relative scale (determined by the $\lambda$-value) is related to the different learning regimes (i.e., depending on the architecture, different $\lambda$ values will put the network in the lazy or rich regime). While this is important for understanding single-task learning behaviour, placing the MTL model in the lazy regime (which happens when we move away from the zero-balancedness) would mainly suppress feature learning rather than illuminate MTL-specific effects. Furthermore, a recent workshop paper [2] investigated the consequences of moving beyond the bottleneck assumption (A3). They found that the advantages of removing this assumption are, in fact, quite limited; what plays a much more decisive role is the effective rank of the model, which directly governs its expressivity.
>
> > Q2: The practice of Multi-MNIST appears to more or less mirror the provided theory. Would this happen on larger-scale MTL datasets too?
>
> The theory does not impose any restriction on the dimensionality or scale of the dataset, so there is a priori no reason to believe that our predictions would fail to extend to larger-scale MTL settings. The underlying dynamics apply equally in higher-dimensional regimes. However, such datasets make the resulting analyses more challenging to interpret visually, which is why we use smaller-scale examples for clarity. The theoretical conclusions themselves remain unchanged.
>
> > Q3: Could you please share some more intuition as to how your results could influence the practical design of MTL techniques?
>
> **We added a subsection to the appendix where we address this point in detail (M3) and added a plot with the cosine similarities of gradients with respect to shared and task-specific layers (M2) to make the connection with practical MTL techniques more clear**.
> There are two main practical implications from our results.
>
> **Concerning task grouping**, we show that there must be some signal in the data that could guide task grouping algorithms: since the degree of alignment between tasks implies how tasks interact, it is likely that this will affect performance as well. As such, **our results open a new data-driven perspective, one that moves away from computationally expensive techniques that require repeatedly training full MTL models**. Based on the current understanding that negative cosine similarities of the gradients are associated with negative transfer, together with our empirical finding that aligned tasks maintain strongly positive cosine values (M2), a natural future direction is to estimate task alignment and group tasks accordingly.
>
> **With respect to dynamic weighting algorithms**: **Our observations imply that the choice or design of a dynamic weighting algorithm should be task- and alignment-dependent rather than one-size-fits-all**. Namely, Aligned tasks tend to learn at similar rates and produce far fewer conflicting gradients (as shown in our additional plots, M2), while conflicting and orthogonal tasks exhibit highly dynamic task importance, and their gradients show more negative cosine similarities, which is commonly interpreted as negative interference. This suggests that such tasks require dynamic weighting, and that more sophisticated dynamic-weighting algorithms are likely to provide greater benefits, relative to simple uniform weighting, for conflicting and orthogonal task pairs than for aligned ones.
>
> While determining the exact form that such strategies should take lies beyond the scope of this paper, we believe that our theoretical framework provides a principled basis that can inform and inspire future work on more effective and efficient MTL algorithms.
>
> [1] Dominé et al. (2025). From Lazy to Rich: Exact Learning Dynamics in Deep Linear Networks.
> [2] Dominé et al. (2025). https://openreview.net/forum?id=GtXz0JfWR1.

---

### Official Review · Reviewer_HJRd · 2025-10-30

**Soundness:** 3
**Presentation:** 3
**Contribution:** 2
**Rating:** 2
**Confidence:** 4

**Summary:**

This paper extends the Riccati formulation of gradient flow from single-task to multi-task learning in deep linear networks. The authors characterize how task alignment (measured through SVD overlap) and magnitude differences affect learning dynamics, showing that these factors determine whether tasks learn simultaneously, sequentially, or in a conflicting manner. The main theoretical contribution is an analytical solution for the evolution of shared and task-specific components in linear MTL networks, which provides a motivation for dynamic loss weighting schemes.

**Strengths:**

**Practical motivation**: The work addresses a real gap - the lack of theoretical understanding of task interactions in MTL. The connection to dynamic loss weighting is a valuable insight, even if not fully developed.

**Clear presentation**: The paper is well-written with effective visualizations. Figure 2 and 3 nicely illustrate the key phenomena of how task alignment affects learning dynamics.

**Systematic analysis**: The categorization of task relationships (aligned/orthogonal/conflicting) and their impact on learning provides a useful framework for thinking about MTL.

**Weaknesses:**

**Limited theoretical novelty**: The core contribution is essentially applying Braun et al. (2022)'s framework to a multi-task setting. The mathematical machinery remains unchanged - just the dimensions and notation are extended. The proofs follow the same structure with minimal technical innovation.

**Restrictive MSE-only assumption**: The paper assumes all tasks use MSE loss, which severely limits its applicability. Real-world MTL typically involves heterogeneous tasks - classification (cross-entropy), regression (MSE), ranking, etc. For example, in computer vision, depth estimation (MSE) often trains jointly with semantic segmentation (CE). The interaction between different loss types could fundamentally change the dynamics, yet the current framework cannot handle this. This makes the theoretical insights less relevant to practical MTL systems.

**Lack of algorithmic contributions**: Despite providing theoretical motivation for dynamic loss weighting, the paper fails to propose any concrete adaptive weighting algorithms based on their analysis. This is a significant missed opportunity, the theoretical insights about task alignment and learning dynamics could directly inform the design of new weighting strategies.  Without demonstrating how their theory translates into practical algorithms, the paper remains purely descriptive rather than prescriptive.

**Shallow insights**: While the paper describes *what* happens under different task alignments, it doesn't explain *why* these phenomena occur at a deeper level. For instance, why exactly do conflicting tasks create non-monotonic learning curves? The analysis stays at the level of observing consequences of the analytical solutions rather than providing fundamental understanding.

**Experimental limitations**:
- All experiments use toy problems with small dimensions
- No comparison with existing dynamic weighting methods (GradNorm, PCGrad, etc.)

**Gap between theory and practice**: The results are confined to linear networks with restrictive assumptions (whitened inputs, zero-balanced weights, MSE loss only). The paper doesn't bridge this gap or provide actionable insights for real MTL systems. How do these results guide the design of actual MTL algorithms?

**Missing key analyses**:
- No discussion of which task alignments are optimal for generalization
- No theoretical analysis of sample complexity in MTL vs STL
- The convergence analysis (Theorem 3) is just stating what the solution converges to, without rates or conditions

**Questions:**

1. **Can you provide learning dynamics for realistic problem sizes?** The current examples are too small to be convincing. What happens with, say, 100-dimensional inputs and 10 tasks?

2. **How do your theoretical predictions compare with actual dynamic weighting algorithms?** It would strengthen the paper to show that your theory explains when methods like GradNorm succeed or fail.

3. **What happens when assumptions are violated?** Real data isn't whitened, and weights aren't zero-balanced. How robust are your conclusions?

4. **Can you derive optimal task groupings from your framework?** Given task statistics, can you predict which tasks should be trained together vs separately?

5. **The "conflicting" case ($0 < \alpha_i< 1$) seems to cover a huge range of scenarios.** Can you provide more granular analysis? When does partial alignment help vs hurt?

6. **How does the number of tasks NT scale in your analysis?** All examples show $N_T=2$. Does the framework become intractable or reveal new phenomena for many tasks?

7. **What's the computational cost of computing these theoretical predictions?** Could they be used in practice to guide training?

---

> ### Author Response · Authors · 2025-11-20
> **Response to the Reviewer’s Concerns**
>
> We thank the reviewer for the valuable feedback. We address each concern below and address the questions in a separate comment. **To clarify the practical relevance of our work, we have made several modifications to the main manuscript (M1-M4), as outlined in the general comment**.  At the same time, we would like to emphasize that the primary goal of this paper is to provide fundamental theoretical insights rather than a ready-to-use algorithm. We kindly ask the reviewer to take these additions and clarifications into account when assessing the paper, as we believe the presented insights form a solid theoretical foundation that can meaningfully pave the way for future practical progress in multi-task learning (as also pointed out by reviewer 15xH).
>
> **Concerning the Theoretical Novelty**
>
> We clarify that our theoretical contribution does not lie in the lemmas or equations inherited from Braun et al., **but in extending their framework to the MTL setting and deriving new structural relationships between task alignment and task interactions**. This provides a new perspective for understanding interaction dynamics in multi-task neural networks.
>
> **Concerning the mathematical machinery, the novelty consists of showing that the solutions of the Riccati equation presented by Braun et. al (2022) still hold** when drawing an explicit difference between task-specific output layers; to this end, we had to rederive the second assumption (the balancedness assumption) to see under which assumptions this solution could be extended to this setup. We saw that it is enough to change the STL version of the zero-balancedness assumption to:
> $W_S^TW_S=W_1W_1^T+W_2W_2^T$. Additionally, we introduced $\mathbf{\xi}$  and $W_{MTL}^0$ and their SVD, which allowed for the direct analysis of the impact of task alignment on the resulting dynamics where the combined spectrum plays a central role. Finally, we extended the convergence theorem to the case with multiple tasks, which allows for a comparison between learned representations obtained in MTL and STL without any training. Additionally, we derive solutions for Tabula Rasa initialisations in Appendix B3 and perform a theoretical analysis.
>
> **Concerning the Assumptions**
>
> We would like to emphasize that these are **standard assumptions in the theoretical deep-learning literature** and are not central to our contribution. Next, we would like to highlight that **we showcase additional experiments in the Appendix on non-linear and deeper networks and discuss how this affects our predictions.** Furthermore, several of these assumptions (zero-balancedness, no-bottleneck) have recently been relaxed in the single-task setting. The resulting insights indicate that relaxing them mainly leads to phenomena associated with transitions between the lazy and rich regimes, a well-known property of neural network training that is not specific to MTL. There is no reason to expect that MTL networks would behave differently under the same conditions, and therefore, such relaxations would not produce qualitatively new insights for this contribution.  For example, recent work [1] has extended the Riccati framework to more general initialization schemes. In particular, the zero-balanced initialization assumption has been generalized to a $\lambda$-balanced setting, enabling smooth interpolation between the rich and lazy regimes and shifting training dynamics from sigmoidal to exponential and shows the impact of architecture on the transition between these regimes.
>
> **Concerning the Experimental Limitations**
>
> We kindly note that **we already present a detailed analysis of the real-world Multi-MNIST dataset in Appendix D.5**, where we compare empirical learning dynamics with our theoretical predictions. We moved these results to the main body of the paper in the revised manuscript (M1). Across networks of different depths, and even with nonlinearities, we observe a high degree of consistency with our predicted convergence behaviour and training trajectories. We agree that this result deserves greater prominence, and we will move it into the main text using the additional page available in the camera-ready version.
>
> Since our contribution is theoretical and we do not introduce a new optimization method, we did not view benchmarking against existing algorithms as necessary. **To strengthen the connection to practical methods, we now provide cosine-similarity and gradient-magnitude similarity plots for the aligned, conflicting, and orthogonal settings used in our experiments (M2)**. These measurements show clear distinctions in gradient behaviour across these regimes and further support the practical relevance of our framework. We also added these plots to the paper.
>
> [1] Dominé et al (2025). From Lazy to Rich: Exact Learning Dynamics in Deep Linear Networks

---

> ### Author Response · Authors · 2025-11-20
> **Response to Reviewer's questions**
>
> We thank the reviewer again for the valuable feedback. Below, we address each question individually and respond to the concerns in a separate comment.
>
> > Can you provide learning dynamics for realistic problem sizes? The current examples are too small to be convincing. What happens with, say, 100-dimensional inputs and 10 tasks?
>
> and
> > How does the number of tasks NT scale in your analysis? All examples show 2. Does the framework become intractable or reveal new phenomena for many tasks?
>
> The Riccati solution and the resulting insights do not impose any restriction on dimensionality of the input. The figures we present and the corresponding analyses use low-dimensional examples purely for ease of visualization and interpretation. However, the underlying dynamics apply equally in higher-dimensional settings.  We further support this point with additional experiments in the appendix and the analysis on the Multi-MNIST dataset (M1).
>
> Regarding the extension to more tasks, **we show in the appendix that the solution naturally generalizes beyond the two-task case**. Since our analysis focuses on pairwise interactions, introducing more tasks simply amounts to applying the same pairwise analysis to each task pair.
>
> >How do your theoretical predictions compare with actual dynamic weighting algorithms? It would strengthen the paper to show that your theory explains when methods like GradNorm succeed or fail
>
> This is an interesting question. To examine the link between task alignment and common interaction measures (e.g., gradient cosine similarity and norms), we plotted their evolution for aligned, conflicting, and orthogonal tasks (added to Figure 4 in the manuscript, M2). The Figure shows that: perfectly aligned tasks show consistently high cosine values, reflecting strong gradient similarity and essentially no negative transfer. In contrast, conflicting and orthogonal tasks exhibit alternating positive and negative cosine values, indicating periods of negative transfer under current definitions. This indicates that the benefit of dynamic weighting algorithms such as PCGrad or CAGrad might depend on the degree of alignment present in the data.
>
> > What happens when assumptions are violated? Real data isn't whitened, and weights aren't zero-balanced. How robust are your conclusions?
>
> Dominé et al (2025),  extended the Riccati framework to more general initialization schemes. In particular, the zero-balanced initialization assumption has been generalized to a $\lambda$-balanced setting, enabling a transition between the rich and lazy regimes. While this is important for understanding single-task learning behaviour, **placing the MTL model in the lazy regime would mainly suppress feature learning rather than illuminate MTL-specific effects.**
>
> Concerning the Whitened inputs: this is standard in analytical work because it enables exact, closed-form solutions. Much of the existing theoretical literature, including Fukumizu (1998), Braun et al. (2022), and Kunin et al. (2024), relies on this assumption for that reason.
>
> > Can you derive optimal task groupings from your framework? Given task statistics, can you predict which tasks should be trained together vs separately?
>
> This is a fundamental and long-standing challenge in MTL, without an immediate solution. Our work contributes a new perspective by showing that the statistical structure of tasks, in particular, their alignment, plays a central role. Based on the current understanding that negative cosine similarities of the gradients are associated with negative transfer, together with our empirical finding that aligned tasks maintain strongly positive cosine values, **a natural future direction is to estimate task alignment and group tasks accordingly**. We highlight this perspective in a new subsection of Appendix (M3).
>
> >  The "conflicting" case seems to cover a huge range of scenarios. Can you provide more granular analysis? When does partial alignment help vs hurt?
>
> We note that a granular analysis of the Multi-MNIST dataset appears in Appendix D.5. We have moved it into the main manuscript for greater visibility (M1). Importantly, determining when tasks help or hurt is far more nuanced than a binary distinction. **As we show, this relationship is dynamic throughout training and depends on the underlying task structure and alignment, as well as on the notion of negative transfer** (which remains theoretically underexplored).
>
> > What's the computational cost of computing these theoretical predictions? Could they be used in practice to guide training?
>
> **We believe our insights provide a principled foundation for future work, using the SVD-based characterization of tasks as a starting point for more theoretically grounded, and potentially far less computationally expensive,weighting strategies.** We added a section outlining this intuition to help guide future research (M3), as designing an explicit weighting algorithm lies beyond the scope of the present paper.

---

### Official Review · Reviewer_6Wtf · 2025-11-01

**Soundness:** 3
**Presentation:** 2
**Contribution:** 2
**Rating:** 4
**Confidence:** 4

**Summary:**

The work extends a Riccati framework to deep linear multi-task learning and provides an analysis of the training dynamics and task interactions. It analyzes how task alignment and magnitude imbalance shape interaction patterns: orthogonal, conflicting, and aligned.

**Strengths:**

1. It categorizes tasks into orthogonal, aligned, and conflicting cases and clarifies, in a principled way, when interference grows.
2. It visualizes how alignment and magnitude imbalance lead to learning delays and changing task importance.

**Weaknesses:**

1. As the authors already note, the analysis is limited to deep linear models, which weakens confidence about transfer to realistic nonlinear systems. Beyond that, the paper offers little guidance on how to operationalize the insights, making scalability and generalizability hard to judge. For example, the computational cost of estimating alignment spectra and applying Riccati-based diagnostics on deep networks with large datasets is unclear. It is also uncertain whether online alignment estimation during multi-task training or optimization is feasible.

2. In practice, across many multi-task settings, relationships between tasks rarely fit neatly into “orthogonal,” “aligned,” or “conflicting.” Different aspects can appear simultaneously to varying degrees. How should we handle such mixed cases in real systems? These aspects are not adequately discussed in the paper.

**Questions:**

Please refer to the Weaknesses section and provide detailed responses addressing each point.

---

> ### Author Response · Authors · 2025-11-20
> **Response to Reviewer's concerns and questions**
>
> We thank the reviewer for the thoughtful feedback and address the concerns below. We believe we have thoroughly addressed all concerns raised by the reviewer, and we kindly ask the reviewer to reconsider their score in light of these clarifications and improvements.
>
> > W1. In practice, across many multi-task settings, relationships between tasks rarely fit neatly into “orthogonal,” “aligned,” or “conflicting.” Different aspects can appear simultaneously to varying degrees. How should we handle such mixed cases in real systems? These aspects are not adequately discussed in the paper}
>
> **We kindly note that we already discussed the case of a real-world dataset with a mix of aligned, conflicting and orthogonal cases in detail in Appendix D5. We agree this deserves more visibility and therefore moved this section to the main manuscript.** We agree that this deserves greater visibility and will move this material into the main manuscript, especially since the camera-ready page limit increases from 9 to 10 pages. In this section, we analyze in detail how our framework can be applied to real-world datasets and demonstrate that our theoretical predictions align closely with the behaviour of a neural network trained on the Multi-MNIST dataset.
>
> > W2. As the authors already note, the analysis is limited to deep linear models, which weakens confidence about transfer to realistic nonlinear systems. Beyond that, the paper offers little guidance on how to operationalize the insights, making scalability and generalizability hard to judge. For example, the computational cost of estimating alignment spectra and applying Riccati-based diagnostics on deep networks with large datasets is unclear. It is also uncertain whether online alignment estimation during multi-task training or optimization is feasible.
>
> While we acknowledge that real-world applications involve far more complexity, theoretical deep learning is still in its early stages. Due to the mathematical challenges involved, most theoretical work, including ours, relies on linear network models. Extending the analytical solutions to more realistic, nonlinear settings remains an open problem, even for single-task learning, let alone multi-task learning. In this paper, we aim to advance the theoretical understanding of multi-task learning, a domain that is significantly less developed than standard deep learning theory. We do so by combining recent theoretical insights from standard deep learning theory with domain-specific aspects of MTL. Many theoretical articles demonstrate the usefulness of the deep linear network setting (e.g., [1,2,3,4] and Appendix A1 for many examples). We included experiments on nonlinear neural networks in the appendix (sections D4 and D5) and show how this affects our results,. Understanding how our mathematical analysis can be fully generalized to nonlinear or more practical setups is, unfortunately, beyond the scope of this paper. Nonetheless, we hope this work contributes meaningfully to the foundation upon which future theoretical progress in multi-task learning can be built.
>
> [1]Braun et. al (2022). Exact learning dynamics of deep linear networks with prior knowledge [2] Dominé et al (2025). From Lazy to Rich: Exact Learning Dynamics in Deep Linear Networks [3] Saxe et al (2019). Mathematical theory of semantic development in deep neural networks [4] Gidel et al. (2019). Implicit Regularization of Discrete Gradient Dynamics in Linear Neural Networks

---

### Official Review · Reviewer_RWSh · 2025-11-02

**Soundness:** 3
**Presentation:** 3
**Contribution:** 3
**Rating:** 6
**Confidence:** 3

**Summary:**

This work studies theoretically the dynamic of inter-task learning in linear multi-task learning (MTL) networks. To this end, the authors write down the gradient flow equations of the MTL network using the Riccati formulation and derive from it a closed-form solution for the dynamics of the networks at any step. Then, the authors use these equations to simulate the dynamics of the network under different relationships across tasks (aligned, conflicted, and orthogonal) as well as the effect of having tasks with relatively different magnitudes. Interestingly, the authors also use this approach to obtain theoretical results and simulations on the representations obtained by this linear MTL networks and compare them with their single-task counterparts.

**Strengths:**

- **S1.** The manuscript seem to be theoretically sound and technically involved. I have not looked into the proofs, but I am pleased to see that the authors corroborated that their findings match a linear MTL in the appendix.
- **S2.** While technical, the paper is well written and can be easily followed with a bit of patience.
- **S3.** Experiments are rather extensive given the assumptions on the model, and the conclusions drawn while expected (from my view, these conclusions were relatively known from empirical results over the years), it is great to obtain a theoretical validation.
- **S4.** I find particularly interesting the results regarding the representations learned by the models in the MTL setup.

**Weaknesses:**

- **W1.** My main concern is the extent to which the assumptions made make the setting interesting. In particular, the assumption that concerns me the most is assumption A3: It is relatively well known in the community that if there is no bottleneck the model will learn all tasks one way or another. The more interesting set-up is when there is a bottleneck and the model needs to learn to use a shared representation across tasks. The linearity of the model also concerns me, but to a lesser extent.
- **W2.** Another concern is the novelty of the theoretical results. In particular, the article constantly references for the results and proofs to Braun et al. (2022). I'd appreciate it if the authors could make clear what is the contribution of this work in relation with the aforementioned work.
- **W3.** I find the plots particularly difficult to read. In particular, I am unable to give meaning to those as in Figure 3. I understand that each line represents the evolution over time of one entry of the matrix $W_iW_s$. However, I cannot interpret these values myself.
- **W4.** Maybe due to the previous point, I do not see how the authors can draw statements regarding the temporal component of the dynamics. For example, in line 395 where is says "Over time, a task that was initially harmless may later interfere with or suppress the learning of others."
- **W5.** It would be nice to have a section on related work on MTL analysis as well as in MTL techniques to alleviate gradient conflict (such as FAMO, PCGrad, GradNorm, RotoGrad, among many others). Moreover, it would be a great plus to see whether any of these techniques helps with the linear/ReLU MTL examples that appear in the appendix.

**Questions:**

- **Q1.** How does the overlap in definition 2.1 with the cosine similarity between task gradients, which is the de-facto measure used in MTL works?
- **Q2.** Eq. (11) is rather unclear. Are the $i$ and $j$ powers? Is the $0$ supposed to be a $o$?

Other feedback:
- Regarding the motivation in the second paragraph of the intro, I'd like to share with the authors [this recent study](https://arxiv.org/abs/2505.10347) that points out at the reasons of why _sometimes_ simple heuristics work as well as other MTL optimizers.
- Similarly, I'd also like to share [this recently accepted paper](https://web3.arxiv.org/abs/2510.18258) where the study also the effect of tasks in MTL through the lens of neural tangent kernel methods.
- You are sharing the same label across equations, and sometimes point to the appendix instead that to the main content.
- I assume the "+" in line 263 is supposed to be a "-".
- The caption of Fig. 4 needs to be updated.

---

> ### Author Response · Authors · 2025-11-21
> **Response to the Reviewer’s Concerns**
>
> We thank the reviewer for the valuable feedback and positive assessment of our contributions. We address each concern below and respond to the questions in a separate comment. In the revised manuscript, we moved the Multi-MNIST analysis into the main text and  updated Figure 4 to include the evolution of cosine similarities of shared and task-specific gradients across aligned, conflicting, and orthogonal settings, further illustrating how our theory manifests in practice (as highlighted in the general comment to the reviewers M1–M4).
>
> > W1. My main concern is the extent to which the assumptions made make the setting interesting.
>
> To better understand the impact of architecture and the zero-balanced assumption, we refer to [1], where the zero-balanced assumption is relaxed to $\lambda$-balanced initializations and the architectural effects under this setup are analysed (see their Fig. 5). They show that different $\lambda$ values yield different learning regimes of the network, and enable a transition between the rich and lazy regimes. Furthermore, their analysis shows that the architecture directly affects how the sign of the relative scale (determined by the $\lambda$-value) is related to the different learning regimes (i.e., depending on the architecture, different $\lambda$ values will put the network in the lazy or rich regime). **While this is important for understanding single-task learning behaviour, placing the MTL model in the lazy regime (which happens when we move away from the zero-balancedness) would mainly suppress feature learning rather than illuminate MTL-specific effects.** Furthermore, a recent workshop paper [2] investigated the consequences of moving beyond the bottleneck assumption (A3). They found that the advantages of removing this assumption are, in fact, quite limited; what plays a much more decisive role is the effective rank of the model, which directly governs its expressivity.
>
> > W2. Another concern is the novelty of the theoretical results
>
> We clarify that our theoretical contribution does not lie in the lemmas or equations, **but in extending their framework to the MTL setting and deriving new structural relationships between task alignment and task interactions**. This provides a new perspective for understanding interaction dynamics in multi-task neural networks.
>
> **Concerning the mathematical machinery, the novelty consists of showing that the solutions of the Riccati equation presented by Braun et. al (2022) still hold** when drawing an explicit difference between task-specific output layers; to this end, we had to rederive the second assumption (the balancedness assumption) to see under which assumptions this solution could be extended to this setup. We saw that it is enough to change the STL version of the zero-balancedness assumption to:
> $W_S^TW_S=W_1W_1^T+W_2W_2^T$. Additionally, we introduced $\mathbf{\xi}$  and $W_{MTL}^0$ and their SVD, which allowed for the direct analysis of the impact of task alignment on the resulting dynamics where the combined spectrum plays a central role. Finally, we extended the convergence theorem to the case with multiple tasks, which allows for a comparison between learned representations obtained in MTL and STL without any training. Additionally, we derive solutions for Tabula Rasa initialisations in Appendix B3 and perform a theoretical analysis.
>
> > W3&W4 . I find the plots particularly difficult to read. In particular, I am unable to give meaning to those as in Figure 3. I understand that each line represents the evolution over time of one entry of the matrix. I do not see how the authors can draw statements regarding the temporal component of the dynamics. For example, in line 395 where is says "Over time, a task that was initially harmless may later interfere with or suppress the learning of others."
>
> We explain the plots in a separate comment related to the questions, but note that we clarified the point made in line 395 in the revised manuscript.
>
> > W5. It would be nice to have a section on related work on MTL analysis as well as in MTL techniques to alleviate gradient conflict (such as FAMO, PCGrad, GradNorm, RotoGrad, among many others). Moreover, it would be a great plus to see whether any of these techniques helps with the linear/ReLU MTL examples that appear in the appendix.
>
> **We would like to kindly note that a discussion of dynamic-weighting algorithms is already included in the additional related work section of the appendix (Sections A2 and A3)**. We will add a few more examples of such algorithms in the camera-ready version. To highlight the connection between alignment and MTL techniques that mitigate gradient conflicts, **we have added plots of gradient cosine similarities to Figure 4 (M2), showing how different task alignments produce different cosine similarity patterns**. These results suggest that such MTL techniques may benefit conflicting and orthogonal tasks more than aligned ones.

---

> ### Author Response · Authors · 2025-11-21
> **Response to the Reviewer's questions**
>
> We thank the reviewer again for the valuable feedback. Below, we address each question individually and respond to the concerns in separate comments. We also appreciate the additional references and the attention to typographical errors; these have been corrected in the revised version.
>
> > Q1. How does the overlap in definition 2.1 with the cosine similarity between task gradients, which is the de-facto measure used in MTL works?
>
> This is an interesting question. **To examine the link between task alignment and common interaction measures (e.g., gradient cosine similarity and norms), we plotted their evolution for aligned, conflicting, and orthogonal tasks (added to Figure 4 in the manuscript, M2)**. The Figure shows that: perfectly aligned tasks show consistently high cosine values, reflecting strong gradient similarity and essentially no negative transfer due to conflicting gradients. In contrast, conflicting and orthogonal tasks exhibit alternating positive and negative cosine values, indicating periods of negative transfer under current definitions. This indicates that the benefit of dynamic weighting algorithms such as PCGrad or CAGrad might depend on the degree of alignment present in the data.
>
> > Q2. Eq. (11) is rather unclear. Are the $i$ and $j$ powers? Is the $0$ supposed to be a $o$?
>
> The $i$ and $j$ are not exponents but indices referring to the dimensions of the task correlation matrices we compare. We agree that the notation can be improved. For clarity, we will change it to $\Delta M^{i,j}_{12}$. The "0'' should indeed have been an "o''. We apologize for the confusion and have corrected this in the revised version. As explained in the manuscript, this indicates the amount of magnitude imbalance between the dimensions $i$ and $j$ of two tasks $T_1$ and $T_2$.
>
> > Q3. I find the plots particularly difficult to read. In particular, I am unable to give meaning to those as in Figure 3. I understand that each line represents the evolution over time of one entry of the matrix. However, I cannot interpret these values myself. Maybe due to the previous point, I do not see how the authors can draw statements regarding the temporal component of the dynamics. For example, in line 395 where is says "Over time, a task that was initially harmless may later interfere with or suppress the learning of others."
>
> The purpose of Figure 3 is to illustrate how the characteristic sigmoidal behaviour of linear networks emerges in the multi-task setting.
>
> From single-task learning, it is well understood that in a two-layer linear network, singular modes are learned sequentially, from the strongest to the weakest mode. This leads to the familiar sigmoidal evolution of the network function: intervals of no learning (when no new mode is being acquired), interspersed with phases of rapid progress (when a singular mode is picked up).
>
> Our plot extends this intuition to the multi-task setting. **The key question we address is how the task-specific network functions evolve relative to one another when multiple tasks are trained simultaneously. Specifically, the curves allow us to examine whether the tasks pick up singular modes concurrently or sequentially, and how this behavior is shaped by their alignment.**
>
> When comparing Figures 3(b) and 3(c), the key difference is in how the curves evolve over time. For orthogonal tasks, the singular modes of Task 2 project only weakly onto the shared space, so the solid curves (Task 1) converge well before Task 2. This shows that learning one task can leave the other largely unaffected for a substantial period. With an additional magnitude imbalance, this effect becomes even more pronounced.
>
> When the magnitude imbalance is removed (Figure 3f), the network functions of the two tasks remain disentangled, but the time lag disappears because both tasks now possess singular modes of comparable strength. Although the resulting trajectories may superficially resemble those of conflicting tasks, the underlying mechanism is fundamentally different, as is also evident from the learned representations in Figure 5.
>
> The dynamical nature of the interactions between tasks and the impact of task alignment is further illustrated by the additional plots of cosine similarities of the gradients we added to Figure 4 (M2). We added more explanation to the paragraph concerning "Motivation to weight tasks dynamically" in the revised manuscript.

---

### Author Response · Authors · 2025-11-20
**General Response to Reviewers and Overview of Revisions**

First of all, we would like to thank all reviewers for their valuable feedback and insights. As one reviewer noted, “This work is definitely a step towards providing a better understanding of MTL learning dynamics, and will hopefully pave the way for practical advances.” We fully agree with this perspective and believe that the theoretical insights we provide fills an important gap in the current literature and offers a foundation on which practical methods can be built.

**To further clarify how our theoretical insights can inform practical developments, we have made the following modifications (M1-M4) to the paper**:
- **M1. We have moved our analysis of the real-world Multi-MNIST dataset (previously in Appendix D.5) into the main manuscript. This example demonstrates how realistic datasets, whose tasks are neither perfectly aligned, conflicting, nor orthogonal, can still be analyzed within our framework**, and how the resulting behavior of a neural network trained on these tasks aligns with our theoretical predictions.
- **M2. We updated Figure 4 to include the evolution of cosine similarities of shared and task-specific gradients across aligned, conflicting, and orthogonal settings, further illustrating how our theory manifests in practice.** We see that the alignment impacts the cosine similarity between gradients during training, which is currently seen as a sign of negative interference in the MTL community. Building on this insight, we expect that complex dynamic weighting algorithms (PCgrad, CaGrad) will provide comparatively smaller gains when tasks are strongly aligned, and potentially much larger improvements in settings with conflicting or orthogonal tasks, where gradient interactions are more challenging.
- **M3. We added a subsection to the Appendix detailing the practical implications of our work**, explaining how our analysis can inform the design of dynamic weighting algorithms and task grouping strategies.
- **M4.** We add a sentence to explain Figure 3 in more detail.

We hope these changes will alleviate the reviewers’ concerns and lead to a more favourable assessment of the work. We address individual concerns and questions below.

---

### Author Response · Authors · 2025-12-03
**Message to the Area Chair**

Dear Area Chair,

We would like to highlight three points concerning the rebuttal:
-   During the first week, we made significant improvements to the paper to address the raised concerns. In particular, we added an additional page to the main manuscript with an application to a real-world dataset and clarified the practical implications. We believe these additions successfully resolve the reviewers’ concerns regarding practical insights and the relevance of our work.
-  We provided detailed arguments regarding the assumptions in our analysis, showing that our core insights remain valid even when these assumptions are relaxed.
- Some scores are primarily influenced by the fact that our paper does not propose a new algorithm for task grouping/dynamic loss weighting. However, the goal of our work is to advance theoretical understanding, a type of contribution that is comparatively rare in the MTL literature and, we believe, valuable.

We kindly ask to take these points into consideration, and we thank the AC for the time and effort devoted to the evaluation.

---

### Meta-Review · Area_Chair_e2fD · 2026-01-07

**Summary:**

1. The setting/assumptions are too restrictive (deep linear networks, whitened inputs, balanced/zero-balanced init, MSE-only, and in some cases no bottleneck), so it’s unclear how relevant the results are to real nonlinear MTL.
2. Theoretical novelty seems limited because the work largely extends Braun et al. (2022)’s Riccati framework with similar machinery/proof structure.
3. The paper lacks actionable/algorithmic contributions: it motivates dynamic loss weighting but does not propose a concrete adaptive weighting method or show how to operationalize insights.
4. The “aligned/orthogonal/conflicting” taxonomy is too coarse for real data, where task relationships are mixed and evolve over training.
5. Figures/plots are hard to interpret.
6. There is little guidance on practicality/scalability: the computational cost and feasibility of estimating alignment spectra/diagnostics (possibly online) on large deep networks is unclear.
7. The analysis is shallow/purely descriptive in places (e.g., it reports phenomena like non-monotonic curves under conflict without a deeper “why”), and misses some additional analyses.
8. Some technical clarity issues exist (notation ambiguity, unclear equation indexing/typos, overlap definition vs cosine similarity, unclear Eq. (11) indexing).

**Reviewer Concerns:**

1. Regarding the setting, the authors have clarified the limitations of the technical assumptions and mentioned recent work relaxing some of these assumptions.
2. Regarding the technical contribution, the authors have clarified that the technical novelty lies in extending Braun et al. (2022)'s framework to the MTL case and introducing task-alignment structure.
3. Regarding the lack of algorithmic contributions, the authors emphasize that the goal of the paper is to provide fundamental theoretical insights rather than a ready-to-use algorithm. In addition, the authors have added discussion on practical implications (task grouping and when dynamic weighting may help) and plots of gradient cosine similarities.
4. Regarding real data, the authors propose to move the analysis of the real-world Multi-MNIST dataset into the main text, which demonstrates that the proposed framework can handle real-world datasets where the tasks are neither perfectly aligned, conflicting, nor orthogonal.
5. Regarding the figures, the authors have improved the interpretation of the plots.
6. Regarding guidance on practicality, the authors clarify that their work provides a principled foundation for future work by using the SVD-based characterization of tasks as a starting point for more theoretically grounded and potentially far less computationally expensive strategies.
7. For deeper analyses, the authors largely defer them as future directions, so these are still outstanding concerns.

**Reviewer Scores:**

1. For Reviewer RWSh, the concerns have been mostly addressed, so their score might remain 6 or potentially be increased to 7.
2. For Reviewer 6Wtf, the concern about the mixed multi-task case should have been addressed as the authors emphasized their analysis of the Multi-MNIST dataset. The reviewer's concern about practical guidance is not fully addressed. Thus the reviewer's score might be slightly increased to 5.
3. For Reviewer HJRd, the concerns about the setting, technical contributions, and experimental limitations should have been addressed by the rebuttal. While the questions about the missing in-depth analyses are not fully addressed so far and require substantial work. Thus the reviewer's score might be increased to 4 or 5.
4. The Reviewer 15xH's concerns are mostly addressed by the rebuttal, so their score might remain 6 or slightly be increased to 7.

---

### Decision · Program_Chairs · 2026-01-26

Reject